# Study design features increase replicability in brain-wide association studies

Kaidi Kang[1✉], Jakob Seidlitz[2,3,4], Richard A. I. Bethlehem[5], Jiangmei Xiong[1], Megan T. Jones[1], Kahini Mehta[3,4,6], Arielle S. Keller[7,8], Ran Tao[1,9], Anita Randolph[10,11], Bart Larsen[10,11], Brenden Tervo-Clemmens[11,12], Eric Feczko[10,11], Oscar Miranda Dominguez[10,11], Steven M. Nelson[10,11], Lifespan Brain Chart Consortium*, Jonathan Schildcrout[1], Damien A. Fair[10,11,13], Theodore D. Satterthwaite[3,4,6], Aaron Alexander-Bloch[2,3,4,14] & Simon Vandekar[1,14✉]

Brain-wide association studies (BWAS) are a fundamental tool in discovering brain–behaviour associations[1,2]. Several recent studies have shown that thousands of study participants are required for good replicability of BWAS[1–3]. Here we performed analyses and meta-analyses of a robust effect size index using 63 longitudinal and cross-sectional MRI studies from the Lifespan Brain Chart Consortium[4] (77,695 total scans) to demonstrate that optimizing study design is critical for increasing standardized effect sizes and replicability in BWAS. A meta-analysis of brain volume associations with age indicates that BWAS with larger variability of the covariate and longitudinal studies have larger reported standardized effect size. Analysing age effects on global and regional brain measures from the UK Biobank and the Alzheimer's Disease Neuroimaging Initiative, we showed that modifying study design through sampling schemes improves standardized effect sizes and replicability. To ensure that our results are generalizable, we further evaluated the longitudinal sampling schemes on cognitive, psychopathology and demographic associations with structural and functional brain outcome measures in the Adolescent Brain and Cognitive Development dataset. We demonstrated that commonly used longitudinal models, which assume equal between-subject and within-subject changes can, counterintuitively, reduce standardized effect sizes and replicability. Explicitly modelling the between-subject and within-subject effects avoids conflating them and enables optimizing the standardized effect sizes for each separately. Together, these results provide guidance for study designs that improve the replicability of BWAS.

BWAS use non-invasive MRI to identify associations between inter-individual differences in behaviour, cognition, biological or clinical measurements and brain structure or function[1,2]. A fundamental goal of BWAS is to identify true underlying biological associations that improve our understanding of how brain organization and function are linked to health across the lifespan.

Recent studies have raised concerns about the replicability of BWAS[1–3]. Statistical replicability is typically defined as the probability of obtaining consistent results from hypothesis tests across different studies. Like statistical power, replicability is a function of both the standardized effect size and the sample size[5–7]. Low replicability in BWAS has been attributed to a combination of small sample sizes, small standardized effect sizes and bad research practices (such as p-hacking and publication bias)[1,2,8–12]. The most obvious solution to increase the replicability in BWAS is to increase study sample sizes. Several recent studies have shown that thousands of study participants are required to obtain replicable findings in BWAS[1,2]. However, massive sample sizes are often infeasible in practice.

Standardized effect sizes (such as Pearson's correlation and Cohen's d) are statistical values that not only depend on the underlying biological association in the population but also on the study design. Two studies of the same biological effect with different study designs will have

[1]Department of Biostatistics, Vanderbilt University Medical Center, Nashville, TN, USA. [2]Department of Child and Adolescent Psychiatry and Behavioral Sciences, The Children's Hospital of Philadelphia, Philadelphia, PA, USA. [3]Department of Psychiatry, University of Pennsylvania, Philadelphia, PA, USA. [4]Lifespan Brain Institute of The Children's Hospital of Philadelphia and Penn Medicine, Philadelphia, PA, USA. [5]Department of Psychology, University of Cambridge, Cambridge, UK. [6]Penn Lifespan Informatics and Neuroimaging Center (PennLINC), Perelman School of Medicine, University of Pennsylvania, Philadelphia, PA, USA. [7]Department of Psychological Sciences, University of Connecticut, Mansfield, CT, USA. [8]Institute for the Brain and Cognitive Sciences, University of Connecticut, Mansfield, CT, USA. [9]Vanderbilt Genetics Institute, Vanderbilt University Medical Center, Nashville, TN, USA. [10]Department of Pediatrics, University of Minnesota Medical School, Minneapolis, MN, USA. [11]Masonic Institute for the Developing Brain, University of Minnesota, Minneapolis, MN, USA. [12]Department of Psychiatry and Behavioral Sciences, University of Minnesota Medical School, Minneapolis, MN, USA. [13]Institute of Child Development, University of Minnesota, Minneapolis, MN, USA. [14]These authors contributed equally: Aaron Alexander-Bloch, Simon Vandekar. *A list of authors and their affiliations appears at the end of the paper. A full list of members and their affiliations appears in the Supplementary Information. ✉e-mail: kaidi.kang@vanderbilt.edu; simon.vandekar@vumc.org

different standardized effect sizes. For example, contrasting brain function of groups with depression versus those without depression will have a different Cohen's *d* effect size if the study design measures more extreme depressed states contemporaneously with measures of brain function, as opposed to less extreme depressed states, even if the underlying biological effect is the same. Although researchers cannot increase the magnitude of the underlying biological association, its standardized effect size – and thus its replicability – can be increased by critical features of study design.

In this study, we focus on identifying modifiable study design features that can be used to improve the replicability of BWAS by increasing standardized effect sizes. Increasing standardized effect sizes through study design before data collection stands in contrast to bad research practices that can artificially inflate reported effect sizes, such as *p*-hacking and publication bias. There has been very little research regarding how modifications to the study design might improve BWAS replicability. Specifically, we focus on two major design features that directly influence standardized effect sizes: variation in sampling scheme and longitudinal designs[1,13–15]. Of note, these design features can be implemented without inflating the sample estimate of the underlying biological effect when using correctly specified models[16]. By increasing the replicability of BWAS through study design, we can more efficiently utilize the US$1.8 billion average annual investment in neuroimaging research from the US National Institutes of Health (https://reporter.nih.gov/search/_dNnH1VaiEKU_vZLZ7L2xw/projects/charts).

Here we conducted a comprehensive investigation of cross-sectional and longitudinal BWAS designs by capitalizing on multiple large-scale data resources. Specifically, we begin by analysing and meta-analysing 63 neuroimaging datasets including 77,695 scans from 60,900 cognitively normal participants from the Lifespan Brain Chart Consortium[4] (LBCC). We leverage data from the UK Biobank (UKB; up to 29,031 scans), the Alzheimer's Disease Neuroimaging Initiative (ADNI; 2,232 scans) and the Adolescent Brain Cognitive Development study (ABCD; up to 17,210 scans) to investigate the most commonly measured phenotypes of brain structure and function. To ensure that our results are broadly generalizable, we evaluated associations with diverse covariates of interest, including age, sex, cognition and psychopathology. To facilitate comparison between BWAS designs, we also introduce a new version of the robust effect size index (RESI)[17–19] that allows us to demonstrate how longitudinal study design directly impacts standardized effect sizes.

## Standardized effect sizes depend on study design

To fit each study-level analysis, we regressed each of the global brain measures (total grey matter volume (GMV), total subcortical grey matter volume (sGMV), total white matter volume (WMV) and mean cortical thickness) and regional brain measures (regional GMV and cortical thickness, based on Desikan–Killiany parcellation[20]) on sex and age in each of the 63 neuroimaging datasets from the LBCC. Age was modelled using a non-linear spline function in linear regression models for the cross-sectional datasets and generalized estimating equations (GEEs) for the longitudinal datasets (Methods). Site effects were removed before the regressions using ComBat[21,22] (Methods). Analyses for total GMV, total sGMV and total WMV used all 63 neuroimaging datasets (16 longitudinal; Supplementary Table 1). Analyses of regional brain volumes and cortical thickness used 43 neuroimaging datasets (13 longitudinal; Methods and Supplementary Table 2).

Throughout the present study, we used the RESI[17–19] as a measure of standardized effect size. The RESI is a recently developed index that is equal to 1/2 Cohen's *d* under the same assumptions for Cohen's *d*[17] (Methods; section 3 in Supplementary Information). We used the RESI as a standardized effect size because it is broadly applicable to many types of models and is robust to model misspecification.

To investigate the effects of study design features on the RESI, we performed meta-analyses for the four global brain measures and two regional brain measures in the LBCC to model the association of study-level design features with standardized effect sizes. Study design features are quantified as the sample mean, standard deviation and skewness of the age covariate as non-linear terms, and a binary variable indicating the design type (cross-sectional or longitudinal). After obtaining the estimates of the standardized effect sizes of age and sex in each analysis of the global and regional brain measures, we conducted meta-analyses of the estimated standardized effect sizes using weighted linear regression models with study design features as covariates (Methods).

For total GMV, the partial regression plots of the effect of each study design feature demonstrate a strong cubic-shape relationship between the standardized effect size for total GMV–age association and study population mean age. This cubic shape indicates that the strength of the age effect varies with respect to the age of the population being studied. The largest age effect on total GMV in the human lifespan occurs during early and late adulthood (Fig. 1a and Supplementary Table 3). There is also a strong positive linear effect of the study population standard deviation of age and the standardized effect size for total GMV–age association. For each unit increase in the standard deviation of age (in years), the expected standardized effect size increases by about 0.1 (Fig. 1a). This aligns with the well-known relationship between correlation strength and covariate standard deviation indicated by statistical principles[23]. Plots for total sGMV, total WMV and mean cortical thickness show U-shaped changes of the age effect with respect to the study population mean age (Fig. 1b–d). A similar but sometimes weaker relationship is shown between expected standardized effect size and study population standard deviation and skewness of the age covariate (Fig. 1b–d and Supplementary Tables 4–6). Finally, the meta-analyses also show a moderate effect of study design on the standardized effect size of age on each of the global brain measures (Fig. 1a–d and Supplementary Tables 3–6). The average standardized effect size for total GMV–age associations in longitudinal studies (RESI = 0.39) is substantially larger than in cross-sectional studies (RESI = 0.08) after controlling for the study design variables, corresponding to a more than 380% increase in the standardized effect size for longitudinal studies. This value quantifies the systematic differences in the standardized effect sizes between the cross-sectional and longitudinal studies among the 63 neuroimaging studies. Of note, longitudinal study design does not improve the standardized effect size for biological sex, because sex does not vary within participants in these studies (Supplementary Tables 7–10 and Supplementary Fig. 2).

For regional GMV and cortical thickness, similar effects of study design features also occur across regions (Fig. 1e,f; 34 regions per hemisphere). In most of the regions, the standardized effect sizes of age on regional GMV and cortical thickness are strongly associated with the study population standard deviation of age. Longitudinal study designs generally tend to have a positive effect on the standardized effect sizes for regional GMV–age associations and a positive but weaker effect on the standardized effect sizes for regional cortical thickness–age associations. To improve the comparability of standardized effect sizes between cross-sectional and longitudinal studies, we propose a new effect size index: the cross-sectional RESI for longitudinal datasets (section 3 in Supplementary Information). The cross-sectional RESI for longitudinal datasets represents the RESI in the same study population, if the longitudinal study had been conducted cross-sectionally. This newly developed effect size index allows us to quantify the benefit of using a longitudinal study design in a single dataset (section 3.3 in Supplementary Information).

The meta-analysis results demonstrate that standardized effect sizes are dependent on study design features, such as mean age, standard deviation of the age of the sample population, and cross-sectional or longitudinal design. Moreover, the results suggest that modifying

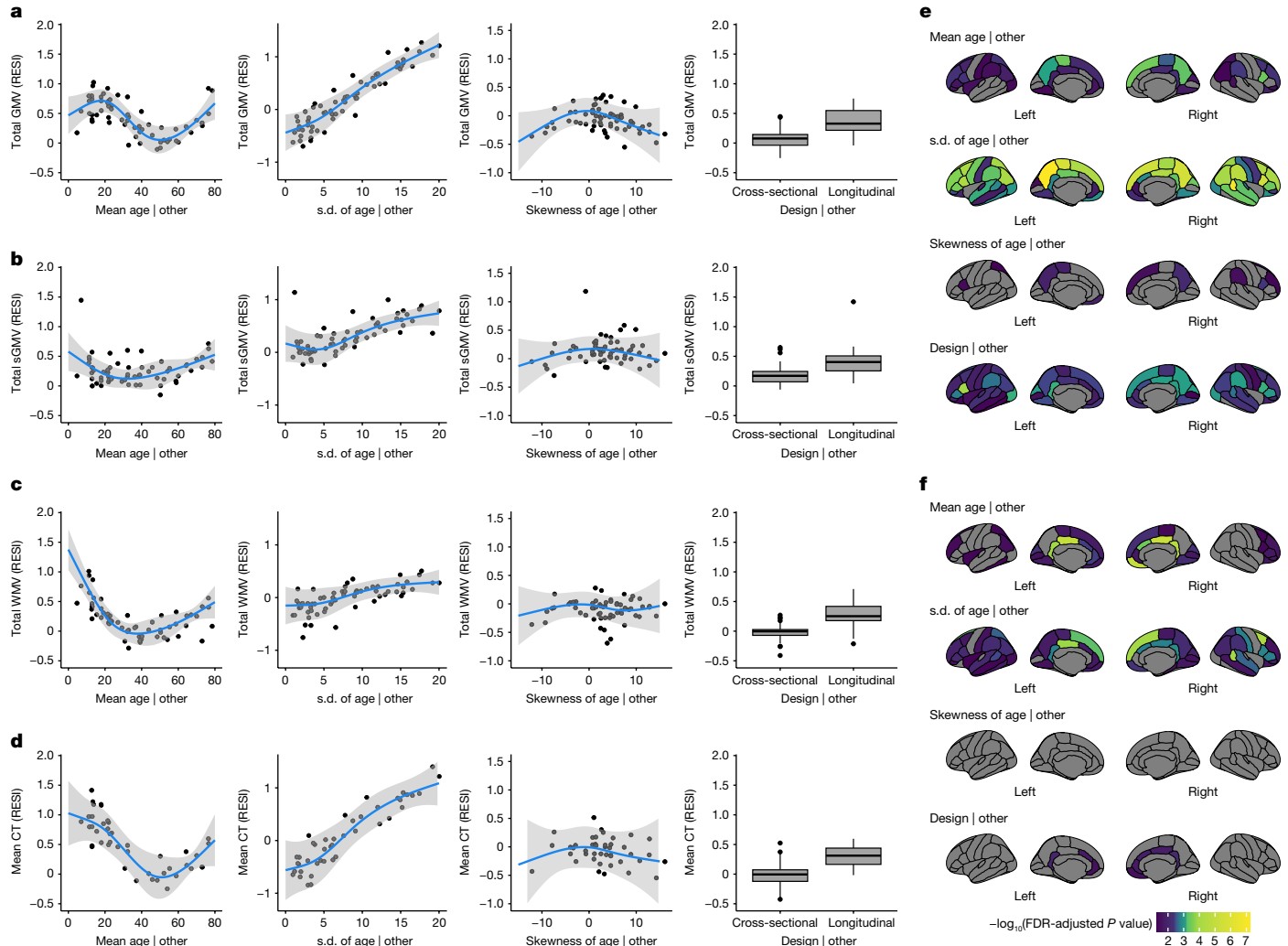

**Fig. 1 | Meta-analyses reveal study design features that are associated with larger standardized effect sizes of age on different brain measures.** **a–d**, Partial regression plots of the meta-analyses of standardized effect sizes (RESI) for the association between age and global brain measures – total GMV (**a**), total sGMV (**b**), total WMV (**c**) and mean cortical thickness (CT) (**d**) – show that standardized effect sizes vary with respect to the mean and standard deviation (s.d.) of age in each study. Box plots show the median (horizontal line), interquartile range (grey box), min–max values (vertical lines), and outliers (points). '|other' means after fixing the other features at constant levels: design was cross-sectional, mean age of 45 years, sample age s.d. = 7 and/or skewness of age = 0 (symmetric). The blue curves are the expected standardized effect sizes for age from the locally estimated scatterplot smoothing (LOESS) curves. The grey shading areas are the 95% confidence bands from the LOESS curves. **e**,**f**, The effects of study design features on the standardized effect sizes for the associations between age and regional brain measures (regional GMV (**e**) and regional cortical thickness (**f**)). Regions with Benjamini–Hochberg-adjusted *P* < 0.05 are shown in colour. FDR, false discovery rate.

study design features, such as increasing variability and conducting longitudinal studies, can increase the standardized effect sizes in BWAS.

## Improved sampling boosts replicability

To investigate the effect of modifying the variability of the age covariate on increasing standardized effect sizes and replicability, we implemented three sampling schemes that produce different sample standard deviations of the age covariate. We treated the large-scale cross-sectional UKB data as the population and draw samples whose age distributions follow a pre-specified shape (bell shaped, uniform and U shaped; Methods and Extended Data Fig. 1a). In the UKB, the U-shaped sampling scheme on age increases the standardized effect size for the total GMV–age association by 60% compared with bell shaped and by 27% compared with uniform (Fig. 2a), with an associated increase in replicability (Fig. 2b). To achieve 80% replicability for detecting the total GMV–age association (Methods), fewer than 100 participants are sufficient if using the U-shaped sampling scheme,

whereas about 200 participants are needed if the bell-shaped sampling scheme is used (Fig. 2b). A similar pattern can be seen for the regional outcomes of GMV and cortical thickness (Fig. 2c–f). The U-shaped sampling scheme typically provides the largest standardized effect sizes of age and the highest replicability, followed by the uniform and bell-shaped schemes. The U-shaped sampling scheme shows greater region-specific improvement in the standardized effect sizes and replicability for regional GMV–age and regional cortical thickness–age associations than the bell-shaped scheme (Extended Data Fig. 1d,e).

To investigate the effect of increasing the variability of the age covariate longitudinally, we implemented sampling schemes to adjust the between-subject and within-subject variability of age in the bootstrap samples from the longitudinal ADNI dataset. In the bootstrap samples, each participant had two measurements (baseline and a follow-up). To imitate the true operation of a study, we selected the two measurements of each participant based on baseline age and the follow-up age by targeting specific distributions for the baseline age and the age change

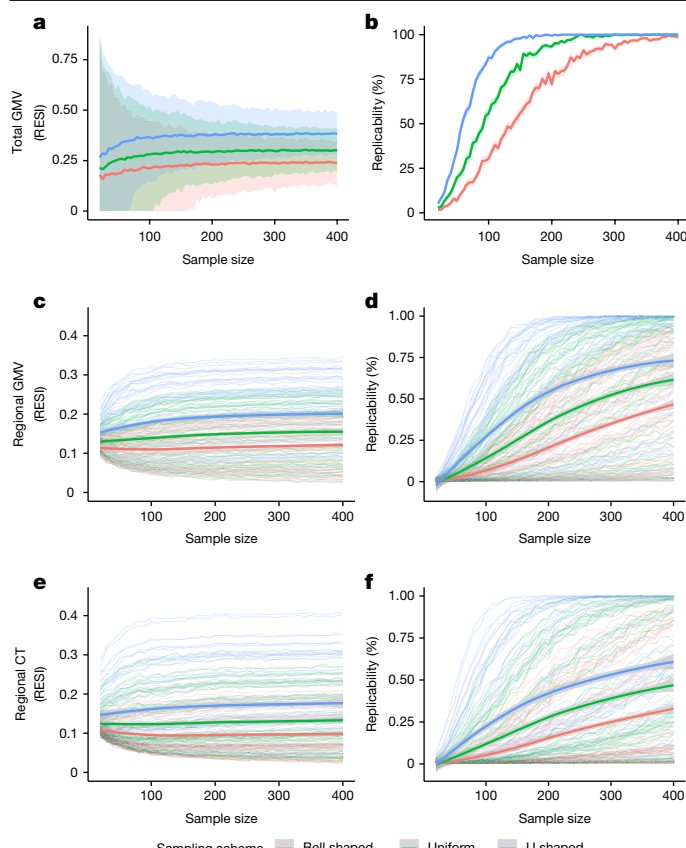

**Fig. 2 | Increased standardized effect sizes and replicability for age associations with different brain measures under three sampling schemes in the UKB study.** *n* = 29,031 for total GMV and *n* = 29,030 for regional GMV and cortical thickness. The sampling schemes target different age distributions to increase the variability of age: bell shaped < uniform < U shaped (Extended Data Fig. 1a). **a**,**b**, Using the sampling schemes, increasing age variability increases the standardized effect sizes (**a**) and replicability (**b**; at significance level of 0.05) for total GMV–age association. **c**–**f**, The same result holds for regional GMV (**c**,**d**) and regional cortical thickness (**e**,**f**). The curves represent the average standardized effect size or estimated replicability at a given sample size and sampling scheme. The shaded areas represent the corresponding 95% confidence bands. The bold curves are the average standardized effect size or replicability across all regions with significant uncorrected effects using the full UKB data (**c**–**f**).

at the follow-up time point (Methods; Extended Data Fig. 1b,c). Increasing between-subject and within-subject variability of age increases the average observed standardized effect sizes, with corresponding increases in replicability (Fig. 3). A U-shaped between-subject sampling scheme on age increases the standardized effect size for total GMV–age association by 23.6% compared with bell shaped and by 12.1% compared with uniform, when using the uniform within-subject sampling scheme (Fig. 3a).

In addition, we investigated the effect of the number of measurements per participant on the standardized effect size and replicability in longitudinal data using the ADNI dataset. Adding a single additional measurement after the baseline increases the standardized effect size for total GMV–age association by 156% and replicability by 350%. The benefit of additional measurements is minimal (Fig. 3c,d). Finally, we also evaluated the effects of the longitudinal sampling schemes on regional GMV and cortical thickness in the ADNI dataset (Fig. 3e–h). When sampling two measurements per participant, the between-subject and within-subject sampling schemes producing larger age variability increase the standardized effect size and replicability across most regions.

Together, these results suggest that having larger spacing in between-subject and within-subject age measurements increases standardized effect size and replicability. Most of the benefit of the number of within-subject measurements is due to the first additional measurement after baseline.

## Sampling benefit varies by brain measure

As standardized effect sizes for brain–age associations are often larger than for brain–behaviour associations, we investigated whether the proposed sampling schemes are effective on various non-brain covariates and their associations with structural and functional brain measures in all participants (with and without neuropsychiatric symptoms) with cross-sectional and longitudinal measurements from the ABCD dataset. The non-brain covariates include the NIH toolbox[24], Child Behavior Checklist (CBCL), body mass index (BMI), birth weight and handedness (Methods; Supplementary Tables 11 and 12). Functional connectivity is used as a functional brain measure and is computed for all pairs of regions in the Gordon atlas[25] (Methods). We used the bell-shaped and U-shaped target sampling distributions to control the between-subject and within-subject variability of each non-brain covariate (Methods). For each non-brain covariate, we show the results for the four combinations of between-subject and within-subject sampling schemes. Overall, there is a consistent benefit to increasing between-subject variability of the covariate (Fig. 4 and Extended Data Fig. 2). These preferred sampling schemes lead to more than 1.8 factor reduction in sample size for 80% replicability and more than 1.4 factor increase in the standardized effect size for over 50% of associations. Moreover, 72% of covariate-outcome associations had increased standardized effect sizes by increasing the between-subject variability of the covariates (Extended Data Fig. 3).

Importantly, increasing within-subject variability decreases the standardized effect sizes for many structural associations (Fig. 4a–f and Extended Data Fig. 2a–f), suggesting that conducting longitudinal analyses can result in decreased replicability compared with cross-sectional analyses. For the functional connectivity outcomes, there is a slight positive effect of increasing within-subject variability (Fig. 4g,h and Extended Data Fig. 2g,h). To evaluate the lower replicability of the structural associations with increasing within-subject variability, we compared cross-sectional standardized effect sizes of the non-brain covariates on each brain measure using the baseline measurements to the standardized effect sizes estimated using the full longitudinal data (Fig. 5a–d and Extended Data Fig. 4). Consistent with the reduction in standardized effect size by increasing within-subject variability, for most structural associations (GMV and cortical thickness), conducting cross-sectional analyses using the baseline measurements results in larger standardized effect sizes (and higher replicability) than conducting analyses using the full longitudinal data. This finding holds when fitting a cross-sectional model only using the 2-year follow-up measurement (Extended Data Fig. 4). Identical results are found using linear mixed models with individual-specific random intercepts, which are commonly used in BWAS (Supplementary Fig. 3). Together, these results suggest that the benefit of conducting longitudinal studies and larger within-subject variability is highly dependent on the brain–behaviour association. Counterintuitively, longitudinal designs can reduce the standardized effect sizes and replicability.

## Accurate longitudinal models are crucial

To investigate why increasing within-subject variability or using longitudinal designs is not beneficial for some associations, we examined an assumption common to GEEs and linear mixed models in BWAS. These widely used models assume that there is consistent association strength between the brain measure and non-brain covariate across between-subject and within-subject changes in the non-brain covariate.

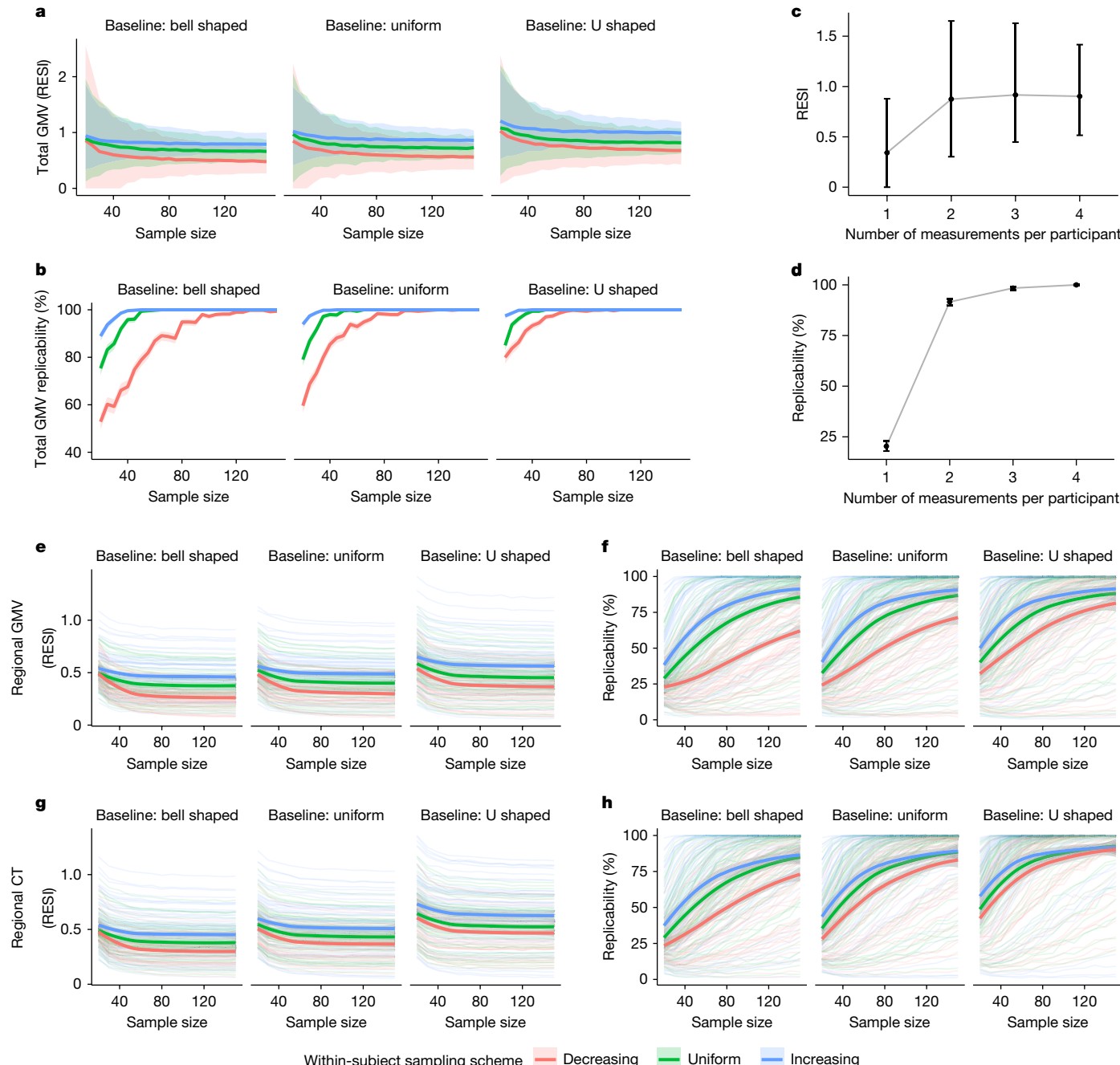

**Fig. 3 | Increased standardized effect sizes and replicability for age associations with structural brain measures under different longitudinal sampling schemes in the ADNI data.** Three different sampling schemes (Extended Data Fig. 1b,c) are implemented in bootstrap analyses to modify the between-subject and within-subject variability of age, respectively. **a,b**, Implementing the sampling schemes results in higher (between-subject and/or within-subject) variability and increases the standardized effect size (**a**) and replicability (**b**; at significance level of 0.05) for the total GMV–age association. The curves represent the average standardized effect size or estimated replicability, and the shaded areas are the 95% confidence bands across the 1,000 bootstraps. **c,d**, Increasing the number of measurements from one to two per participant provides the most benefit on standardized effect size (**c**) and replicability (**d**) for the total GMV–age association when

using uniform between-subject and within-subject sampling schemes and $n = 30$. The points represent the mean standardized effect sizes or estimated replicability, and the whiskers are the 95% confidence intervals. **e**–**h**, Increased standardized effect sizes (**e,g**) and replicability (**f,h**) for the associations of age with regional GMV (**e,f**) and regional cortical thickness (**g,h**) across all brain regions under different sampling schemes. The bold curves are the average standardized effect size or estimated replicability across all regions with significant uncorrected effects using the full ADNI data. The shaded areas are the corresponding 95% confidence bands. Increasing the between-subject and within-subject variability of age by implementing different sampling schemes can increase the standardized effect sizes of age, and the associated replicability, on regional GMV and regional cortical thickness.

However, the between-subject and within-subject association strengths can differ because non-brain measures can be more variable than structural brain measures for various reasons. For example, crystallized composite scores may vary within a participant longitudinally because

of time-of-day effects, lack of sleep or natural noise in the measurement. By contrast, GMV is more precise and it is not vulnerable to other sources of variability that might accompany the crystallized composite score. This combination leads to a low within-subject association

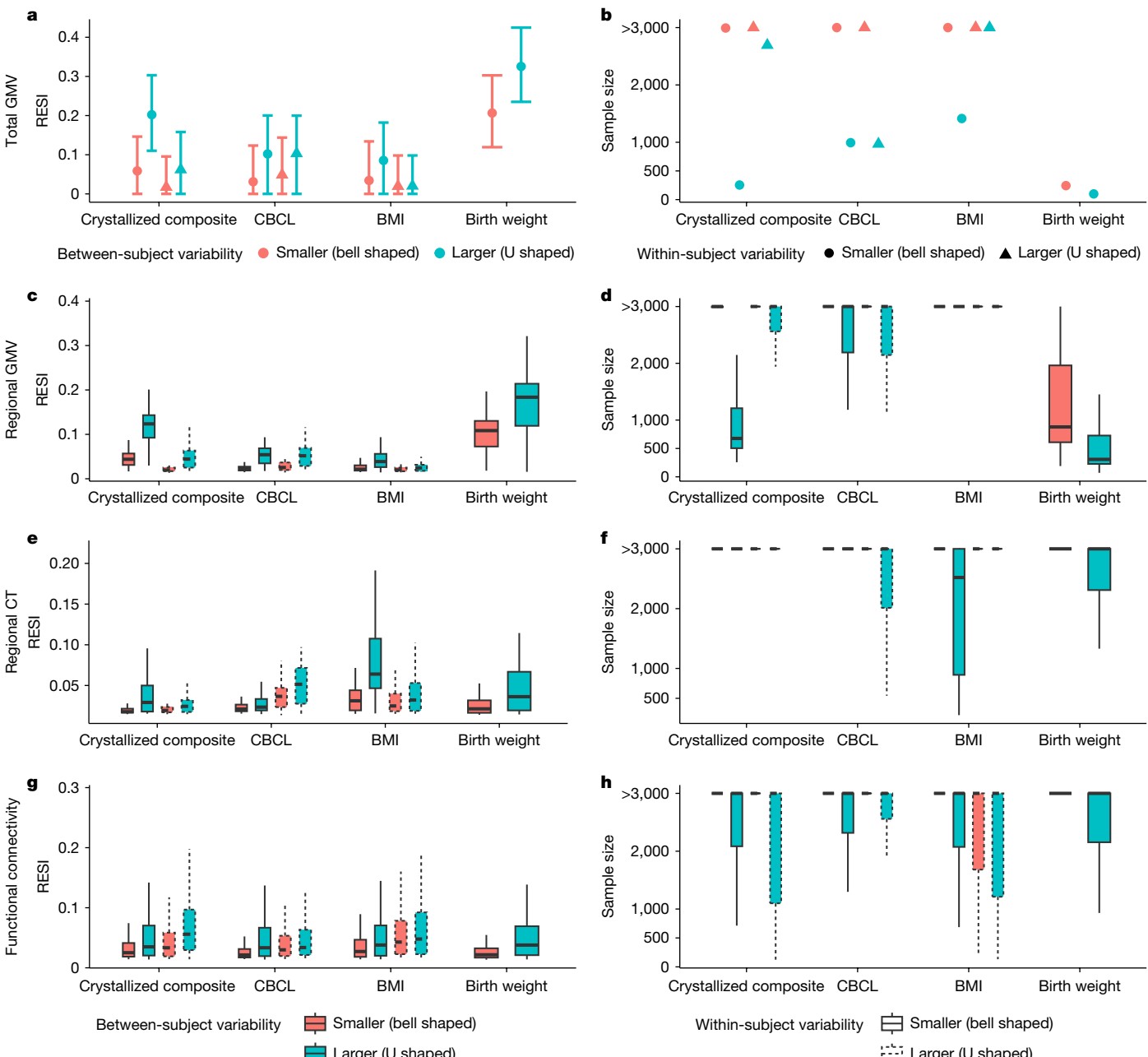

**Fig. 4 | Heterogeneous improvement of standardized effect sizes for select cognitive, mental health and demographic associations with structural and functional brain measures in the ABCD study with bootstrapped samples of n = 500. a,b**, U-shaped between-subject sampling scheme (blue) that increases between-subject variability of the non-brain covariate produces larger standardized effect sizes (**a**) and reduces the number of participants scanned to obtain 80% replicability (**b**) in total GMV. The points and triangles are the average standardized effect sizes across bootstraps, and the whiskers are the 95% confidence intervals. Increasing within-subject sampling (triangles)

can reduce standardized effect sizes. **c–f**, A similar pattern holds in regional GMV (**c,d**) and regional cortical thickness (**e,f**) as in panels **a,b**. The boxplots show the distributions of the standardized effect sizes across regions (or region pairs for functional connectivity). Box plots are as in Fig. 1. **g,h**, By contrast, regional pairwise functional connectivity standardized effect sizes improve by increasing between-subject (blue) and within-subject (dashed borders) variability (**g**) with a corresponding reduction in the number of participants scanned for 80% replicability (**h**). See Extended Data Fig. 2 for the results for all non-brain covariates examined.

between these variables (Supplementary Table 13). Functional connectivity measures are more similar to crystallized composite scores in that they are subject to higher within-subject variability and natural noise, so they have a higher potential for stronger within-subject associations with crystallized composite scores (that is, they are more likely to vary together based on many factors such as time of day and lack of sleep). To demonstrate this, we fitted models that estimated distinct effects for between-subject and within-subject associations in the ABCD dataset (Methods) and found that there are large between-subject

parameter estimates and small within-subject parameter estimates in total and regional GMV (Fig. 5e, Supplementary Table 13 and section 5.2 in Supplementary Information), whereas the functional connectivity associations are distributed more evenly across between-subject and within-subject parameters (Fig. 5f). If the between-subject and within-subject associations are different, these widely used longitudinal models average the two associations (equation (13) in section 5 in Supplementary Information). Fitting these associations separately avoids averaging the larger effect with the smaller effect and can inform

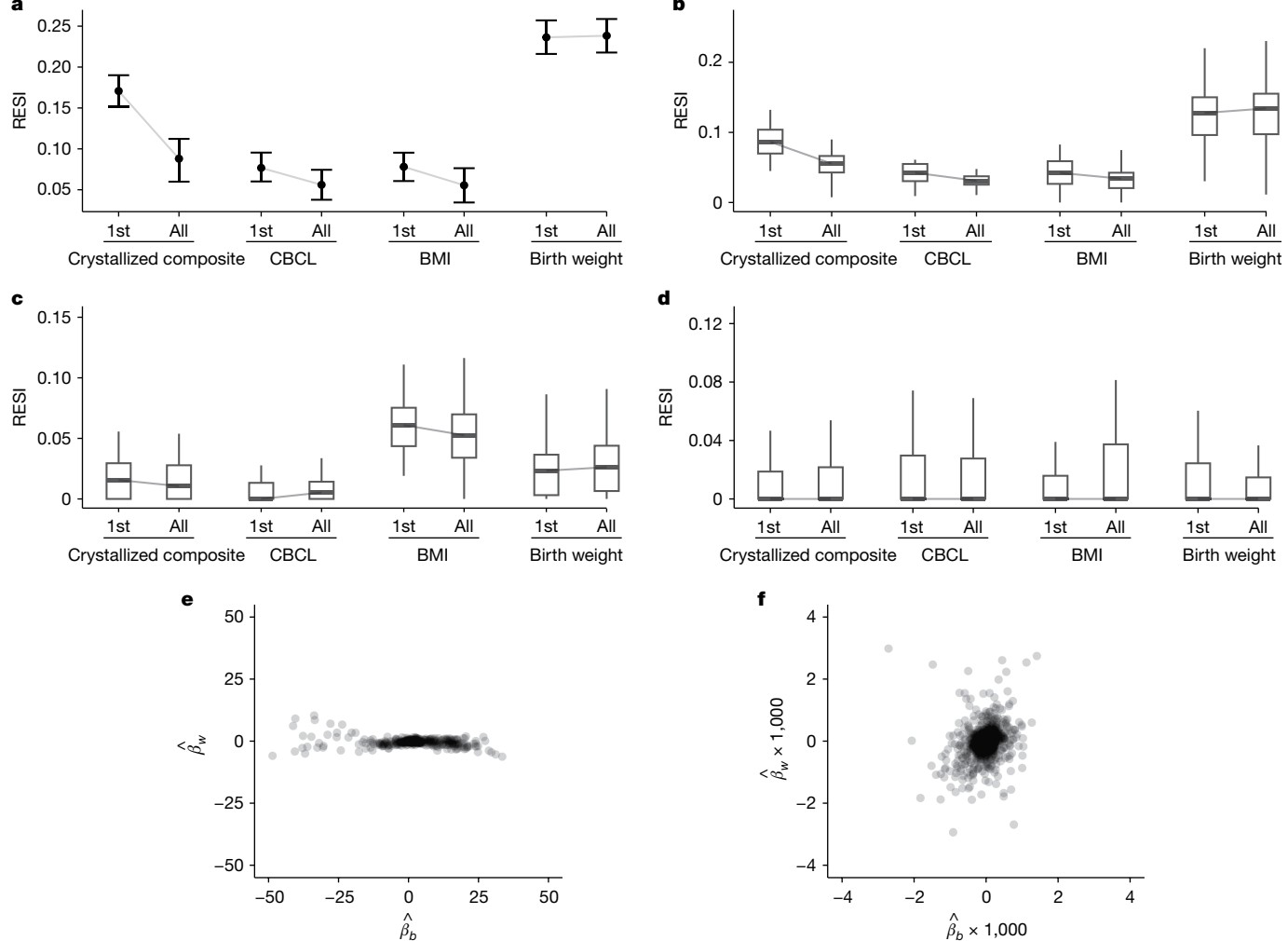

**Fig. 5 | Longitudinal study designs can reduce standardized effect sizes and replicability due to differences in between-subject versus within-subject associations of brain and behavioural measures.** Plots show the distribution of the standardized effect sizes. **a**–**c**, Cross-sectional analyses (using only the baseline measures; indicated by '1st' on the *x* axes) can have larger standardized effect sizes than the same longitudinal analyses (using the full longitudinal data; indicated by 'all' on the *x* axes) for total GMV (**a**), regional GMV (**b**) and regional cortical thickness (**c**) in the ABCD dataset. Data in **a** are estimates (points) with

95% confidence intervals (whiskers). Box plots in **b**–**d** are as in Fig. 1. **d**, Functional connectivity measures do not show such a reduction of standardized effect sizes in longitudinal modelling. See Extended Data Fig. 4 for the results for all non-brain covariates examined. **e**,**f**, Most regional GMV associations (**e**) have larger between-subject parameter estimates ($\beta_b$; *x* axis) than within-subject parameter estimates ($\beta_w$; *y* axis; see equation (13) in Supplementary Information), whereas functional connectivity associations (**f**) show less heterogeneous relationships between the two parameters.

our understanding of brain–behaviour associations (section 5.2 in Supplementary Information). This approach ameliorates the reduction in standardized effect sizes caused by longitudinal designs for structural brain measures in the ABCD (Extended Data Figs. 5 and 6 and section 5 in Supplementary Information). This longitudinal model has a similar between-subject standardized effect size to the cross-sectional model (see 'Estimation of the between-subject and within-subject effects' in the Methods section; Extended Data Fig. 6). In short, longitudinal designs can be detrimental to replicability when the between-subject and within-subject effects differ and the model is incorrectly specified.

## Optimal design considerations

With increasing evidence of small standardized effect sizes and low replicability in BWAS, optimizing study design to increase standardized effect sizes and replicability is a critical prerequisite for progress[1,26]. Our results demonstrate that standardized effect size and replicability can be increased by enriched sampling of participants with small and large

values of the covariate of interest. This is well known in linear models in which the standardized effect size is explicitly a function of the standard deviation of the covariate[23]. We showed that designing a study to have a larger covariate standard deviation increases standardized effect sizes by a median factor of 1.4, even when there is non-linearity in the association, such as with age and GMV (Supplementary Fig. 1). When the association is very non-monotonic – as in the case of a U-shape relationship between covariate and outcome – sampling the tails more heavily could decrease replicability and diminish our ability to detect non-linearities in the centre of the study population. In such a case, sampling to obtain a uniform distribution of the covariate balances power across the range of the covariate and can increase replicability relative to random sampling when the covariate has a normal distribution in the population. Increasing between-subject variability is beneficial in more than 72% of the association pairs that we studied, despite the presence of such non-linearities (Extended Data Fig. 3).

Because standardized effect sizes are dependent on study design, careful design choices can simultaneously increase standardized effect sizes and study replicability. Two-phase, extreme group and

outcome-dependent sampling designs can inform which participants should be selected for imaging from a larger sample to increase the efficiency and standardized effect sizes of brain–behaviour associations[27–33]. For example, given the high degree of accessibility of cognitive and behavioural testing (for example, to be performed virtually or electronically), individuals scoring at the extremes on a testing scale or battery ('phase I') could be prioritized for subsequent brain scanning ('phase II'). When there are multiple covariates of interest, multivariate two-phase designs can be used to increase standardized effect sizes and replicability[34]. Multivariate designs are also needed to stratify sampling to avoid confounding by other sociodemographic variables. Together, the use of optimal designs can increase both standardized effect sizes and replicability relative to a design that uses random sampling[31]. If desired, weighted regression (such as inverse probability weighting) can be combined with optimized designs to estimate a standardized effect size that is consistent with the standardized effect size if the study had been conducted in the full population[34–36]. Choosing highly reliable psychometric measurements or interventions (for example, medications or neuromodulation within a clinical trial)[37–39] may also be effective for increasing replicability. The decision to pursue an optimized design will depend on other practical factors, such as the cost and complexity of acquiring other (non-imaging) measures of interest and the specific translational goals of the research.

## Longitudinal design considerations

In the meta-analysis, longitudinal studies of the total GMV–age associations have, on average, more than 380% larger standardized effect sizes than cross-sectional studies. However, in subsequent analyses, we noticed that the benefit of conducting a longitudinal design is highly dependent on both the between-subject and the within-subject effects. When the between-subject and the within-subject effects are equal and the within-subject brain measurement error is low, longitudinal studies offer larger standardized effect sizes than cross-sectional studies[40] (section 5.1 in Supplementary Information). This combination of equal between-subject and within-subject effects and low within-subject measurement error is the reason that there is a benefit of longitudinal design in the ADNI for the total GMV–age association (Supplementary Fig. 4). Comparing efficiency per measurement supports the approach of collecting two measurements per participant in this scenario (section 5.1 in Supplementary Information).

Longitudinal models offer the unique ability to separately estimate between-subject and within-subject effects. When the between-subject and within-subject effects differ but we still fit them with a single effect, we mistakenly assume they are equal, and the interpretation of that coefficient becomes complicated: the effect becomes a weighted average of the between-subject and within-subject effects whose weights are determined by the study design features (section 5 in Supplementary Information). The apparent lack of benefit of longitudinal designs in the ABCD on the study of GMV associations is because within-subject changes in the non-brain measures are not associated with within-subject changes in the GMV (Fig. 5e and Supplementary Table 13). The smaller standardized effect sizes that we found in longitudinal analyses are due to the contribution from the smaller within-subject effect to the weighted average of the between-subject and within-subject effects (equation (14) in section 5 in Supplementary Information). Fitting the between-subject and within-subject effects separately prevents averaging the two effects (section 5.2 in Supplementary Information). These two effects are often not directly comparable with the effect obtained from a cross-sectional model because they have different interpretations[41–45] (section 5.2 in Supplementary Information). Using sampling strategies to increase between-subject and within-subject variability of the covariate will increase the standardized effect sizes for between-subject and within-subject associations, respectively (Extended Data Fig. 5).

## Design and analysis recommendations

Although it is difficult to provide universal recommendations for study design and analysis, the present study provides general guidelines for designing and analysing BWAS for optimal standardized effect sizes and replicability based on both empirical and theoretical results (Extended Data Figs. 7 and 8). Although the decision for a particular design or analysis strategy may depend on unknown features of the brain and non-brain measures and their associations, these characteristics can be evaluated in pilot data or the analysis dataset (Supplementary Fig. 4 and section 5.2 in Supplementary Information). One general principle that increases standardized effect sizes for most associations is to increase the covariate standard deviation (for example, through two-phase, extreme group and outcome-dependent sampling), which is practically applicable to a wide range of BWAS contexts. Longitudinal designs can be helpful and optimal even when the between-subject and within-subject effects differ, if modelled correctly. Moreover, longitudinal BWAS enable us to study between-subject and within-subject effects separately, and they should be used when the two effects are hypothesized to be different. Although striving for large sample sizes remains important when designing a study, our findings emphasize the importance of considering other design features to improve standardized effect sizes and replicability of BWAS.

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

**Lifespan Brain Chart Consortium**

**Aaron F. Alexander-Bloch**[2,3,4], **Richard A. I. Bethlehem**[5], **Damien A. Fair**[10,11], **Theodore D. Satterthwaite**[3,4,6], **Jakob Seidlitz**[2,3,4] & **Simon Vandekar**[1]

## Methods

### LBCC dataset and processing

The original LBCC dataset included 123,984 MRI scans from 101,457 human participants across more than 100 studies (which include multiple publicly available datasets[46–56]) and was described in previous work[4] (see Supplementary Information and supplementary table S1 from ref. 4). We filtered to the subset of cognitively normal participants whose data were processed using FreeSurfer (v6.1). Studies were curated for the analysis by excluding duplicated observations and studies with fewer than 4 unique age points, sample size less than 20 and/or only participants of one sex. If there were fewer than three participants having longitudinal observations, only the baseline observations were included and the study was considered cross-sectional. If a participant had changing demographic information during the longitudinal follow-up (for example, changing biological sex), only the most recent observation was included. We updated the LBCC dataset with the ABCD release 5, resulting in a final dataset that includes 77,695 MRI scans from 60,900 cognitively normal participants with available total GMV, sGMV and GMV measures across 63 studies (Supplementary Table 1). In this dataset, 74,148 MRI scans from 57,538 participants across 43 studies have complete-case regional brain measures (regional GMV, regional surface area and regional cortical thickness, based on Desikan–Killiany parcellation[20]; Supplementary Table 2). The global brain measure mean cortical thickness was derived using the regional brain measures (see below).

**Structural brain measures.** Details of data processing have been described in our previous work[4]. In brief, total GMV, sGMV and WMV were estimated from T1-weighted and T2-weighted (when available) MRIs using the 'aseg' output from FreeSurfer (v6.0.1). All three cerebrum tissue volumes were extracted from the aseg.stats files output by the recon-all process: 'Total cortical gray matter volume' for GMV; 'Total cerebral white matter volume' for WMV; and 'Subcortical gray matter volume' for sGMV (inclusive of the thalamus, caudate nucleus, putamen, pallidum, hippocampus, amygdala and nucleus accumbens area; https://freesurfer.net/fswiki/SubcorticalSegmentation). Regional GMV and cortical thickness across 68 regions (34 per hemisphere, based on Desikan–Killiany parcellation[20]) were obtained from the aparc.stats files output by the recon-all process. Mean cortical thickness across the whole brain is the weighted average of the regional cortical thickness weighted by the corresponding regional surface areas.

### Preprocessing specific to ABCD

**Functional connectivity measures.** Longitudinal functional connectivity measures were obtained from the ABCD-BIDS community collection, which houses a community-shared and continually updated ABCD neuroimaging dataset available under Brain Imaging Data Structure (BIDS) standards. The data used in these analyses were processed using the abcd-hcp-pipeline (v0.1.3), an updated version of The Human Connectome Project MRI pipeline[57]. In brief, resting-state functional MRI time series were demeaned and detrended, and a generalized linear model was used to regress out mean white matter, cerebrospinal fluid and global signal, as well as motion variables and then band-pass filtered. High-motion frames (filtered frame displacement > 0.2 mm) were censored during the demeaning and detrending. After preprocessing, the time series were parcellated using the 352 regions of the Gordon atlas (including 19 subcortical structures) and pairwise Pearson correlations were computed among the regions. Functional connectivity measures were estimated from resting-state fMRI time series using a minimum of 5 min of data. After Fisher's $z$-transformation, the connectivities were averaged across the 24 canonical functional networks[25], forming 276 inter-network connectivities and 24 intra-network connectivities.

**Cognitive and other covariates.** The ABCD dataset is a large-scale repository aiming to track the brain and psychological development of over 10,000 children 9–16 years of age by measuring hundreds of variables, including demographic, physical, cognitive and mental health variables[58]. We used release 5 of the ABCD study to examine the effect of the sampling schemes on other types of covariates including cognition (fully corrected $T$-scores of the individual subscales and total composite scores of the NIH Toolbox[24]), mental health (total problem CBCL syndrome scale) and other common demographic variables (BMI, birth weight and handedness). For each of the covariates, we evaluated the effect of the sampling schemes on their associations with the global and regional structural brain measures and functional connectivity after controlling for non-linear age and sex (and for functional connectivity outcomes only, mean frame displacement).

For the analyses of structural brain measures, there were three non-brain covariates with fewer than 5% non-missing follow-ups at both 2-year and 4-year follow-ups (that is, the Dimensional Change Card Sort Test, Cognition Fluid Composite and Cognition Total Composite Score; Supplementary Table 11), and only their baseline cognitive measurements were included in the analyses. For the remaining 11 variables (that is, the Picture Vocabulary Test, Flanker Inhibitory Control and Attention Test, List Sorting Working Memory Test, Pattern Comparison Processing Speed Test, Picture Sequence Memory Test, Oral Reading Recognition Test, Crystallized Composite, CBCL, birth weight, BMI and handedness), all of the available baseline, 2-year and 4-year follow-up observations were used. For the analyses of the functional connectivity, only the baseline observations for the List Sorting Working Memory Test were used due to missingness (Supplementary Table 12).

The records with BMI lying outside the lower and upper 1% quantiles (that is, BMI < 13.5 or BMI > 36.9) were considered misinput and replaced with missing values. The variable handedness was imputed using the last observation carried forwards.

### Statistical analysis

**Removal of site effects.** For multisite or multistudy neuroimaging studies, it is necessary to control for potential heterogeneity between sites to obtain unconfounded and generalizable results. Before estimating the main effects of age and sex on the global and regional brain measures (total GMV, total WMV, total sGMV, mean cortical thickness, regional GMV and regional cortical thickness), we applied ComBat[21] and LongComBat[22] in cross-sectional datasets and longitudinal datasets, respectively, to remove the potential site effects. The ComBat algorithm involves several steps including data standardization, site-effect estimation, empirical Bayesian adjustment, removing estimated site effects and data rescaling.

In the analysis of cross-sectional datasets, the models for ComBat were specified as a linear regression model illustrated below using total GMV:

$$\text{GMV} \sim \text{ns(age, d.f.} = 2) + \text{sex} + \text{site},$$

where ns denotes natural cubic splines on 2 d.f., which means that there were two boundary knots and one interval knot placed at the median of the covariate age. Splines were used to accommodate non-linearity in the age effect. For the longitudinal datasets, the model for LongComBat used a linear mixed effects model with participant-specific random intercepts:

$$\text{GMV} \sim (1|\text{participant}) + \text{ns(age, d.f.} = 2) + \text{sex} + \text{site}.$$

When estimating the effects of other non-brain covariates in the ABCD dataset, ComBat was used to control the site effects, respectively, for each of the cross-sectional covariates. The ComBat models were specified as illustrated below using GMV:

$$\text{GMV} \sim \text{ns(age, d.f.} = 2) + \text{sex} + x + \text{site},$$

where $x$ denotes the non-brain covariate. LongComBat was used for each of the longitudinal covariates with a linear mixed effects model with participant-specific random intercepts only:

$$\text{GMV} \sim (1|\text{participant}) + \text{ns}(\text{age, d.f.} = 2) + \text{sex} + x + \text{site}.$$

When estimating the effects of other covariates on the functional connectivity (FC) in the ABCD data, we additionally controlled for the mean frame displacement (FD) of the frames remaining after scrubbing. The longComBat models were specified as:

$$\text{FC} \sim (1|\text{participant}) + \text{ns}(\text{age, d.f.} = 2) + \text{sex}$$
$$+ \text{ns}(\text{mean\_FD, d.f.} = 3 + x + \text{site}.$$

The Combat and LongComBat were implemented using the neuro-Combat[59] and longCombat[60] R packages. Site effects were removed before all subsequent analyses including the bootstrap analyses described below.

**RESI for association strength.** The RESI is a recently developed standardized effect size index that has consistent interpretation across many model types, encompassing all types of test statistics in most regression models[17,18]. In brief, the RESI is a standardized effect size parameter describing the deviation of the true parameter value (or values) $\beta$ from the reference value (or values) $\beta_0$ from the statistical null hypothesis $H_0: \beta = \beta_0$,

$$S = \sqrt{(\beta - \beta_0)^T \Sigma_\beta^{-1} (\beta - \beta_0)},$$

where $S$ denotes the parameter RESI, $\beta$ and $\beta_0$ can be vectors, $T$ denotes the transpose of a matrix, $\Sigma_\beta$ is the covariance matrix for $\sqrt{N}\hat{\beta}$ (where $\hat{\beta}$ is the estimator for $\beta$, $N$ is the number of participants; section 3 in Supplementary Information).

In previous work, we defined a consistent estimator for RESI[17],

$$\hat{S} = \left( \max\left\{ 0, \frac{T^2 - m}{N} \right\} \right)^{1/2},$$

where $T^2$ is the chi-squared test statistics $T^2 = N(\beta - \beta_0)^T \Sigma_\beta^{-1} (\beta - \beta_0)$ for testing the null hypothesis $H_0: \beta = \beta_0$, $m$ is the number of parameters being tested (that is, the length of $\beta$) and $N$ is the number of participants.

As RESI is generally applicable across different models and data types, it is also applicable to the situation where Cohen's $d$ was defined. In this scenario, the RESI is equal to ½ Cohen's $d$[17], so Cohen's suggested thresholds for effect size can be adopted for RESI: small (RESI = 0.1), medium (RESI = 0.25) and large (RESI = 0.4). Because RESI is robust, when the assumptions of Cohen's $d$ are not satisfied, such as when the variances between the groups are not equal, RESI is still a consistent estimator, but Cohen's $d$ is not. The confidence intervals for RESI in our analyses were constructed using 1,000 non-parametric bootstraps[18].

The systematic difference in the standardized effect sizes between cross-sectional and longitudinal studies puts extra challenges on the comparison and aggregation of standardized effect size estimates across studies with different designs. To improve the comparability of standardized effect sizes between cross-sectional and longitudinal studies, we proposed a new effect size index: the cross-sectional RESI (CS-RESI) for longitudinal datasets. The CS-RESI for longitudinal datasets represents the RESI in the same study population if the longitudinal study had been conducted cross-sectionally. Detailed definition, point estimator and confidence interval construction procedure for CS-RESI can be found in section 3 in Supplementary Information. Comprehensive statistical simulation studies were also performed to demonstrate the valid performance of the proposed estimator and confidence interval for CS-RESI (section 3.2 in Supplementary Information). With

CS-RESI, we can quantify the benefit of using a longitudinal study design in a single dataset (section 3.3 in Supplementary Information).

**Study-level models.** After removing the site effects using ComBat or LongComBat in the multisite data, we estimated the effects of age and sex on each of the global or regional brain measures using GEEs and linear regression models in the longitudinal datasets and cross-sectional datasets, respectively. The mean model was specified as below after ComBat or LongComBat:

$$y_{ij} \sim \text{ns}(\text{age}_{ij}, \text{d.f.} = 2) + \text{sex}_i,$$

where $y_{ij}$ was taken to be a global brain measure (that is, total GMV, WMV, sGMV or mean cortical thickness) or regional brain measure (that is, regional GMV or cortical thickness) at the $j$-th visit from the participant $i$ and $j = 1$ for cross-sectional datasets. The age effect was estimated with natural cubic splines with 2 d.f., which means that there were two boundary knots and one interval knot placed at the median of the covariate age. For the GEEs, we used an exchangeable correlation structure as the working structure and identity linkage function. The model assumes the mean was correctly specified, but made no assumption about the error distribution. The GEEs were fitted with the 'geepack' package[61] in R. We used the RESI as a standardized effect size measure. RESIs and confidence intervals were computed using the 'RESI' R package (v1.2.0)[19].

**Meta-analysis of the age and sex effects.** The weighted linear regression model for the meta-analysis of age effects across the studies was specified as:

$$\hat{S}_{\text{age},k} = \text{design}_k + \text{ns}[\text{mean(age)}_k, 3] + \text{ns}[\text{s.d.(age)}_k, 3]$$
$$+ \text{ns}[\text{skew(age)}_k, 3] + \epsilon_k,$$

where $\hat{S}_{\text{age},k}$ denotes the estimated RESI for study $k$, and the weights were the inverse of the standard error of each RESI estimate. The sample mean, standard deviation (s.d.) and skewness of the age were included as non-linear terms estimated using natural splines with 3 d.f. (that is, two boundary knots plus two interval knots at the 33rd and 66th percentiles of the covariates), and a binary variable indicating the design type (cross-sectional or longitudinal) was also included.

The weighted linear regression model for the meta-analysis of sex effects across the studies was specified as

$$\hat{S}_{\text{sex},k} = \text{design}_k + \text{ns}[\text{mean(age)}_k, 3] + \text{ns}[\text{s.d.(age)}_k, 3]$$
$$+ \text{ns}[\text{pr(male)}_k, 3] + \epsilon_k,$$

where $\hat{S}_{\text{sex},k}$ denotes the estimated RESI of sex for study $k$, and the weights were the inverse of the standard error of each RESI estimate. The sample mean, standard deviation of the age covariate and the proportion of males in each study were included as non-linear terms estimated using natural splines with 3 d.f., and a binary variable indicating the design type (cross-sectional or longitudinal) was also included.

These meta-analyses were performed for each of the global and regional brain measures. Inferences were performed using robust standard errors[62]. In the partial regression plots, the expected standardized effect sizes for the age effects were estimated from the meta-analysis model after fixing mean age at 45 years, standard deviation of age at 7 years and/or skewness at 0; the expected standardized effect sizes for the sex effects were estimated from the meta-analysis model after fixing mean age at 45 years, standard deviation of age at 7 years and/ or proportion of males at 0.5.

**Sampling schemes for age in the UKB and ADNI.** We used bootstrapping to evaluate the effect of different sampling schemes with different target sample covariate distributions on the standardized effect sizes

and replicability in the cross-sectional UKB and longitudinal ADNI datasets. For a given sample size and sampling schemes, 1,000 bootstrap replicates were conducted. The standardized effect size was estimated as the mean standardized effect size (that is, RESI) across the bootstrap replicates. The 95% confidence interval for the standardized effect size was estimated using the lower and upper 2.5% quantiles across the 1,000 estimates of the standardized effect size in the bootstrap replicates. Power was calculated as the proportion of bootstrap replicates producing $P$ values less than or equal to 5% for those associations that were significant at 0.05 in the full sample. In the UKB, only one region was not significant for age in each of GMV and cortical thickness, and in the ADNI, only one and four regions were not significant for age in GMV and cortical thickness, respectively. Replicability in previous work has been defined as having a significant $P$ value and the same sign for the regression coefficient. Because we were fitting non-linear effects, we defined replicability as the probability that two independent studies have significant $P$ values; this is equivalent to the definition of power squared. The 95% confidence intervals for replicability were derived using Wilson's method[63].

In the UKB dataset, to modify the (between-subject) variability of the age variable, we used the following three target sampling distributions (Extended Data Fig. 1a): bell shaped, where the target distribution had most of the participants distributed in the middle age range; uniform, where the target distribution had participants equally distributed across the age range; and U shaped, where the target distribution had most of the participants distributed closer to the range limits of the age in the study. The samples with U-shaped age distribution had the largest sample variance of age, followed by the samples with uniform age distribution and the samples with bell-shaped age distribution. The bell-shaped and U-shaped functions were proportional to a quadratic function. To sample according to these distributions, each record was first inversely weighted by the frequency of the records with age falling in the range of ±0.5 years of the age for that record to achieve the uniform sampling distribution. Each record was then rescaled to derive the weights for bell-shaped and U-shaped sampling distributions. The records with age < 50 or age > 78 years were winsorized at 50 or 78 years when assigning weights, respectively, to limit the effects of outliers on the weight assignment, but the actual age values were used when analysing each bootstrapped data.

In each bootstrap from the ADNI dataset, each participant was sampled to have two records. We modified the between-subject and within-subject variability of age, respectively, by making the 'baseline age' follow one of the three target sampling distributions used for the UKB dataset and the 'age change' independently follow one of three new distributions: decreasing, uniform and increasing (Extended Data Fig. 1b,c). The increasing and decreasing functions were proportional to an exponential function. The samples with increasing distribution of age change had the largest within-subject variability of age, followed by the samples with the uniform distribution of age change and the samples with decreasing distribution of age change.

To modify the baseline age and the age change from baseline independently, we first created all combinations of the baseline record and one follow-up from each participant, and derived the baseline age and age change for each combination. The 'bivariate' frequency of each combination was obtained as the number of combinations with values of baseline age and age change falling in the range of ±0.5 years of the values of baseline age and age change for this combination. Then, each combination was inversely weighted by its bivariate frequency to target a uniform bivariate distribution of baseline age and age change. The weight for each combination was then rescaled to make the baseline age and age change follow different sampling distributions independently. The combinations with baseline age < 65 or age > 85 years were winsorized at 65 or 85 years, and the combinations with age change greater than 5 years were winsorized at 5 years when assigning weights to limit the effects of outliers on the

weight assignment, but the actual ages were used when analysing each bootstrapped data.

The sampling methods could be easily extended to the scenario in which each participant had three records (and more than three) in the bootstrap data by making combinations of the baseline and two follow-ups. Each combination was inversely weighted to achieve uniform baseline age and age change distributions, respectively, by the 'trivariate' frequency of the combinations with baseline age and the two age changes from baseline for the two follow-ups falling into the range of ±0.5 years of the corresponding values for this combination. As we only investigated the effect of modifying the number of measurements per participant under uniform between-subject and within-subject sampling schemes (Fig. 3c,d), we did not need to consider rescaling the weights here to achieve other sampling distributions but they could be done similarly. For the scenario in which each participant only had one measurement (Fig. 3c,d), the standardized effect sizes and replicability were estimated only using the baseline measurements.

All site effects were removed using ComBat or LongComBat before performing the bootstrap analysis.

**Sampling schemes for other non-brain covariates in the ABCD.** We used bootstrapping to study how different sampling strategies affect the RESI in the ABCD dataset. Each participant in the bootstrap data had two measurements. We applied the same weight assignment method described above for the ADNI dataset to modify the between-subject and within-subject variability of a covariate. We made the baseline covariate and the change in covariate follow bell-shaped and/or U-shaped distributions, to let the sample have larger or smaller between-subject and/or within-subject variability of the covariate, respectively. The baseline covariate and change of covariate were winsorized at the upper and lower 5% quantiles to limit the effect of outliers on sampling. For each cognitive variable, only the participants with non-missing baseline measurements and at least one non-missing follow-up were included.

Generalized linear models and GEEs were fitted to estimate the effect of each non-brain covariate on the structural brain measures after controlling for age and sex,

$$\text{GMV} \sim \text{ns(age, d.f.} = 2) + \text{sex} + x,$$

where $x$ denotes one of the non-brain covariates. For the GEEs, we used an exchangeable correlation structure as the working structure and identity linkage function.

Only the between-subject sampling schemes were applied for the non-brain covariates that were stable over time (for example, birth weight and handedness). In other words, the participants were sampled based on their baseline covariate values, and then a follow-up was selected randomly for each participant. The sampling schemes to increase the between-subject variability in the covariate handedness, which was a binary variable (right-handed or not), was specified differently. The expected proportion of right-handed participants in the bootstrap samples was 50% under the sampling scheme with larger between-subject variability and 10% under the sampling scheme with smaller between-subject variability.

For given between-subject and/or within-subject sampling schemes, we obtained 1,000 bootstrap replicates. The standardized effect size was estimated as the mean standardized effect size across the bootstrap replicates. The 95% confidence intervals for standardized effect size were estimated using the lower and upper 2.5% quantiles across the 1,000 estimates of the standardized effect size in the bootstrap replicates. The sample sizes needed for 80% replicability were estimated based on the (mean) estimated standardized effect size and $F$-distribution (see below).

**Analysis of functional connectivity in the ABCD.** In a subset of the ABCD in which we have preprocessed longitudinal functional

connectivity data at two time points (baseline and 2-year follow-up), we only restricted our analysis to the participants with non-missing measurements at both of the two time points. In the GEEs used to estimate the effects of non-brain covariates on functional connectivity, the mean model was specified as below after LongComBat:

$$y_{ij} \sim ns(age_{ij}, d.f. = 2) + sex_i + ns(mean\_FD, d.f. = 3) + x,$$

where $y_{ij}$ was taken to be a functional connectivity outcome, and $x$ denotes a non-brain covariate. The mean frame-wise displacement (mean_FD) was also included as a covariate with natural cubic splines with 3 d.f. We used an exchangeable correlation structure as the working structure and identity linkage function in the GEEs. The frame count of each scan was used as the weights.

When evaluating the effect of different sampling schemes on the standardized effect sizes, we obtained 1,000 bootstrap replicates for given between-subject and/or within-subject sampling schemes. The standardized effect size was estimated as the mean standardized effect size across the bootstrap replicates. Confidence intervals were computed as described above. The sample sizes needed for 80% replicability were estimated based on the (mean) estimated standardized effect sizes and $F$-distribution (see below).

**Sample size calculation for a target power or replicability with a given standardized effect size.** After estimating the standardized effect size for an association, the calculation of the corresponding sample size $N$ needed for detecting this association with $\gamma \times 100\%$ power at significance level of $\alpha$ was based on an $F$-distribution. d.f. denotes the total degree of freedom of the analysis model, $F(z;\lambda)$ denotes the cumulative density function for a random variable $z$, which follows the (non-central) $F$-distribution with degrees of freedom being 1 and $N -$ d.f. and non-centrality parameter $\lambda$. The corresponding sample size $N$ is:

$$N = \{N : F(F^{-1}(1-\alpha); \lambda = N\hat{S}^2) = \gamma\},$$

where $\hat{S}$ is the estimated RESI for the standardized effect size. Power curves for the RESI are given in figure 3 of Vandekar et al.[17]. Replicability was defined as the probability that two independent studies have significant $P$ values, which is equivalent to power squared.

**Estimation of the between-subject and within-subject effects.** For the non-brain covariates that were analysed longitudinally in the ABCD dataset, GEEs with exchangeable correlation structures were fitted to estimate their cross-sectional and longitudinal effects on structural and functional brain measures after controlling for age and sex, respectively. The mean model was specified as illustrated with GMV:

$$GMV \sim ns(age, d.f. = 2) + sex + X\_bl + X\_change,$$

where $X\_bl$ denotes the participant-specific baseline covariate values, and the $X\_change$ denotes the difference of the covariate value at each visit to the participant-specific baseline covariate value (see section 5.2 in Supplementary Information). The participants without baseline measures were not included in the modelling. The model coefficients for the terms $X\_bl$ and $X\_change$ represent the between-subject and within-subject effects of this non-brain covariate on total GMV, respectively. For the functional connectivity data, the same covariates and weighting were used as described above. Using the first time point as the between-subject term was a special case that ensured that comparing the parameter using the baseline cross-sectional model was equal to the parameter for the between-subject effect in the longitudinal model. In this model, the between-subject variance was defined as the variance of the baseline measurement, and the within-subject variance was the mean square of $X\_change$. This model specification ensured that the

sampling schemes independently affected the between-subject and within-subject variances separately (equation (16) in section 5.2 in Supplementary Information).

### Reporting summary
Further information on research design is available in the Nature Portfolio Reporting Summary linked to this article.

## Data availability
Participant-level data from many datasets are available according to study-level data access rules. Study-level model parameters are available at https://github.com/KaidiK/RESI_BWAS. We acknowledge the usage of several openly shared MRI datasets, which are available at the respective consortia websites and are subject to the sharing policies of each consortium: OpenNeuro (https://openneuro.org/), UKB (https://www.ukbiobank.ac.uk/), ABCD (https://abcdstudy.org/), the Laboratory of NeuroImaging (https://loni.usc.edu/), data made available through the Open Science Framework (https://osf.io/), the Human Connectome Project (http://www.humanconnectomeproject.org/) and the OpenPain project (https://www.openpain.org). The ABCD data repository grows and changes over time. The ABCD data used in this paper are from the NIMH Data Archive (https://doi.org/10.15154/1503209) and the ABCD BIDS Community Collection (ABCC; https://collection3165.readthedocs.io). Data used in this article were provided by the Brain Consortium for Reliability, Reproducibility and Replicability (3R-BRAIN) (https://github.com/zuoxinian/3R-BRAIN). Data used in the preparation of this article were obtained from the Australian Imaging Biomarkers and Lifestyle (AIBL) flagship study of ageing funded by the Commonwealth Scientific and Industrial Research Organisation (CSIRO), which was made available at the ADNI database (https://adni.loni.usc.edu/aibl-australian-imaging-biomarkers-and-lifestyle-study-of-ageing-18-month-data-now-released/). The AIBL researchers contributed data but did not participate in analysis or writing of this report. AIBL researchers are listed at https://www.aibl.csiro.au. Data used in preparation of this article were obtained from the ADNI database (https://adni.loni.usc.edu/). The investigators within the ADNI contributed to the design and implementation of the ADNI and/or provided data but did not participate in analysis or writing of this report. A complete listing of ADNI investigators can be found at https://adni.loni.usc.edu/wp-content/uploads/how_to_apply/ADNI_Acknowledgement_List.pdf. More information on the ARWIBO Consortium can be found at https://www.arwibo.it/. More information on CALM team members can be found at https://calm.mrc-cbu.cam.ac.uk/team/ and in the Supplementary Information. Data used in this article were obtained from the developmental component 'Growing Up in China' of the Chinese Color Nest Project (http://deepneuro.bnu.edu.cn/?p=163). Data used in the preparation of this article were obtained from the IConsortium on Vulnerability to Externalizing Disorders and Addictions (c-VEDA), India (https://cveda-project.org/). Data used in the preparation of this article were obtained from the Harvard Aging Brain Study (HABS P01AG036694; https://habs.mgh.harvard.edu). Data used in the preparation of this article were obtained from the IMAGEN Consortium (https://imagen-europe.com/). The POND Network (https://pond-network.ca/) is a Canadian translational network in neurodevelopmental disorders, primarily funded by the Ontario Brain Institute. The LBCC dataset used in the preparation of this article includes data obtained from the ADNI database (https://adni.loni.usc.edu). The ADNI was launched in 2003 as a public–private partnership, led by Principal Investigator M. W. Weiner. The primary goal of the ADNI has been to test whether serial MRI, positron emission tomography, other biological markers, and clinical and neuropsychological assessment can be combined to measure the progression of mild cognitive impairment and early Alzheimer's disease. Its data collection and sharing for

this project were funded by the ADNI (National Institutes of Health grant U01 AG024904) and Department of Defense ADNI (Department of Defense award number W81XWH-12-2-0012). ADNI is funded by the National Institute on Aging, the National Institute of Biomedical Imaging and Bioengineering, and through contributions from the following: AbbVie, Alzheimer's Association; Alzheimer's Drug Discovery Foundation; Araclon Biotech; BioClinica; Biogen; Bristol-Myers Squibb; CereSpir; Cogstate; Eisai; Elan Pharmaceuticals; Eli Lilly and Company; EuroImmun; F. Hoffmann-La Roche and its affiliated company Genentech; Fujirebio; GE Healthcare; IXICO; Janssen Alzheimer Immunotherapy Research & Development; Johnson & Johnson Pharmaceutical Research & Development; Lumosity; Lundbeck; Merck & Co.; Meso Scale Diagnostics; NeuroRx Research; Neurotrack Technologies; Novartis Pharmaceuticals Corporation; Pfizer; Piramal Imaging; Servier; Takeda Pharmaceutical Company; and Transition Therapeutics. The Canadian Institutes of Health Research are providing funds to support ADNI clinical sites in Canada. Private sector contributions are facilitated by the Foundation for the National Institutes of Health (www.fnih.org). The grantee organization is the Northern California Institute for Research and Education, and the study is coordinated by the Alzheimer's Therapeutic Research Institute at the University of Southern California. ADNI data are disseminated by the Laboratory for NeuroImaging at the University of Southern California.

## Code availability

All code used to produce the analyses presented in this study is available at https://github.com/KaidiK/RESI_BWAS.

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

**Acknowledgements** S.V. was supported by R01MH123563 from the National Institute of Mental Health (NIMH). A.A.-B. and J.Seidlitz were partially supported by R01MH132934 and R01MH133843 from the NIMH. B.T.-C. was supported by K23DA057486 from National Institute of Drug Abuse (NIDA). T.D.S. was supported by R01MH120482, R01MH112847, R01MH113550 and R37MH125829 from the NIMH, R01EB022573 from the National Institute of Biomedical Imaging and Bioengineering (NIBIB), the AE Foundation and the Penn-CHOP Lifespan Brain Institute. B.L. was supported by R00MH127293 from the NIMH. D.F. was supported by U01DA041148R and U24DA055330 from the NIDA, R37MH125829, R01MH096773 and R01MH115357 from the NIMH, the Masonic Institute for the Developing Brain, and the Lynne and Andrew Redleaf Foundation. Data processing done at the University of Cambridge is supported by the NIHR Cambridge Biomedical Research Centre (BRC-1215-20014) and NIHR Applied Research Collaboration East of England. Data used in the preparation of this article include data obtained from the ADNI database (https://adni.loni.usc.edu). As such, the investigators in the ADNI contributed to the design and implementation of the ADNI and/or provided data, but did not participate in the analysis or writing of this report. A complete list of the ADNI investigators is available (https://adni.loni.usc.edu/wp-content/uploads/how_to_apply/ADNI_Acknowledgement_List.pdf). Data were used from the following consortia: 3R-BRAIN, AIBL, Alzheimer's Disease Neuroimaging Initiative (ADNI), Alzheimer's Disease Repository Without Borders Investigators, CALM Team, CCNP, COBRE, cVEDA, Harvard Aging Brain Study, IMAGEN, POND, and The PREVENT-AD Research Group; and lists of members and their affiliations appears in the Supplementary Information. Any views expressed are those of the authors and not necessarily those of the funders, IHU-JU2, the NIHR or the Department of Health and Social Care.

**Author contributions** K.K., J. Seidlitz, S.V. and A.A.-B. conceived the work. K.K., S.V., R.T. and J. Schildcrout performed the methodology. K.K. and S.V. conducted the analysis. K.K., S.V., J. Seidlitz, R.A.I.B, K.M., A.S.K., R.T., A.R., B.L., B.T.-C., E.F., O.M.D., S.M.N., J. Schildcrout, D.F., T.D.S. and A.A.-B. interpreted the results. A.A.-B. J. Seidlitz, R.A.I.B., K.M., A.S.K., A.R., B.L., B.T.-C., E.F., O.M.D., S.M.N., D.F., T.D.S. and consortia authors performed data acquisition and curation. J.X. and M.T.J. performed the validation. K.K., J. Seidlitz, A.A.-B. and S.V. drafted the manuscript. All authors revised the manuscript.

**Competing interests** J.Seidlitz and R.A.I.B. are directors and hold equity in Centile Bioscience. A.A.-B. holds equity in Centile Bioscience and received consulting income from Octave Bioscience in 2023. S.M.N. consults for Turing Medical, which commercializes FIRMM. This interest has been reviewed and managed by the University of Minnesota in accordance with its conflict of interest policies. All other authors declare no competing interests.

**Additional information**
**Correspondence and requests for materials** should be addressed to Kaidi Kang or Simon Vandekar.

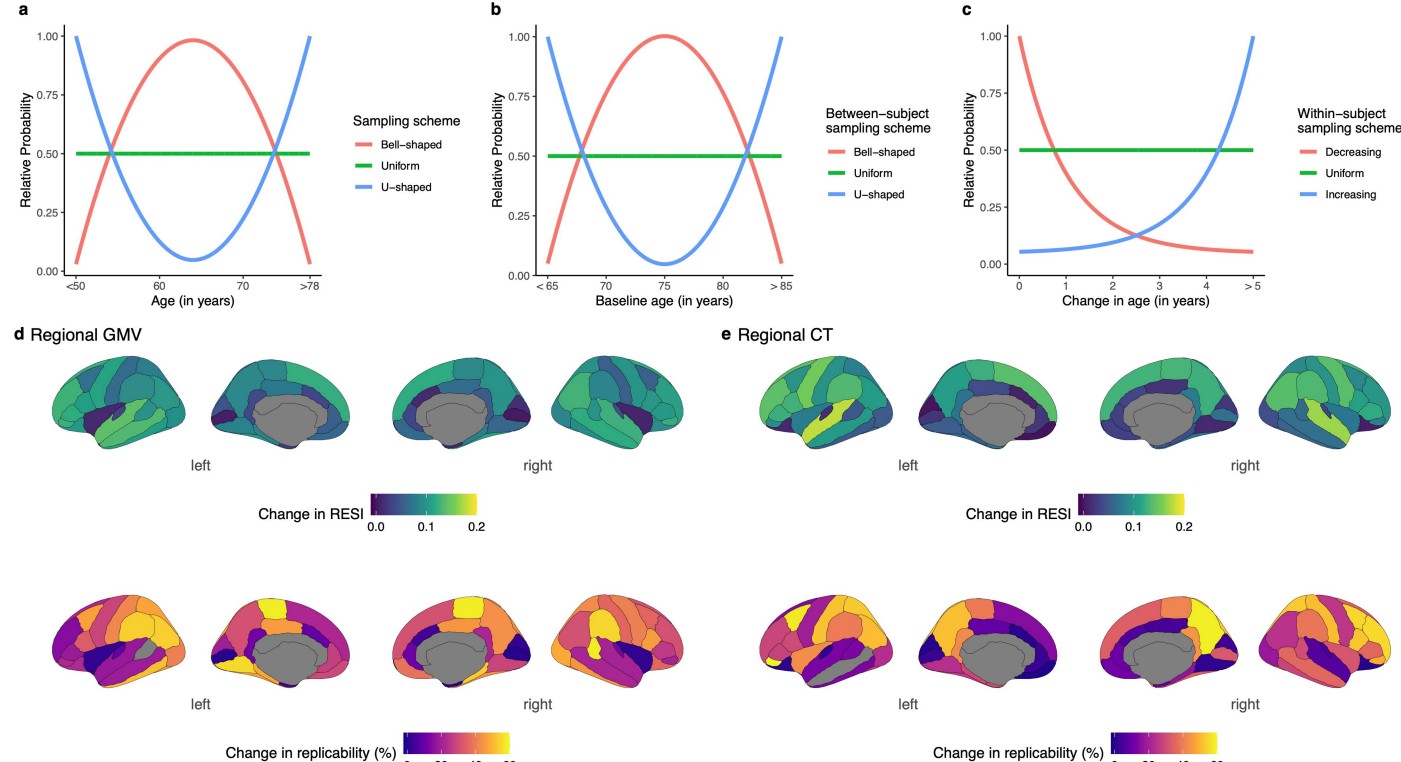

**Extended Data Fig. 1 | The illustration of implemented sampling schemes and the region-specific improvement in the standardized effect sizes and replicability for the age associations in UKB.** (**a**) The sampling scheme implemented in UKB. The sampling schemes adjust the variability of age in the samples by assigning heavier or lighter weights to the participants with age at the two tails of the population. The U-shaped scheme produces the largest variability of age in the samples, followed by uniform and bell-shaped sampling schemes. (**b-c**) Between- and within-subject sampling schemes implemented in ADNI. (**b**) The between-subject variability of age is adjusted by assigning

heavier or lighter weights to the participants with baseline age closer to the two tails of the population baseline age distribution. (**c**) The within-subject variability in age is adjusted by increasing or decreasing the probability of selecting the follow-up observation(s) with a larger change in age since baseline. (**d-e**) Region-specific improvement in the RESI and replicability in UKB for the association between age and (**d**) regional gray matter volume (GMV) and (**e**) regional cortical thickness (CT), respectively, by using U-shaped sampling scheme compared with bell-shaped sampling scheme, when $N = 300$.

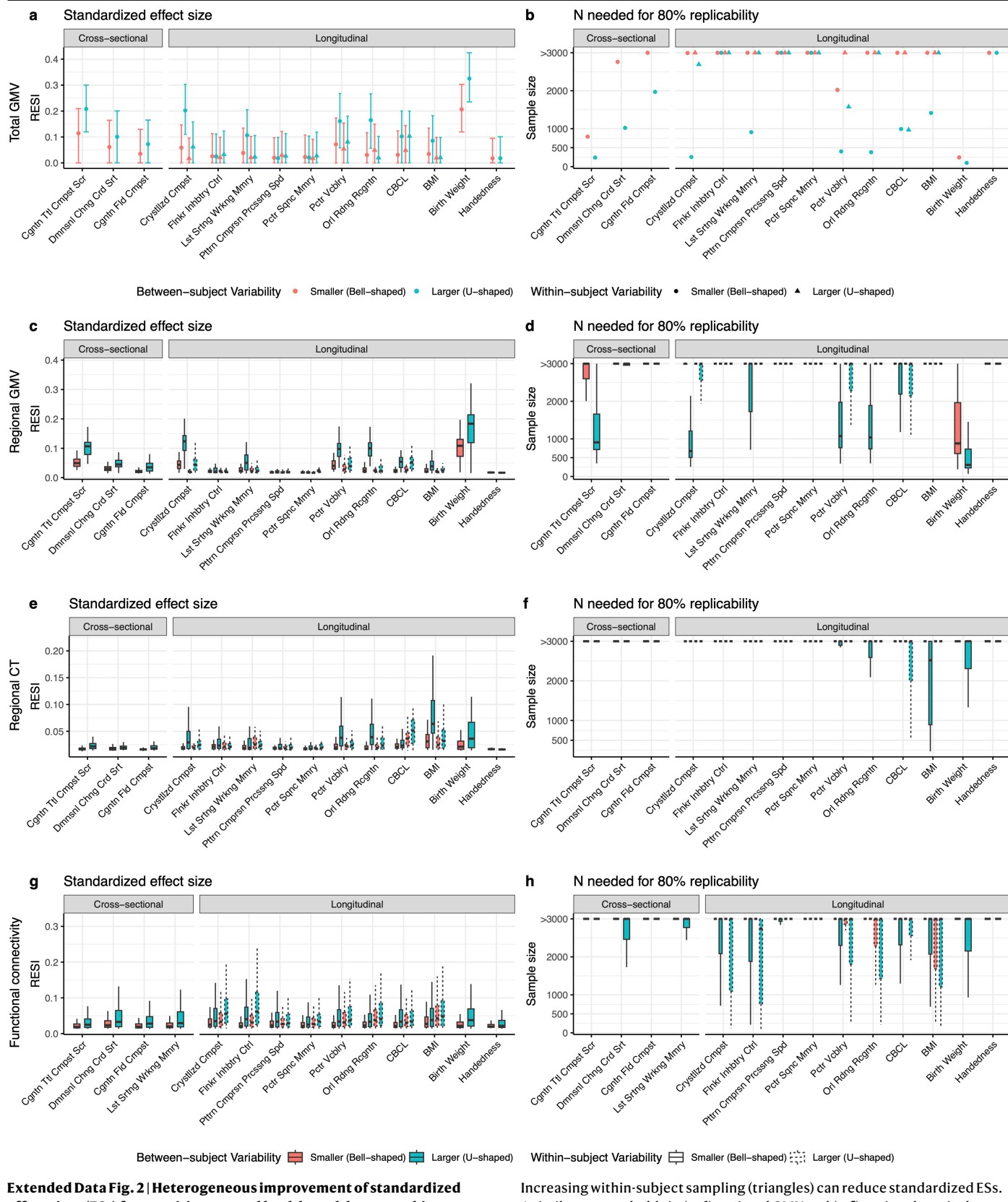

**Extended Data Fig. 2 | Heterogeneous improvement of standardized effect sizes (ESs) for cognitive, mental health, and demographic associations with structural and functional brain measures in the ABCD study with bootstrapped samples of *N* = 500.** (**a**) U-shaped between-subject sampling scheme (blue) that increases between-subject variability of the non-brain covariate produces larger standardized ESs and (**b**) reduces the number of participants scanned to obtain 80% replicability in total gray matter volume (GMV). The points and triangles are the average standardized ESs across bootstraps and the whiskers are the 95% confidence intervals.

Increasing within-subject sampling (triangles) can reduce standardized ESs. A similar pattern holds in (**c-d**) regional GMV and (**e-f**) regional cortical thickness (CT); boxplots show the distributions of the standardized ESs across regions. In contrast, (**g**) regional pairwise functional connectivity (FC) standardized ESs are improved by increasing between- (blue) and within-subject variability (dashed borders) with a corresponding reduction in the (**h**) number of participants scanned for 80% replicability. **c-h**, Boxplots show the median (horizontal line), interquartile range (grey box), and min-max values (vertical lines).

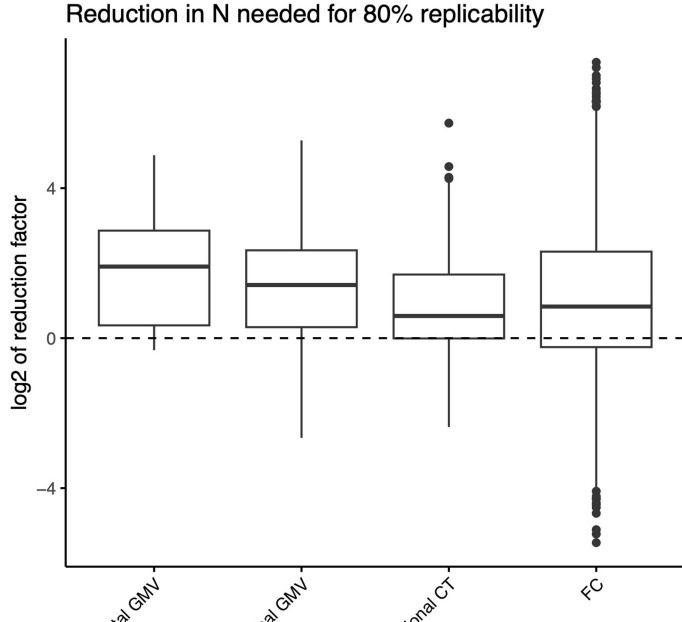

**Extended Data Fig. 3 | Boxplots showing the distributions of (log2 of) reduction factors of the sample size *N* needed for 80% replicability by increasing between-subject variability of the covariates across all the associations with each of the outcomes in ABCD (Fig. 4 and Extended Data Fig. 2).** The reduction factors are derived by comparing the sample sizes needed for 80% replicability with U-shaped to the one with bell-shaped between-subject sampling scheme when the within-subject sampling scheme is bell-shaped (Extended Data Fig. 1b). GMV, gray matter volume; CT, cortical thickness; FC, functional connectivity. Boxplots show the median (horizontal line), interquartile range (grey box), min-max values (vertical lines), and outliers (points).

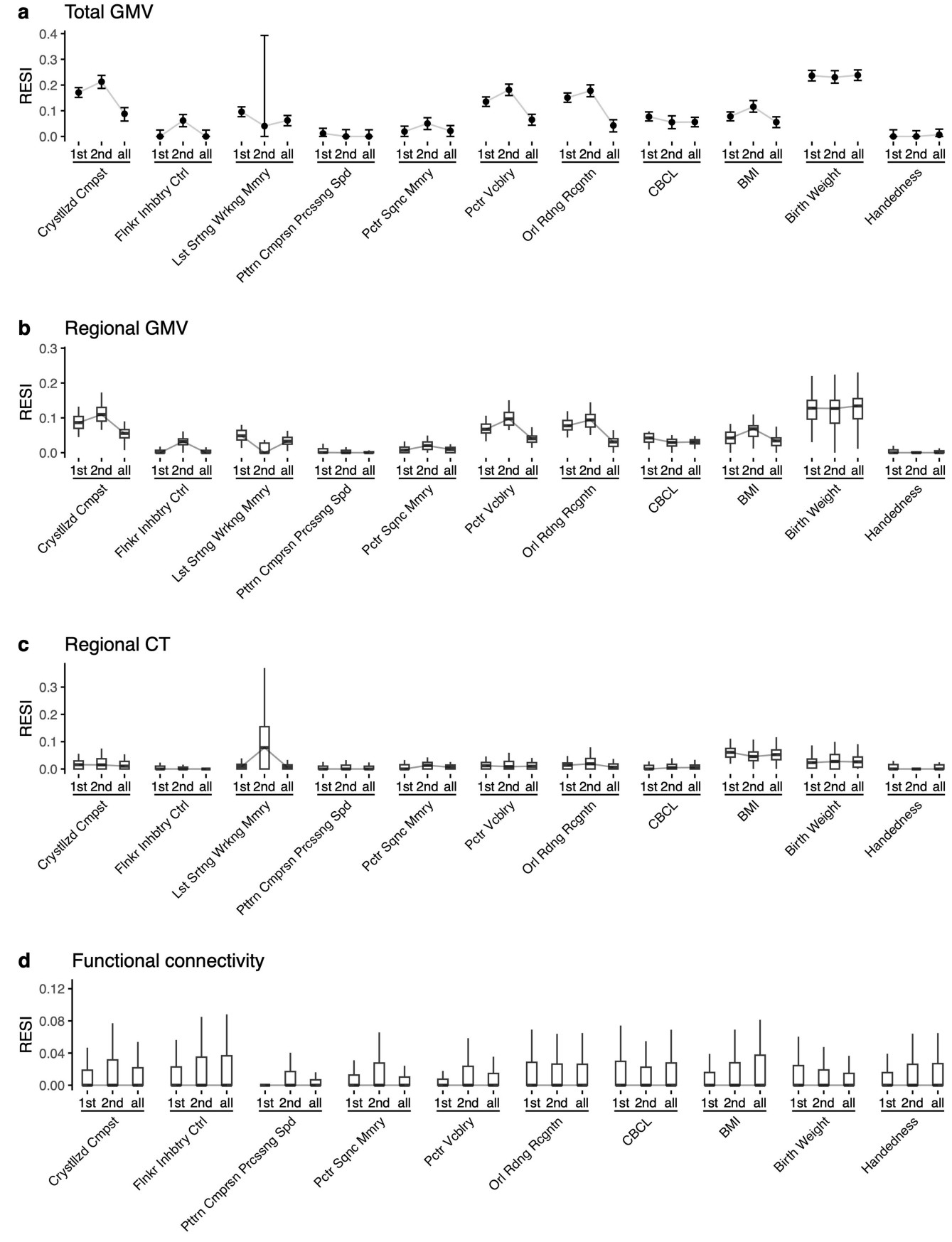

**Extended Data Fig. 4 |** See next page for caption.

**Extended Data Fig. 4 | Longitudinal study designs can reduce standardized effect sizes (ESs) and replicability.** Boxplots show the distributions of the standardized ESs across regions. The cross-sectional analyses use only the baseline or the 2nd measures (indicated by "1st"s or "2nd"s on the x-axes, respectively). The longitudinal analyses use the full longitudinal data (indicated by "all"s on the x-axes). (**a-c**) Cross-sectional analyses can have larger standardized ESs than the same longitudinal analyses for structural brain measures in ABCD. (**d**) The functional connectivity (FC) measures have a slight benefit of longitudinal modeling. GMV, grey matter volume; CT, cortical thickness. **b-d**, Boxplots show the median (horizontal line), interquartile range (grey box), and min-max values (vertical lines).

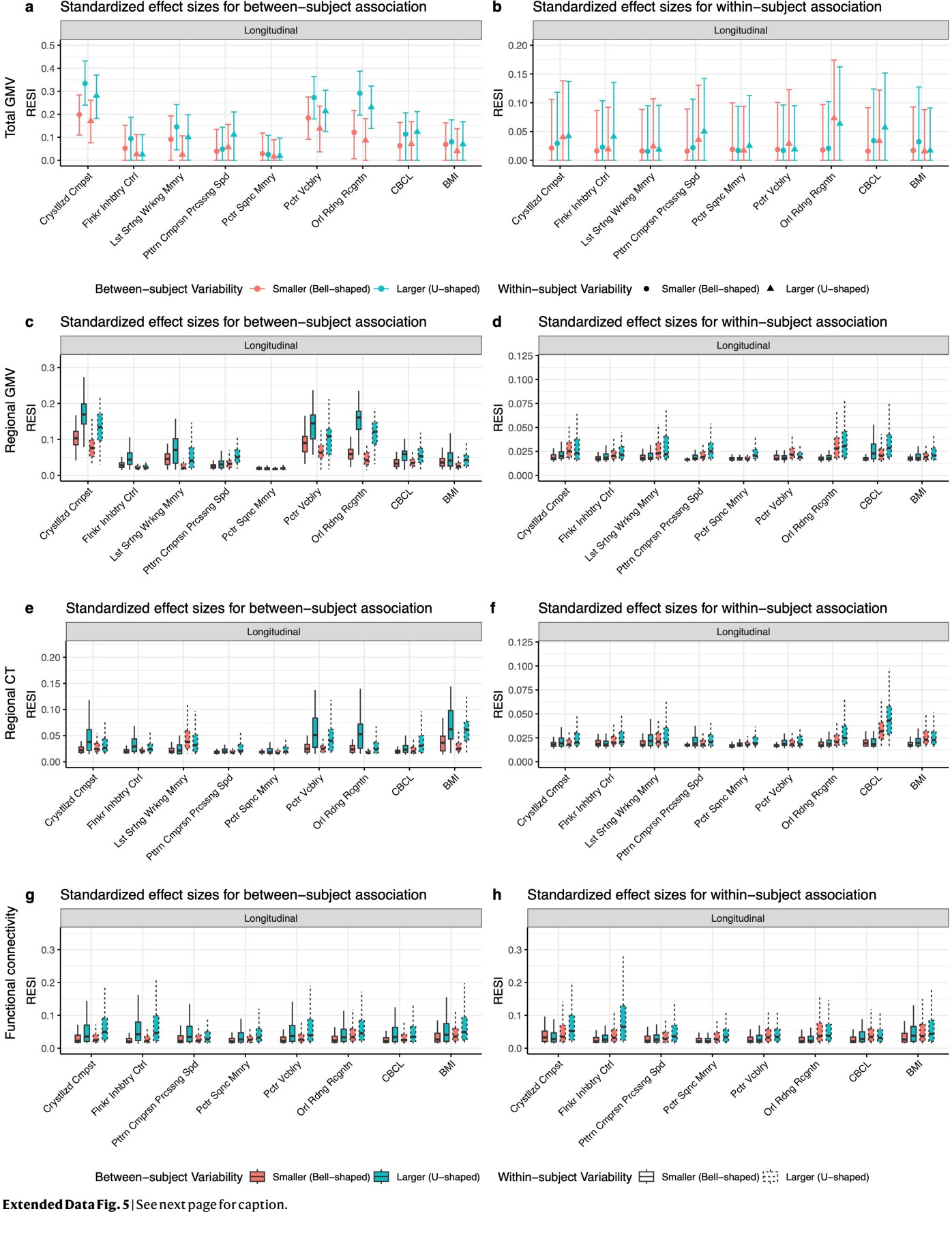

**Extended Data Fig. 5** | See next page for caption.

**Extended Data Fig. 5 | The influence of sampling schemes on the standardized effect sizes (ESs) for between- and within-subject associations, respectively, of cognition, mental health, and demographic covariates with different brain measures in the ABCD study at *N* = 500.** Boxplots show the distribution of the standardized ESs across regions. Between-subject standardized ESs are predominantly affected by the between-subject variance, whereas within-subject standardized ESs are predominantly affected by the within-subject variance. Consistent results were found for structural brain measures total grey matter volume (GMV; a, b), regional GMV (c-d), regional cortical thickness (CT; e,f) and functional brain measures (g,h). The results for covariates birthweight and handedness, which do not vary within participants, are not included as the within-subject sampling schemes do not apply to them. **c-h**, Boxplots show the median (horizontal line), interquartile range (box), and min-max values (vertical lines).

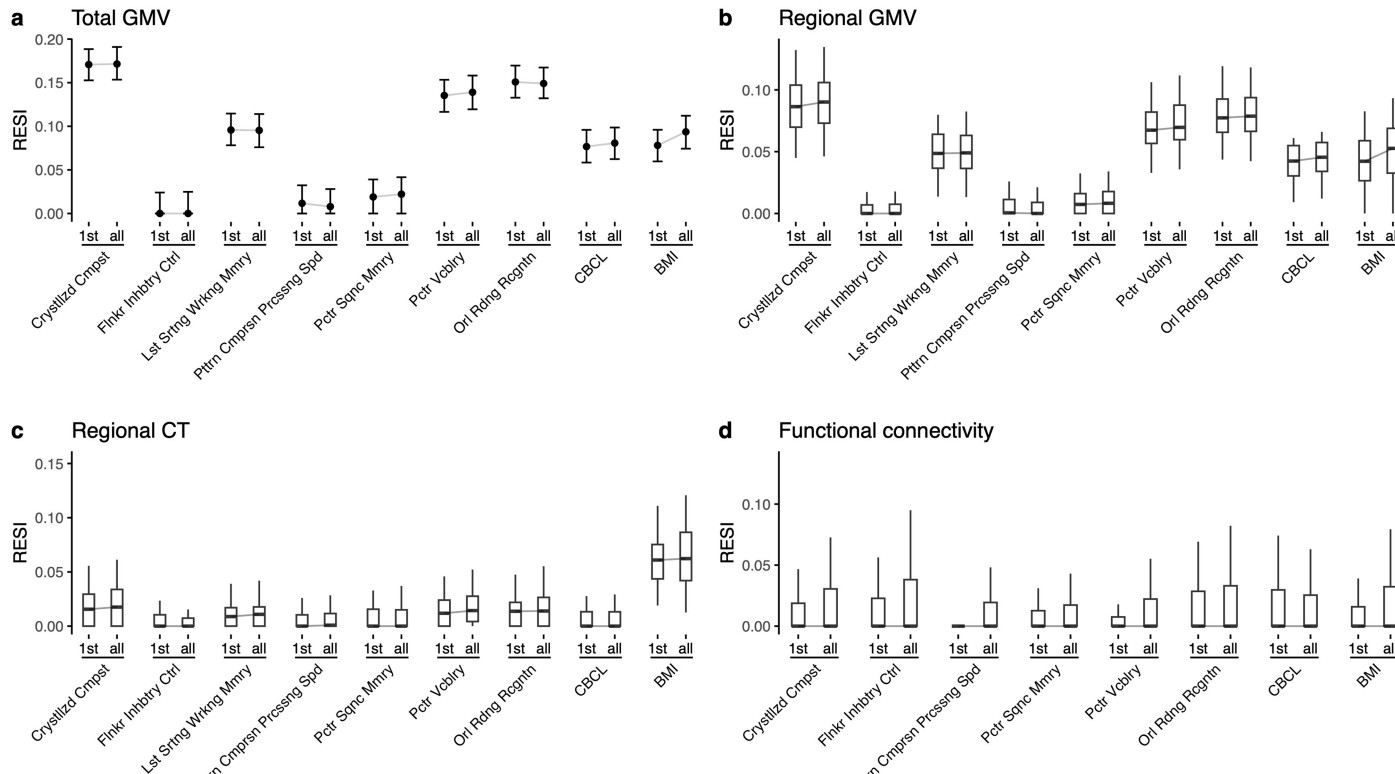

**Extended Data Fig. 6 | The estimated standardized effect sizes (ESs) from cross-sectional and longitudinal analyses, respectively, for the between-subject associations for cognition, mental health, and demographic covariates with different brain measures in the ABCD study.** The estimated RESIs for cross-sectional analyses (that only use the baseline measures) are indicated by "1st"s on the x-axes; the estimated RESIs for the between-subject effects from longitudinal analyses (that use the full longitudinal data and a specification of separate between- and within-subject effects (see Methods: Estimation of the between-subject and within-subject effects) are indicated by "all"s on the x-axes. By separating the between- and within-subject effects in the longitudinal model, we can avoid averaging the different between- and within-subject effects and maintain the benefit of longitudinal designs on the estimated RESIs for the between-subject effects on both structural brain measures (a-c) and functional brain measures (d). The results for covariates birthweight and handedness are not included, as they do not vary within-subjects so only their between-subject effects can be estimated (which are shown in Extended Data Fig. 4). **b-d**, Boxplots show the median (horizontal line), interquartile range (gray box), and min-max values (vertical lines).

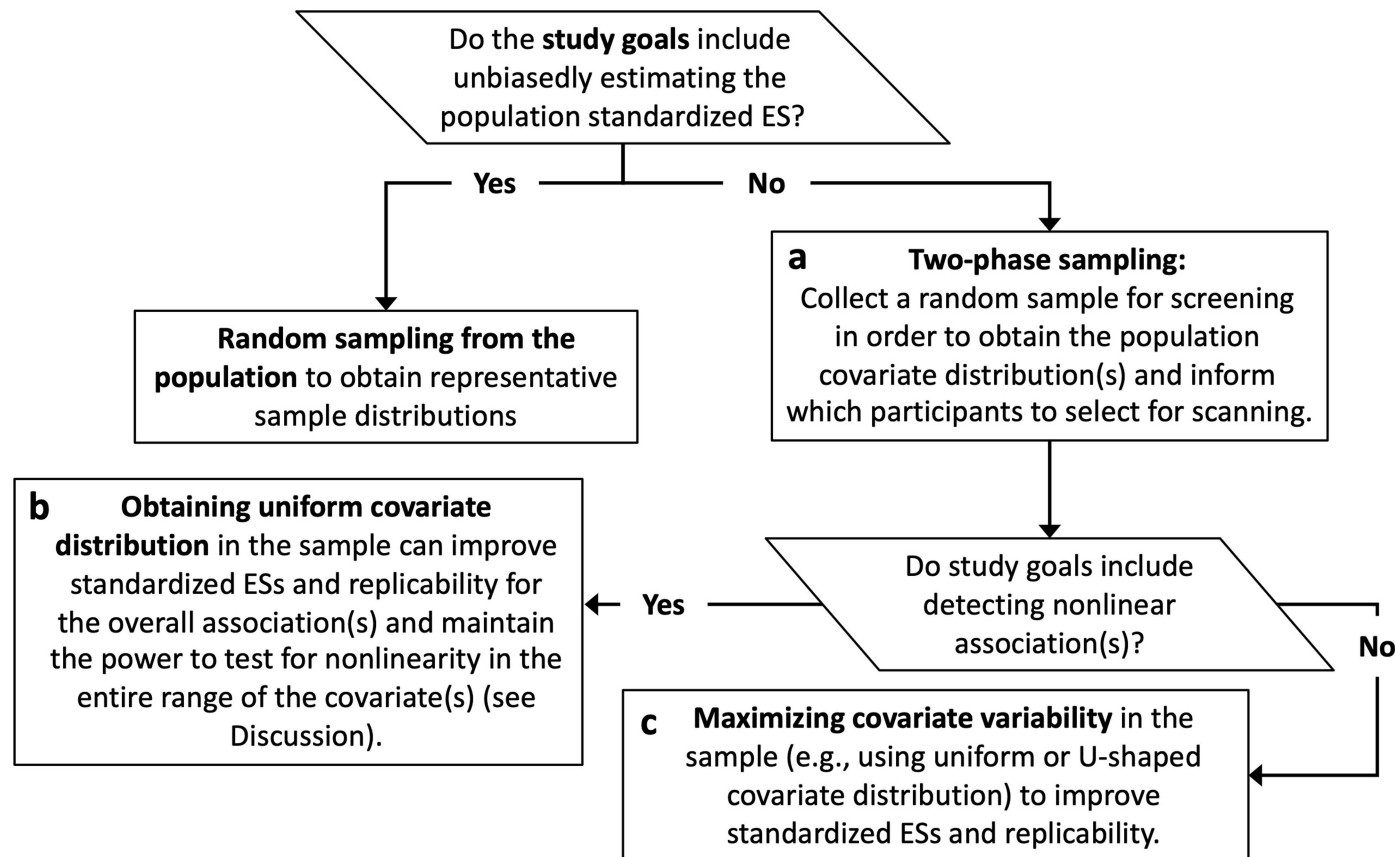

**Extended Data Fig. 7 | Decision tree for modified sampling strategy for a single primary covariate.** Random/representative sampling is needed to unbiasedly estimate the variance of the covariate distribution in the population in order to obtain standardized effect size (ES) estimates consistent with the population. (**a**) A two-phase design is needed to modify the covariate distribution(s) in the sample to increase standardized ESs and replicability, where random sampling is performed first in a larger dataset to collect covariate values and sampling based on collected covariates values is used to optimize the standardized ESs and replicability; unbiased population standardized ES estimates still can be obtained using weighted estimation (see Discussion: Optimal design considerations). (**b**) If the distribution(s) of the covariate(s) in the population is bell-shaped, a uniform covariate distribution in the sample can still increase the standardized ES and replicability in detecting the overall association. (**c**) The particular target distribution will depend on the difficulty of collecting participants in the tail of the distributions (see section 4.1 in Supplementary Information).

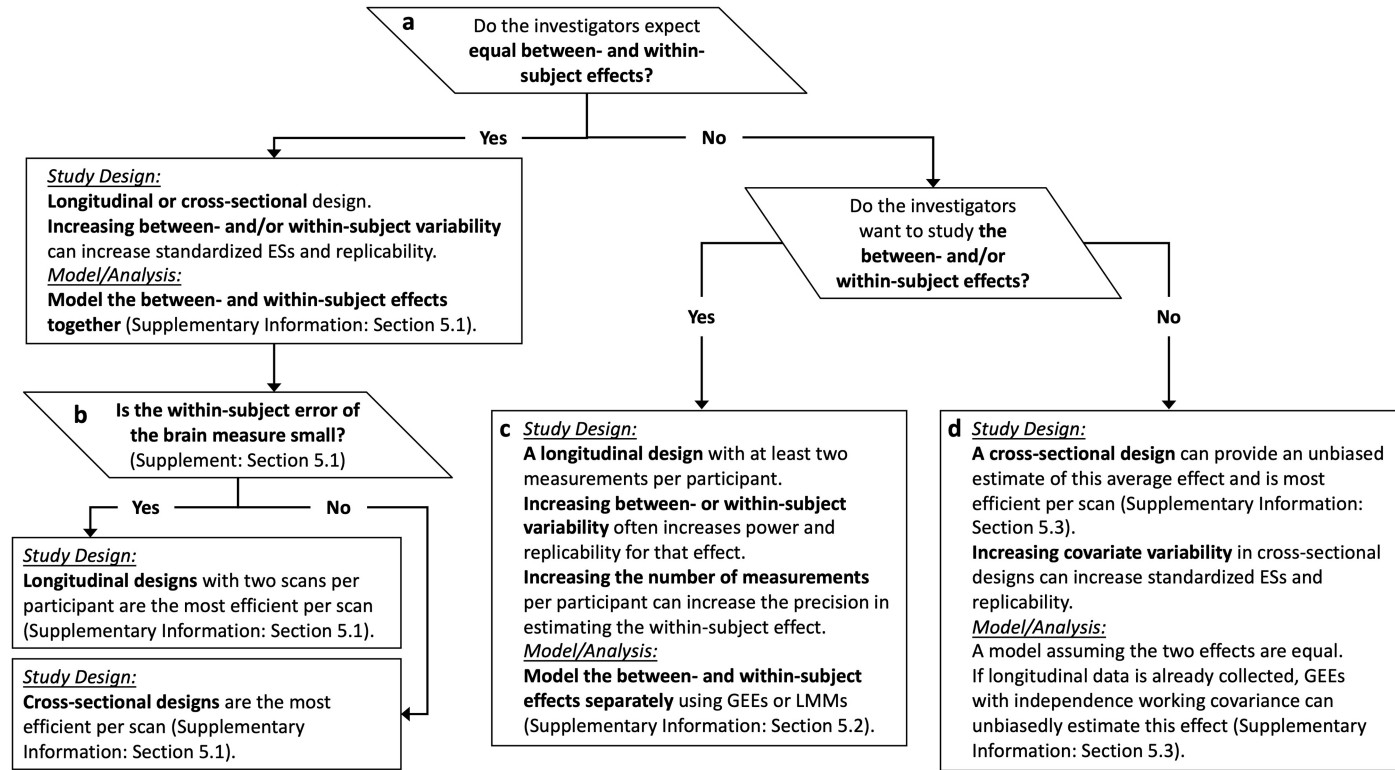

**Extended Data Fig. 8 | Optimal study design and analysis depends on characteristics of the hypothesized association(s).** (**a**) Visualization can be performed in pilot or study data to evaluate this assumption as in Supplementary Fig. 4 (section 5 in Supplementary Information). (**b**) If the between- and within-subject effects are hypothesized to be equal, either a cross-sectional or longitudinal design can be applied, but the efficiency per scan depends on the size of the within-subject error of the brain measure; pilot/ study data can be used to evaluate this question (section 5.1 in Supplementary Information). (**c**) If estimating the between- and within-subject effects separately, a longitudinal design is required and common longitudinal data analysis tools such as generalized estimating equations (GEEs) and linear mixed models (LMMs) with separate between- and within-subject effects are required

to unbiasedly estimate these effects (see section 5.2 in Supplementary Information). (**d**) If there are different between- and within-subject effects, the investigators may still use a model to target the average effect (i.e., a weighted average of the underlying between- and within-subject effects) if they have cross-sectional data, or if they want results from a longitudinal study that are consistent for the same biological effect as cross-sectional studies. For longitudinal studies, a GEE with independence working covariance structure targets the same average effect as the cross-sectional model, but it is less statistically efficient than the cross-sectional model (see section 5.3 in Supplementary Information). All recommendations are based on the empirical findings in the paper and the theory for exchangeable covariance longitudinal linear models in the Supplementary Information.

# Reporting Summary

## Statistics

For all statistical analyses, confirm that the following items are present in the figure legend, table legend, main text, or Methods section.

| n/a | Confirmed | |
|---|---|---|
| ☐ | ☒ | The exact sample size ($n$) for each experimental group/condition, given as a discrete number and unit of measurement |
| ☐ | ☒ | A statement on whether measurements were taken from distinct samples or whether the same sample was measured repeatedly |
| ☐ | ☒ | The statistical test(s) used AND whether they are one- or two-sided *Only common tests should be described solely by name; describe more complex techniques in the Methods section.* |
| ☐ | ☒ | A description of all covariates tested |
| ☐ | ☒ | A description of any assumptions or corrections, such as tests of normality and adjustment for multiple comparisons |
| ☐ | ☒ | A full description of the statistical parameters including central tendency (e.g. means) or other basic estimates (e.g. regression coefficient) AND variation (e.g. standard deviation) or associated estimates of uncertainty (e.g. confidence intervals) |
| ☐ | ☒ | For null hypothesis testing, the test statistic (e.g. $F$, $t$, $r$) with confidence intervals, effect sizes, degrees of freedom and $P$ value noted *Give P values as exact values whenever suitable.* |
| ☒ | ☐ | For Bayesian analysis, information on the choice of priors and Markov chain Monte Carlo settings |
| ☐ | ☒ | For hierarchical and complex designs, identification of the appropriate level for tests and full reporting of outcomes |
| ☐ | ☒ | Estimates of effect sizes (e.g. Cohen's $d$, Pearson's $r$), indicating how they were calculated |

*Our web collection on statistics for biologists contains articles on many of the points above.*

## Software and code

Policy information about availability of computer code

| Data collection | No software was used in the data collection. This study used repository data and did not collect new data. |
|---|---|
| Data analysis | Data was analyzed by using a combination of open source R code and custom R code which are available on https://github.com/KaidiK/RESI_BWAS. All visualization and statistics represented in graphical format were generated by using the "ggplot2" R package. For multi-site studies, the site effects were removed in advance using the "neuroCombat" and "longCombat" R packages for cross-sectional and longitudinal studies, respectively. In the boxplots shown, they indicate the median and lower and upper hinges correspond to the first and third quartiles (the 25th and 75th percentiles). The upper whisker extends from the hinge to the largest value at most 1.5 * IQR of the hinge. Data beyond the end of the whisker are called "outlier" points and are plotted individually. The linear regression models were fitted using the "lm" function from base "stats" package. The generalized estimating equations (GEEs) were fitted using the "geepack" R package. The robust effect size index estimates and the confidence intervals were derived using the "RESI" R package. A description of the FreeSurfer version and processing pipeline can be found in SI18. |

For manuscripts utilizing custom algorithms or software that are central to the research but not yet described in published literature, software must be made available to editors and reviewers. We strongly encourage code deposition in a community repository (e.g. GitHub). See the Nature Portfolio guidelines for submitting code & software for further information.

## Data

Policy information about availability of data

All manuscripts must include a data availability statement. This statement should provide the following information, where applicable:
- Accession codes, unique identifiers, or web links for publicly available datasets
- A description of any restrictions on data availability
- For clinical datasets or third party data, please ensure that the statement adheres to our policy

> Participant-level data from many datasets are available according to data access policies of the primary studies. Study-level model parameters are available at https://github.com/KaidiK/RESI_BWAS.

## Research involving human participants, their data, or biological material

Policy information about studies with human participants or human data. See also policy information about sex, gender (identity/presentation), and sexual orientation and race, ethnicity and racism.

| Reporting on sex and gender | We included sex as a biological variable, which was self-reported by study participants. |
|---|---|
| Reporting on race, ethnicity, or other socially relevant groupings | This study focuses on biological and cognitive associations. Social categories (such as race and ethnicity) were not considered in the analyses. |
| Population characteristics | We considered age as a population characteristic. In meta-analyses, mean, standard deviation, and kurtosis of age for each sample was included as covariates. |
| Recruitment | This study used existing data from consortium repositories and did not recruit human subjects. |
| Ethics oversight | All contributing studies include their own ethical oversight. Because this is secondary research and the data are deidentified when accessed from the primary study repositories it does not constitute human subjects research by NIH policy and does not require ethical approval from the IRB at Vanderbilt University Medical Center. |

Note that full information on the approval of the study protocol must also be provided in the manuscript.

# Field-specific reporting

Please select the one below that is the best fit for your research. If you are not sure, read the appropriate sections before making your selection.

☐ Life sciences ☒ Behavioural & social sciences ☐ Ecological, evolutionary & environmental sciences

For a reference copy of the document with all sections, see nature.com/documents/nr-reporting-summary-flat.pdf

# Behavioural & social sciences study design

All studies must disclose on these points even when the disclosure is negative.

| Study description | Meta-analyses based on 63 neuroimaging datasests from Lifespan Brain Chart Consortium (LBCC) were conducted to investigate the influence of study design features that can improve the effect sizes in Brain-wide association studies. |
|---|---|
| Research sample | Sixty-three neuroimaging datasets, which includes 16 longitudinal datasets and and 46 cross-sectional datasets, from Lifespan Brain Chart Consortium (LBCC) were used in this research. |
| Sampling strategy | This study used existing data from consortium repositories. |
| Data collection | This study used existing data from consortium repositories. |
| Timing | n/a |
| Data exclusions | The original LBCC dataset includes 123,984 MRI scans from 101,457 human participants across more than 100 studies.. We filtered to the subset of cognitively normal participants whose data were processed using FreeSurfer version 6.1. Studies were curated for the analysis by excluding duplicated observations and studies with less than 4 unique age points, sample size less than 20, and/or only participants of one sex. If there were fewer than three participants having longitudinal observations, only the baseline observations were included and the study was considered cross-sectional. If a subject had changing demographic information during the longitudinal follow-up (e.g., changing biological sex), only the most recent observation was included. We updated the LBCC dataset with the ABCD release 5 resulting in a final dataset that includes 77,695 MRI scans from 60,900 cognitively normal participants who have available total GMV, sGMV and GMV measures across 63 studies, among these, 74,148 MRI scans from 57,538 participants across 43 studies have complete-case regional brain measures. |

| Non-participation | This study used existing data from consortium repositories and non-participation is reported for those studies at the discretion of the primary study investigators. |
|---|---|
| Randomization | This is an observational study and no treatment or randomization was applied. |

# Reporting for specific materials, systems and methods

We require information from authors about some types of materials, experimental systems and methods used in many studies. Here, indicate whether each material, system or method listed is relevant to your study. If you are not sure if a list item applies to your research, read the appropriate section before selecting a response.

## Materials & experimental systems

| n/a | Involved in the study |
|---|---|
| ☒ | ☐ Antibodies |
| ☒ | ☐ Eukaryotic cell lines |
| ☒ | ☐ Palaeontology and archaeology |
| ☒ | ☐ Animals and other organisms |
| ☒ | ☐ Clinical data |
| ☒ | ☐ Dual use research of concern |
| ☒ | ☐ Plants |

## Methods

| n/a | Involved in the study |
|---|---|
| ☒ | ☐ ChIP-seq |
| ☒ | ☐ Flow cytometry |
| ☐ | ☒ MRI-based neuroimaging |

## Plants

| Seed stocks | n/a |
|---|---|
| Novel plant genotypes | n/a |
| Authentication | n/a |

## Magnetic resonance imaging

### Experimental design

| Design type | Structural and Functional MRI |
|---|---|
| Design specifications | n/a |
| Behavioral performance measures | n/a |

### Acquisition

| Imaging type(s) | Structural and functional |
|---|---|
| Field strength | Imaging protocol varied across consortia and study sites. See reference below for details. |

1. Bethlehem R a. I, Seidlitz J, White SR, Vogel JW, Anderson KM, Adamson C, Adler S, Alexopoulos GS, Anagnostou E, Areces-Gonzalez A, Astle DE, Auyeung B, Ayub M, Bae J, Ball G, Baron-Cohen S, Beare R, Bedford SA, Benegal V, Beyer F, Blangero J, Blesa Cábez M, Boardman JP, Borzage M, Bosch-Bayard JF, Bourke N, Calhoun VD, Chakravarty MM, Chen C, Chertavian C, Chetelat G, Chong YS, Cole JH, Corvin A, Costantino M, Courchesne E, Crivello F, Cropley VL, Crosbie J, Crossley N, Delarue M, Delorme R, Desrivieres S, Devenyi GA, Di Biase MA, Dolan R, Donald KA, Donohoe G, Dunlop K, Edwards AD, Elison JT, Ellis CT, Elman JA, Eyler L, Fair DA, Feczko E, Fletcher PC, Fonagy P, Franz CE, Galan-Garcia L, Gholipour A, Giedd J, Gilmore JH, Glahn DC, Goodyer IM, Grant PE, Groenewold NA, Gunning FM, Gur RE, Gur RC, Hammill CF, Hansson O, Hedden T, Heinz A, Henson RN, Heuer K, Hoare J, Holla B, Holmes AJ, Holt R, Huang H, Im K, Ipser J, Jack CR, Jackowski AP, Jia T, Johnson KA, Jones PB, Jones DT, Kahn RS, Karlsson H, Karlsson L, Kawashima R, Kelley EA, Kern S, Kim KW, Kitzbichler MG, Kremen WS, Lalonde F, Landeau B, Lee S, Lerch J, Lewis JD, Li J, Liao W, Liston C, Lombardo MV, Lv J, Lynch C, Mallard TT, Marcelis M, Markello RD, Mathias SR, Mazoyer B, McGuire P, Meaney MJ, Mechelli A, Medic N, Misic B, Morgan SE, Mothersill D, Nigg J, Ong MQW, Ortinau C, Ossenkoppele R, Ouyang M, Palaniyappan L, Paly L, Pan PM, Pantelis C, Park MM, Paus T, Pausova Z, Paz-Linares D, Pichet Binette A, Pierce K, Qian X, Qiu J, Qiu A, Raznahan A, Rittman T, Rodrigue A, Rollins CK, Romero-Garcia R, Ronan L, Rosenberg MD, Rowitch DH, Salum GA, Satterthwaite TD, Schaare HL, Schachar RJ, Schultz AP, Schumann G, Schöll M, Sharp D, Shinohara RT, Skoog I,

Smyser CD, Sperling RA, Stein DJ, Stolicyn A, Suckling J, Sullivan G, Taki Y, Thyreau B, Toro R, Traut N, Tsvetanov KA, Turk-Browne NB, Tuulari JJ, Tzourio C, Vachon-Presseau É, Valdes-Sosa MJ, Valdes-Sosa PA, Valk SL, van Amelsvoort T, Vandekar SN, Vasung L, Victoria LW, Villeneuve S, Villringer A, Vértes PE, Wagstyl K, Wang YS, Warfield SK, Warrier V, Westman E, Westwater ML, Whalley HC, Witte AV, Yang N, Yeo B, Yun H, Zalesky A, Zar HJ, Zettergren A, Zhou JH, Ziauddeen H, Zugman A, Zuo XN, Bullmore ET, Alexander-Bloch AF. Brain charts for the human lifespan. Nature. Nature Publishing Group; 2022 Apr;604(7906):525–533.

2. Feczko E, Conan G, Marek S, Tervo-Clemmens B, Cordova M, Doyle O, Earl E, Perrone A, Sturgeon D, Klein R, Harman G, Kilamovich D, Hermosillo R, Miranda-Dominguez O, Adebimpe A, Bertolero M, Cieslak M, Covitz S, Hendrickson T, Juliano AC, Snider K, Moore LA, Uriartel J, Graham AM, Calabro F, Rosenberg MD, Rapuano KM, Casey BJ, Watts R, Hagler D, Thompson WK, Nichols TE, Hoffman E, Luna B, Garavan H, Satterthwaite TD, Ewing SF, Nagel B, Dosenbach NUF, Fair DA. Adolescent Brain Cognitive Development (ABCD) Community MRI Collection and Utilities [Internet]. bioRxiv; 2021 [cited 2023 Oct 20]. p. 2021.07.09.451638. Available from: https://www.biorxiv.org/content/10.1101/2021.07.09.451638v1

| | |
|---|---|
| Sequence & imaging parameters | Imaging protocol varied across consortia and study sites. See references above for details. |
| Area of acquisition | Global and regional brain structural MRI and functional MRI were used |
| Diffusion MRI | ☐ Used   ☒ Not used |

## Preprocessing

| | |
|---|---|
| Preprocessing software | Based on Freesurfers recon-all command. We limited our analyses to version 6.1 to reduce software version effects. fMRI data were processed using the abcd-hcp-pipeline version 0.1.3. |
| Normalization | Based on Freesurfers recon-all command |
| Normalization template | Based on Freesurfers recon-all command |
| Noise and artifact removal | Based on Freesurfers recon-all command |
| Volume censoring | None |

## Statistical modeling & inference

| | |
|---|---|
| Model type and settings | We used generalized linear models (GLMs) and generalized estimating equations (GEEs) to estimate study-level effects for cross-sectional and longitudinal studies respectively. We used GLMs to perform meta-analysis of study-level results. |
| Effect(s) tested | *Define precise effect in terms of the task or stimulus conditions instead of psychological concepts and indicate whether ANOVA or factorial designs were used.* |

Specify type of analysis:   ☐ Whole brain   ☐ ROI-based   ☒ Both

| | |
|---|---|
| Anatomical location(s) | Cortical gray matter, subcortical gray matter, white matter, and cortical thickness |

| | |
|---|---|
| Statistic type for inference<br><br>(See Eklund et al. 2016) | We used GLMs and GEEs to model global and regional measures of structure and function. For regional analyses, we applied Benjamini-Hochberg to adjust for multiple comparisons and report effect sizes and replicability curves. |
| Correction | For regional analyses, we applied the Benjamini-Hochberg procedure to control the False Discovery Rate. |

## Models & analysis

| n/a | Involved in the study |
|---|---|
| ☒ ☐ | Functional and/or effective connectivity |
| ☒ ☐ | Graph analysis |
| ☒ ☐ | Multivariate modeling or predictive analysis |

