## [Peer Review file · Nature]

Manuscript Title: Study design features increase replicability in brain-wide association studies

Reviewer Comments & Author Rebuttals

Reviewer Reports on the Initial Version:

Referee #1 (Remarks to the Author):

Summary:

In this manuscript Kang et al. utilize a collection of large-scale brain imaging data sets to compare the univariate effect sizes of associations between brain volumes (gray, white) and age, sex and cognitive metrics for different sampling methods.

They contrasted effect sizes for longitudinal and cross-sectional sampling and found that across studies the effect size for brain volume to age is much larger (up to 290%), when sampling longitudinally instead of cross-sectionally. As expected, effects sizes for sex only increased minimally, if at all, with longitudinal sampling, since biological sex does not change across time points.

They also investigated the benefits of oversampling the distribution tails of the independent variable. Depending on the sampling scheme and dataset, oversampling the tails increased effect sizes for age by ~ 20-40% for cross-sectional designs and 10-20% for longitudinal. However, increasing the standard deviation of age within-subject did not increase effect sizes. Thus, they recommend longitudinal studies with large between-subject spacing in age when studying the effects of age on brain volumes.

Effect sizes for associations between brain volumes and cognitive measures were much smaller than those for age. Associations between brain volumes and cognitive measures were also boosted by oversampling the tails of the distribution in an effect-size dependent manner. In contrast to age, effect sizes for associations between brain volumes and cognitive measures (age adjusted) did not get larger with longitudinal sampling.

Big picture points:

This study has many strengths. Foremost the use of 63 total datasets with ~61,000 participants in sum. It investigates an important BWAS question, namely how study designs and sampling strategies can affect effect sizes. The findings that longitudinal designs and sampling strategies that increase the standard deviation of the age distribution enhance associations between age and global structural brain metrics are very convincing. The article makes a very strong case that age effects should be studied longitudinally, and that the between-subject SD for age should be large. In addition, the study suggests that we do not have to obsess over the time interval between repeated within-subject measurements.

This exciting work brings up a series of interesting questions. One set of questions is related to the potential generalizability of the findings. Specifically, whether the findings hold for the popular BWAS metrics of functional connectivity and cortical thickness. What are the effects of analyzing a global structural metric with little measurement error? Would much noisier connection-wise functional connectivity follow similar patterns? How would the findings generalize to other phenotypic measures beyond the six cognitive scores, age and sex analyzed here, especially measures of psychopathology, and various medical variables?

The finding that a longitudinal design increases effect sizes for age, but not cognitive variables indicates that the specific variables of interest are important. Age-normed cognitive metrics are not supposed to vary on re-testing (e.g., IQ), beyond state effects (sleep, stress, caffeine state, etc.) and measurement error. Most BWAS variables differ from age and sex in that they have significant measurement error. In addition, functional brain measures (FC, fMRI) do vary with state effects (in contrast to brain volumes), thus repeated functional sampling (longitudinal) could increase effect sizes even for age-adjusted variables, especially when measurement errors are high. These questions interact with questions about the effects of diverse multi-variate BWAS methods for different brain metric/non-brain metric pairings.

Oversampling the tails of a distribution is a popular approach, for example in case-control GWAS, and this study argues in its favor more broadly since it boosted effect sizes for age and the six cognitive variables evaluated. Several generic concerns are sometimes raised as potential drawbacks when discussing the benefits of tail oversampling, at least in its more extreme forms. One previously raised concern is that some brain/behavioral phenotype relationships, or a subset of features, may be non-linear, including u- or inverted u-shaped. Could under-sampling the middle of a distribution increase power at the cost of distorting the underlying relationships? For example, if using multi-variate methods to predict a disease phenotype, the prediction could be biased towards features that best separate the most severe disease state, at the expense of being able to categorize intermediate severities. In addition, some work has shown that dichotomizing variables increases the risk for confounding (Austin, *Brunner Stat. Med.*, 2004), might the same be true for oversampling the tails?

Despite potential drawbacks of tail oversampling, especially when thinking about multivariate methods, the manuscript clearly provides strong examples in its favor, that in most scenarios should outweigh concerns over misrepresenting the underlying physiology. Therefore, if a study is being designed to investigate a single brain-behavior association (e.g., age & brain volume), or perhaps a narrow set of questions, tail oversampling appears to be the winning approach.

For massive datasets (e.g., ABCD, UKB) that will be used for myriad different research questions, the question of tail oversampling becomes more complicated. Is it possible to oversample the tails of many/most distributions? Furthermore, oversampling requires confidence that the most important metric is known, measured accurately and neurobiologically meaningful. In the case of age all of those are of course true, but it seems to get more complicated for psychopathological questionnaires such as the CBCL, which seem to have little correlation with brain metrics. The current study shows that oversampling does not boost very small brain-behavior associations for some of the cognitive measures. The BWAS effect size for new phenotypic metrics cannot be known

prior to conducting a large sample study, thus oversampling the tails carries the risk of designing a study around metrics without neurobiological correlates. Hence, beyond easily accurately measured biological variables (e.g., age, sex) population sampling might be the safest approach, especially if the large-scale dataset is supposed to serve many research groups if a wide variety of questions of interest. If for example a dataset were to be constructed for BWAS of psychiatric disorders, would it be most prudent to oversample patients with severe symptoms in many dimensions, or would it be better to include patients with extreme anxiety, psychosis, depression, but minimal other symptoms? For the other end of the distribution, are there risks of confounding if the sample is enriched for participants without any psychiatric symptoms, such as higher SES, more likely to be cis-gendered?

Currently, the manuscript does not discuss the potential benefits of perhaps the most extreme forms of brain-behavior (non-BWAS) effect size boosting: interventions and brain lesions. While the data cannot directly speak to the effect sizes of interventions (e.g., medications, neuromodulation, etc.), it might still be worth adding a sentence about RCTs and other within-patient clinical study designs.

Minor detail points:

The abstract and the first section of the main manuscript overlap quite a bit. Some of the redundancies could likely be trimmed out.

Lines 44, 45: Age & sex are not complex behavioral phenotypes and therefore not BWAS variables.

Lines 260-63: Could you give more detail as to what types of 'more complex designs' might be needed to increase effect sizes for cognitive and similar metrics? Are there relevant references?

Lines 265-66: This sentence highlights the 290% effect size increase going from cross-sectional to longitudinal, without additional specifics. It seems that this value is specific to the association between gray matter volume and age, if so, it would be good to include that, perhaps in parentheses?

Line 292-93: The comment "compared to the primary focus on the ABCD sample in previous papers" is missing references. The word 'papers' sounds a tad colloquial in this sentence.

The use of the RESI, CS-RESI, and L-RESI terminology is slightly confusing. For example, the subheading: "Cross-sectional RESI (CS-RESI) quantifies benefit of longitudinal design," creates cognitive conflict. Why call something that quantifies the benefit of longitudinal designs the Cross-sectional RESI?

In Table 1 the heading titles of "Outcome: RESI or L-RESI" and "Outcome: RESI or CS-RESI" are also confusing. Which one is? The confusion is increased because Table 2 has a column labeled RESI or CS-RESI and one labeled L-RESI, if viewed in isolation Table 2 would suggest that RESI and CS-RESI are equivalent, but not L-RESI.

Figure 1 is titled “Partial regression plots of a meta-analysis of RESI and L-RESI for [...],” but all the plot y-axes are labeled RESI.

Could Figure 2a,b be compressed by just showing curves of the ES increase for longitudinal designs, instead showing L-RESI minus RESI and CS-RESI minus RESI?

In Figure 2c-f how are the studies sorted vertically? If they are not already, would it be possible to also sort them by sample size and maybe add a single y-axis that gives the readers an idea of the sample size distributions. Panels c-f suggest that the benefits of longitudinal sampling are less apparent in the more lifespan samples (first 3 rows) in contrast to sample that focused on just younger or just older participants. Could that be because those samples are more likely to capture non-linear relationships?

In Fig. 3b the 30,000 label on the x-axis is partially cut off.

Instead of labeling the sampling schemes 0,1,2, perhaps they could be given succinct descriptive labels (e.g., replace 0 with ‘flat’), which would make it quicker to parse the findings. And perhaps the same could be done for the within-subject sampling schemes which are currently also labeled scheme 0,1.

In Fig. 4, it might be helpful to explicitly say in the figure caption text that the data are from the ABCD. For someone familiar with the different data sets, it can be deduced, but it might be easier for readers, to have it re-stated.

Referee #2 (Remarks to the Author):

This study provides a useful contribution to contemporary discussions about effect sizes in neuroimaging and their implications for sample size requirements, the detectability and reproducibility of findings, and the use of neuroimaging for person-level biomarkers. It is based primarily on analyses relating whole-brain structural measures (gray-matter volume [GMV] and several others) to age and sex, with supplementary analyses of selected cognitive variables. The main findings is that larger variation in the outcome (or phenotype; age and sex) is associated with larger effect sizes (ESs) across studies. In addition, longitudinal designs had substantially larger effect sizes than cross-sectional ones, at least for age and to some degree for sex.

Strengths of the paper include (1) the timeliness of the findings, particularly given widespread confusion about and overgeneralization of previous findings showing small BWAS effect sizes, and (2) its ability to provide a more nuanced view and highlight the importance of some variables, particularly outcome variance, that are not widely appreciated.

One conceptual limitation is that the relationship between larger ES and higher predictor variance is a well known feature in basic statistics, based on the mechanics of linear models. At an intuitive level, a study that compares 12 year olds to 50 year olds will have much larger ESs than one that compares 49 to 50 year olds. This principle is covered in basic statistics books but may not be widely appreciated by a neuroimaging audience. However, the basics are not discussed here and perhaps should be (one good book on this is "Data analysis" by Judd, McClelland, and Ryan; I'm sure there are many others with good sections). For example, it should be possible to qualitatively predict the degree of ES increase with extreme sampling (fig 3) and possibly other results. Could the authors characterize the degree to which their findings are explained by simple statistical mechanics of increasing predictor variance in the GLM framework?

The study does offer more nuance, including (1) derivation and use of robust ES estimates, a strength; (2) discussion of differences between between-person and within-person variance and when longitudinal sampling might be expected to increase ESs and when it won't (i.e., when the relationship of interest is strong within-person, and when there are many sources of between-person noise), (3) Some basic large-sample benchmarking of ESs for age and sex, as a reference point, and (4) Some cases where longitudinal sampling isn't helpful, which could provide clues about when a longitudinal study is important (it is for age, but it's less clear what the benefit is for other variables).

Some limitations and things to improve related to these might be: (1) Provide clear mapping from RESI ESs onto power and sample size requirements; (2) Provide more concrete examples and perhaps a mathematical account of between- and within-person error variance and how it relates to the ES benefit with longitudinal designs (can this be mapped onto the data here to explain the pattern of findings better?); (3) Consider a wider range of variables and the implications of the restricted focus on age, at least in discussion.

I elaborate on some of these briefly below.

1. Can you translate the main results into concrete benefit in power and sample size required for 80% power, for broad audience? This may help readers familiar with Cohen's d to understand the implications of a RESI of, e.g., 0.1. What sample sizes are required to detect effects under different conditions studied? (See, e.g., Spisak 2023 Nature for some interpretable plots). How do these compare with effect sizes in, e.g., Marek 2022, Nature?

2. The findings illustrate statistical principles in a very large sample, but with a very limited number of brain variables and phenotypic predictors. At a statistical level, it stops short of providing a principled and quantitative account of when precisely effect sizes should be larger as a function of within- and between-person predictor variance. This type of contribution would be expected from a statistical paper of this type, and would enhance the value and generalizability of the paper. Analysis by, e.g., Desmond and Glover actually goes farther in this respect, and the mathematical foundations of variance components in multi-level settings could be gainfully employed here.

3. In the introduction, the authors may want to consider a larger conceptual space of variables that moderate BWAS effect sizes, including the reliability of both phenotypic (outcomes) and brain measures; the univariate vs. multivariate nature of the relationship; and whether phenotypes are more or less likely to relate to type types of brain measures studied (e.g., Spisak et al. 2023 Nature; Rosenberg and Finn 2022 Nat Neuroscience). Here, the point is well taken that effect sizes in Marek et al. reflect a mix of null and non-null phenotypes, but the generalizability of the claims here are limited by a focus on only a few global brain measures (e.g., GMV) and sex/age, which are among the very few phenotypes expected to be related to these brain measures. The effect sizes here may thus represent an upper bound. Age and sex have (1) high phenotypic reliability (low-noise measures), (2) relatively high reliability and stable genetic encoding of brain structure (e.g., Elliot 2022 Nature, Duff et al. 2022), and (3) likely stronger true relationships between brain and phenotype than many other phenotypic measures.

Thus, the paper makes the basic point it intends to make in a large sample, but it's hard to see when the conclusions will generalize more broadly and when they won't.

Here's one more specific example. Longitudinal designs that eliminate between-subject error variance are helpful in some cases, but not others (some of the cognitive variables). Is this because more data per person reduces error variance and increases power, in accordance with the central limit theorem? And/or are cognitive effects not associated because they are unreliable, vary over time between assessments, and/or are simply unrelated to global GMV?

The findings here may be specifically related to age as well, and may not generalize. For example, the paper does not consider time between assessments, which is key for longitudinal studies because longer times increase variance related to slowly changing processes like age (increasing ESs), but decrease ESs for variables that may fluctuate more rapidly (e.g., cognitive outcomes? e.g., with season, stressors, etc.). Considering effect size per year, and per year during development or during GMV decline with aging, may help in part. But: will the findings on when longitudinal designs are helpful or not generalize beyond age and the few variables tested here?

The authors conclude: “longitudinal studies have larger standardized ESs than cross-sectional studies”. This broad conclusion may not be warranted, and may apply rather specific to age. The intuitive reason would be that gray matter changes a lot across time, so within-person relationships between age and GMV may be particularly meaningful. Missing is an assessment of whether cognitive variables also have larger ESs in longitudinal studies — e.g., something like Fig 1A for cognitive outcomes.

Minor comments

“increasing the within-subject SD of the cognitive measurements did not increase CS-RESI or L-RESI”
Does this imply that brain structure is not related to within-person variation in cognitive scores?
Could this be because the cognitive scores are not reliable, or because brain structure does not vary over time as a function of score?

To increase generalizability, the authors might also consider which kinds of outcomes are likely to have large ESs. An interesting result is that composite cognitive measures, which presumably average over distinct kinds of task-specific measures (e.g., the rationale behind factor analysis in traditional psychometrics), may have larger ESs in Fig 4, and some measures that are notoriously unreliable because they are based on reaction time difference measures (e.g., Flanker) have ESs near zero.

“adding a second longitudinal measurement per subject” helps. Because more data are available, or because of a within-subject effect? And what is the time lag?

Fig 2: Minor questions: Effect sizes are for which brain measure (GMV only)? Why were these two selected for display? Not because they were the largest, as ABCD is quite a bit larger.

Fig 2A: Why the small-sample bias towards larger effect sizes? Suggests some kind of model fitting (i.e. overfitting), which I don’t think is the case. Is it that RESI is downwardly biased (toward 0) for small samples (e.g., RESI in Fig 4B is smaller for small samples).

Longitudinal designs show larger effects. Calling it a “bias” (e.g., p. 4) may be misleading as there are fewer sources of error variance and more data in longitudinal analyses, reducing error variance and increasing ES. May conflate the statistical properties of averaging more data and assessing within-person effects with biological properties that total GMV may vary within-person with age (during development and aging in particular) ... masking....

But more scans in ADNI don’t help much (fig 3E), so it’s not likely averaging over noise, but rather avoiding between-person confounds and nuisance variables.

Were cognitive measures always collected at the same time as imaging measures? The former may vary as a function of state (within-person), which might explain the lack of advantage of longitudinal designs if they are not collected at the same time. Is it possible that if cognition and brain were collected simultaneously the picture could change? Maybe this is addressed by fig S8?)

“for the GMV-age association ESs were largest in young and late adulthood when age-related changes in GMV are strongest”. I’m not sure this is the case, as the “Brain scales” 2022 Nature paper shows the most rapid change around 1 year, and little change in early adulthood.

<https://www.nature.com/articles/s41586-022-04554-y/figures/1>. What was the coverage of early life in the present subsample?

“our focus on age effects and broad cerebral tissue classes is likely to find stronger ESs”. From a statistical averaging perspective yes, but not necessarily, as previous studies have shown that some brain areas are particularly sensitive. I agree that Marek et al. and some other studies average over null and non-null findings (an important point!) It may be that if the most relevant brain measures were used, particularly with predictive models (e.g. Marek 2022’s and Spisak 2023’s multivariate models) the effects could be substantially larger still. This may warrant discussion.

“guarantee larger ESs when the within-subject correlation of the outcome and the within-subject correlation of the independent variable are both positive”. This is confusing — should one be “between”?

I’m familiar with applying ComBat on whole image data before extracting measures of interest, but here it seems that GMV and other measures were first extracted using FreeSurfer, and then ComBat was applied before analysis. It’s not clear to me how ComBat isn’t redundant with the linear model that adjusts for batch effects in this case. Perhaps it has to do with the multiplicative error model? The error model was not specified in the methods I believe.

More detail on knot point selection, both in terms of number and position, would be helpful in the Methods.

How were datasets acquired and where are they available, if they are publicly available? Which are available from which sources? The data availability statement is vague.

sGMV appears to be partially redundant with GMV. Is that the case, and if so, why are they treated as separate outcomes? Why not separate into cortical GMV and subcortical GMV (sGMV)?

What does it mean to show a negative effect size in this context?

Referee #3 (Remarks to the Author):

The authors examine the role of non-brain variable distribution in the consideration of power for brain/non-brain associations. Using a robust effect size measure they've developed, they conduct a number of investigations to explore effect size and its relationship to experimental design. In the Life Brain Chart Consortium, they find that a study's mean age and age standard deviation both impact effect size (but not age skew), and longitudinal studies have increased effect size. Using resampling analyses with ADNI, they confirmed the impact of longitudinal analyses, but found reducing number of repeated observations didn't reduce power as long as within SD of age was maintained. With ABCD, however, longitudinal effect sizes for brain-behavior associations were not stronger than cross-sectional ones.

Major issues

While this is a useful contribution I find its impact limited as many of the conclusions are tenants of experimental design: Power of a regression increases with linear effect or independent variable variance. (Maybe simplest demonstration of this is with simple linear regression, where $\beta = r_{XY} * SD(Y) / SD(X)$, or in terms of correlation, $r_{XY} = \beta * SD(X) / SD(Y)$. Thus for a given dependent variable Y and linear slope, correlation and thus power increases with SD(X).) Figure 1 column 2 is a direct result of the impact of SD(X) (some studies have narrow age range, others wider). The careful exposition of these results is not without value, but I don't see the general-audience appeal of the findings.

Also, I find the ultimate utility of the work is limited by the authors not directly addressing how to deploy these findings in practice. That is, unlike designing the stimuli timing for a task fMRI study, the setting of large scale studies is limited by a constellation of factors, often cost. What exactly *are* the concrete recommendations of this work? Should researchers *subset* a large-scale dataset to find subsets of phenotypes with large variance? I.e. sacrifice total N to increase SD(X)? Or should they find ways to inexpensively test 1000's of potential subjects, to then only recruit for scanning the extreme range of a phenotype? Extreme phenotype designs have been used for some time in genetics, most recently for studying rare variants (see e.g. <https://www.ncbi.nlm.nih.gov/pmc/articles/PMC7564972/>) and is an example of putting these concepts into practice.

Related, are the practical implications of the findings on cross-sectional vs. longitudinal studies. In the context of brain imaging, a dominant cost factor can be scanning time, and then a reasonable reference is *not* comparing longitudinal with N subjects to a cross-sectional study, but rather a longitudinal study with $n = \sum_i n_i$ measurements to a cross-sectional study with n subjects. Even if a resampling approach isn't clear, perhaps even a simple adjustment (e.g. using a factor n/N) could easily address this question and enrich the results.

Another concern is the use of the variable age, which has various special aspects, but notably explains a terrific amount of variance. I see a similar body of work has just appeared <https://www.nature.com/articles/s41562-023-01642-5> using different measures of effect size, and the added value relative to that work should be discussed. Also, it seems notable that longitudinal

effect sizes were dramatically stronger for age, but not for behavior variables in ABCD... this seems like a paradox but it was never discussed or unpacked.

Regarding the exposition, given that the core of this work concerns effect size, there a surprising lack of clarity about what exactly is meant by "standardized effect size". The issue is that there is no one single measure of effect size, as the exact form of "standardized effect size" varies depending on whether one is working with a one-sample, two-sample or regression setting, no less considering complexities of mixed models [see e.g. Lakens (2013)]. The working assumption of the authors seems to be that the reader is well-familiar with RESI; notably, despite being a centerpiece of the work, RESI is never defined in the body or methods (it is in the supplementary materials). While it is understood that the audience is not statistical, some sort of exposition on RESI should be given in the methods of this work. (E.g. the connection to $d/2$ seems arbitrary; less arbitrary, it seems, is to simply say that RESI is robust version of Cohen's f).

Regarding the choice to focus on RESI, it isn't clear whether RESI be viewed as a convenience, or a unique and special element of this work. If the latter, will these findings generalize should the user choose other measures of effect size? E.g. what if the analyst used linear mixed effects models instead of a GEE? In this univariate (not voxelwise) setting, LME would seem to be the natural choice, requiring comment on the applicability of CS-RESI/L-RESI comparisons when LME is used with longitudinal data.

Minor Issues.

Line 158: There is a cryptic reference to bias: "The bias we note..." This is the first mention of any bias in effect size measures; a careful study of the referenced Fig. 2 does not explain what is being referred to here.

Line 687: Related to Line 158, is the caption of Fig. 2, which refers to a "modified CS-RESI", which doesn't really make any sense... i.e. if Fig. 2A plots CS-RESI - RESI, how can that be "removed" without giving a zero value? Or, more simply, what is the value of Fig 2B? Note also there is no other reference to "modified CS-RESI" anywhere else in the submission.

References

Lakens, D. (2013). Calculating and reporting effect sizes to facilitate cumulative science: A practical primer for t-tests and ANOVAs. *Frontiers in Psychology*, 4(NOV), 1, 12.
<https://doi.org/10.3389/fpsyg.2013.00863>

Author Rebuttals to Initial Comments:

Referee #1 (Remarks to the Author):

Summary:

In this manuscript Kang et al. utilize a collection of large-scale brain imaging data sets to compare the univariate effect sizes of associations between brain volumes (gray, white) and age, sex and cognitive metrics for different sampling methods.

They contrasted effect sizes for longitudinal and cross-sectional sampling and found that across studies the effect size for brain volume to age is much larger (up to 290%), when sampling longitudinally instead of cross-sectionally. As expected, effects sizes for sex only increased minimally, if at all, with longitudinal sampling, since biological sex does not change across time points.

They also investigated the benefits of oversampling the distribution tails of the independent variable. Depending on the sampling scheme and dataset, oversampling the tails increased effect sizes for age by ~ 20-40% for cross-sectional designs and 10-20% for longitudinal. However, increasing the standard deviation of age within-subject did not increase effect sizes. Thus, they recommend longitudinal studies with large between-subject spacing in age when studying the effects of age on brain volumes.

Effect sizes for associations between brain volumes and cognitive measures were much smaller than those for age. Associations between brain volumes and cognitive measures were also boosted by oversampling the tails of the distribution in an effect-size dependent manner. In contrast to age, effect sizes for associations between brain volumes and cognitive measures (age adjusted) did not get larger with longitudinal sampling.

Big picture points:

R1.1: This study has many strengths. Foremost the use of 63 total datasets with ~61,000 participants in sum. It investigates an important BWAS question, namely how study designs and sampling strategies can affect effect sizes. The findings that longitudinal designs and sampling strategies that increase the standard deviation of the age distribution enhance associations between age and global

structural brain metrics are very convincing. The article makes a very strong case that age effects should be studied longitudinally, and that the between-subject SD for age should be large. In addition, the study suggests that we do not have to obsess over the time interval between repeated within-subject measurements.

Thank you for these positive comments about our work.

R1.2: This exciting work brings up a series of interesting questions. One set of questions is related to the potential generalizability of the findings. Specifically, whether the findings hold for the popular BWAS metrics of functional connectivity and cortical thickness. What are the effects of analyzing a global structural metric with little measurement error? Would much noisier connection-wise functional connectivity follow similar patterns? How would the findings generalize to other phenotypic measures beyond the six cognitive scores, age and sex analyzed here, especially measures of psychopathology, and various medical variables?

The generalizability of the results was a really important limitation of our previous draft that we aimed to address in the revisions. To increase the generalizability of the findings, we've added regional analyses of GMV and global and regional CT analyses. These outcomes were added for the meta-analysis and the UKB and ADNI analyses. We also added all those brain measures as well as functional connectivity (FC) as outcomes in the ABCD analyses. For the ABCD analyses, we also added measures of psychopathology (CBCL) and demographic variables (BMI, birthweight and handedness) to the cognitive variables we used as covariates in the initial submission. The additional analyses are reported there and in Figure 5:

Fig. 5. Heterogeneous improvement of effect sizes for cognitive, mental health, and demographic associations with structural and functional brain measures in the ABCD study at N = 500. (a) U-shaped between-subject sampling scheme (blue) that increases between-subject variability of the non-brain covariate produces larger effect sizes and (b) reduces the number of participants scanned to obtain 80% power in total GMV. The points and triangles are the average effect sizes across bootstraps and the whiskers are the 95% confidence intervals. Increasing within-subject sampling (triangles) is sometimes detrimental (i.e., reduces effect sizes). A similar pattern holds in (c-d) regional GMV and (e-f) regional CT. In contrast, (g) regional pairwise FC (functional connectivity) effect sizes increase by increasing between- (blue) and within-subject variability (dashed borders) with a corresponding reduction in the (h) number of participants scanned for 80% power.

Although adding these brain and non-brain measures highlights the heterogeneity in the benefit of modifying longitudinal design to strengthen the effect size, it also indicates there is typically a benefit to increasing between-subject variance. Additionally, we also added Figure 6a-d to show the detrimental impact of using longitudinal design in the ABCD dataset for some associations.

Overall, there is a consistent benefit to increasing between-subject variance of the covariate (Fig. 5), which led to >1.8 factor reduction in sample size needed to scan and >1.4 factor increase in ES for over 50% of associations. Moreover, 72.1% of covariate-outcome associations had increased ESs by increasing between-subject sampling (Fig. S9).

Surprisingly, however, increasing within-subject variability decreases ESs for many structural associations (Fig. 5a-f), suggesting that conducting longitudinal analyses can result in decreased power compared to cross-sectional analyses. To confirm this in ABCD, we compare the ESs of the non-brain measures for each data type estimated using only the baseline measurements to the ESs estimated using the full longitudinal data (Fig. 6a-d). Consistent with the detrimental effect of increasing within-subject variability, for most structural associations (GMV and CT), conducting cross-sectional analyses using only the baseline measures results in larger ESs (and therefore, higher power and replicability) than conducting longitudinal analyses using the full longitudinal data (Fig. 6a-c). For the FC outcomes, there is a slight positive effect of conducting longitudinal analyses (Fig. 6d). Identical results are shown using linear mixed models (LMM) with random intercepts, which are commonly used in BWAS (Fig. S10). Together, these results suggest that the benefit of conducting longitudinal studies and adjusting the within-subject variability is highly dependent on the brain-behavior association and, counterintuitively, longitudinal designs can be detrimental to ESs.

Fig. 6. Longitudinal study designs can be detrimental to effect sizes and replicability due to heterogeneous between- and within-subject associations between brain and behavior measures. (a-c) Cross-sectional analyses (indicated by “1”s on the x-axes) can have larger effect sizes than the same longitudinal analyses (indicated by “2” or “2+”) for structural brain measures in ABCD. **(d)** The FC measures have a slight benefit of longitudinal modeling. The cross-sectional analyses only use the baseline measures; the longitudinal analyses use the full longitudinal data. **(e)** Most regional GMV associations (Fig. 5c) have larger between-subject parameter estimates ($\hat{\beta}_b$, x-axis) than within-subject parameter estimates ($\hat{\beta}_w$, y-axis; see Supplement: Eqn (13)), whereas **(f)** FC associations (Fig. 5g) show more heterogeneous relationships between the two parameters.

To further explain why different outcome-covariate pairs have different benefits of increasing the covariate within-subject variance, we added a final paragraph and additional analyses to that section connecting statistical theory to the empirical results to describe how differences in the between-subject and within-subject associations between outcome and covariate lead to different benefits. We describe this below in response to your comment R1.3.

R1.3: The finding that a longitudinal design increases effect sizes for age, but not cognitive variables indicates that the specific variables of interest are important. Age-normed cognitive metrics are not supposed to vary on re-testing (e.g., IQ), beyond state effects (sleep, stress, caffeine state, etc.) and measurement error. Most BWAS variables differ from age and sex in that they have significant measurement error. In addition, functional brain measures (FC, fMRI) do vary with state effects (in contrast to brain volumes), thus repeated functional sampling (longitudinal) could increase effect sizes even for age-adjusted variables, especially when measurement errors are high. These questions interact with questions about the effects of diverse multi-variate BWAS methods for different brain metric/non-brain metric pairings.

This is an important point that the findings from the first draft touch on that we discuss in much more details in the revision: state/trait association of the non-brain and brain measures. We refer to these as between- and within-subject associations in the paper and show that the relationship between the between- and within-subject associations controls the benefit of conducting a longitudinal study or increasing within-subject variance. We added the longitudinal FC from the ABCD to further investigate this association. The hypotheses you described are correct – increasing within-subject variability was only beneficial when there were large within-subject associations between the covariate and brain imaging outcome. For this reason, FC benefits from increasing within-subject variance with the cognitive and psychopathology measures, but GMV does not (see our response to R1.2). To further clarify these points, we added the last paragraph of the section *Preferred sampling schemes and longitudinal designs depend on state versus trait associations between brain and non-brain measures*, which describes how the relationship between the between- and within-subject associations will affect the benefit of longitudinal designs and provides practical recommendation for analyses

To investigate why increasing within-subject variance or using longitudinal designs is not beneficial for some associations, we examined an assumption common to widely used LMMs and GEEs in BWAS: that there is consistent association strength of the brain and non-brain measure across between- and within-subject changes in the non-brain measurement. These effects are independent of the random effects parameters and can differ if there are different state or trait associations between the brain and non-brain measure. For example, measures of Crystallized Intelligence are subject to within-individual differences in state more than total GMV, which has low state variance so there is no within-subject association between these variables (Tab. S12). In contrast, FC measures are also subject to within-individual differences in state so can have stronger within-subject associations with Crystallized Intelligence. To demonstrate this, we fit models that estimate distinct effects for between- and within-subject associations in ABCD and find that there are large between-subject parameter estimates and small within-subject parameter estimates in total and regional GMV (Tab. S13; Fig 6e), whereas the FC associations are distributed more evenly across between- and within-subject parameters (Fig. 6f). If the between- and within-subject associations are different, fitting these associations separately avoids averaging the larger effect with the smaller effect and can inform our understanding of brain-behavior

associations. This approach ameliorates the detrimental effect of the longitudinal design we saw with structural brain measures in the ABCD (Fig. S11). See Supplemental Section 2 for technical details.

Supplementary Section 2 now has a description of three model fitting procedures that can be used for longitudinal BWAS:

2 Longitudinal Design Considerations

Two factors affect standardized effect sizes such as equation (3): the value of the target parameter (determined through the term involving β) and the efficiency of the estimator of the target parameter (determined by Σ_β). In the main paper, we discuss how between- and within-subject effects may differ across brain non-brain association pairs. When this is true, but the longitudinal model falsely assumes they are equal, then the expected value of the estimated parameter is a weighted average of the two. To understand the expected value of the parameter estimate is affected, we assume the true model

$$\begin{aligned} Y_{ij} &= \beta_0 + \beta_b \bar{x}_i + \beta_w (x_{ij} - \bar{x}_i) + \epsilon_{ij} \\ \text{Var}(\epsilon_{ij}) &= \Sigma_\epsilon, \end{aligned} \quad (9)$$

where we treat Σ_ϵ as a known exchangeable covariance matrix. $i = 1, \dots, n$ indexes the participant and $j = 1, \dots, m$ indexes the number of measurements, assumed to be the same across all participants. This model encompasses linear mixed effects models and generalized estimating equations. β_b represents the between-subject trait-related effects and β_w is the within-subject state-related effects. See [3] for the distinction of random effects and between- and within-subject effects.

Unless explicitly parameterized, as in (9), the fitted model for longitudinal data in BWAS often assumes a mean model with $\beta_b = \beta_w$,

$$Y_{ij} = \beta_0 + \beta x_{ij} + \epsilon_{ij}. \quad (10)$$

This is the most common type of linear mixed models used in BWAS. We hypothesized in the main paper that the detrimental effects of increasing within-subject variance or using a longitudinal study were due to the use of model (10) when model (9) was correct. When model (10) is not correctly specified, the expected value of the estimator of β is a combination of the underlying true cross-sectional and longitudinal effects given by

$$\begin{aligned} \mathbb{E}\hat{\beta} &= \beta_b \left[\frac{(1 - \rho)\rho_x^2}{(1 + (m - 1)\rho)(1 - \rho_x^2) + (1 - \rho)\rho_x^2} \right] \\ &+ \beta_w \left[\frac{(1 + (m - 1)\rho)(1 - \rho_x^2)}{(1 + (m - 1)\rho)(1 - \rho_x^2) + (1 - \rho)\rho_x^2} \right] \end{aligned} \quad (11)$$

where $\rho_x^2 = \frac{\sigma_b^2}{\sigma^2}$, σ_b^2 and σ^2 denote the between-subject and total variance of the covariate, x_{ij} , and ρ is the correlation of the errors ϵ_{ij} . Derivation of this value and subsequent values are given in Section 2.4. Assuming the variance matrix, Σ_ϵ , is known, the variance of $\sqrt{n}\hat{\beta}$ is

$$\text{Var}(\sqrt{n}\hat{\beta}) = \left[m \frac{(1 + (m-1)\rho)\sigma_w^2 + (1-\rho)\sigma_b^2}{\sigma_y^2(1 + (m-1)\rho)(1-\rho)} \right]^{-1},$$

where σ_w^2 is the within-subject variance of the covariate x_{ij} and all other objects are as defined above. The signed RESI effect size (4) is a function of the expected value of $\hat{\beta}$ and its variance.

$$S_{\text{sgn}} = \frac{\mathbb{E}\hat{\beta}}{\sqrt{\text{Var}(\sqrt{n}\hat{\beta})}} = m^{1/2} \{ (1 + (m-1)\rho)\sigma_w^2 + (1-\rho)\sigma_b^2 \}^{-1/2} \sigma_y^{-1/2} \\ \times \left\{ \beta_b \sigma_b^2 \left[\frac{(1-\rho)}{1 + (m-1)\rho} \right]^{1/2} + \beta_w \sigma_w^2 \left[\frac{(1 + (m-1)\rho)}{(1-\rho)} \right]^{1/2} \right\}. \quad (12)$$

Equation (11) implies that, when using a single parameter to model cross-sectional and longitudinal effects, modifications of the longitudinal study design can change the target parameter, as well as the estimation efficiency, so can affect the numerator and denominator of the effect size (12). The formula is visualized in Figure 6. The expected value of the parameter estimator is a convex combination of the between- and within-subject parameters. The expected value of the effect size estimator is a linear combination of β_b and β_w , where the weights are complicated functions of the within subject correlation of the outcome, the number of measurements, and the between- and within-subject variance of the covariate. As shown in the main paper (Fig. 6e-f), the between- and within-subject effects β_b and β_w can differ. In the paper, because our sampling schemes control the variance at the first timepoint and the relative squared distance from the first timepoint, the parameters we estimate are those given in equation (13).

When fitting a cross-sectional version of model (10) to a randomly selected observation from each subject, as in (6), the expected value of the parameter is

$$\mathbb{E}(\hat{\beta}_{CS}) = \beta_b \rho_x^2 + \beta_w (1 - \rho_x^2).$$

This is derived in Section 2.4, but is also obtained by setting $\rho = 0$ and $m = 0$ in (11). The implication of these results are that the longitudinal model (10) is estimating a different parameter than an equivalent cross-sectional model because it incorporates the correlation of the outcome into the weighting. Changing features of a longitudinal design change the target parameter of (10), which will also affect the effect size. In most cases, if the between- and within-subject associations are different, they should be estimated with separate parameters. We propose three different strategies that can be used to model the association between the brain and non-brain covariate in BWAS. The first assumes that the between- and within-subject effects are equal and the second and third assume they are different.

2.1 Equal between- and within-subject effects

When $\beta_b = \beta_w = \beta$ then there is no model misspecification and increasing the between- or within-subject variance of the covariate increases effect sizes for the parameter β . In this case, the parameters β_b and β_w can be distributed out of the sum in (12) and the whole effect size increases linearly with respect σ_w and σ_b .

2.2 Modeling different between- and within-subject effects

When the between- and within-subject effects are different it is probably best to model them separately because it avoids averaging the larger effect with the smaller effect, and because the distinction between these two effects can inform our understanding of brain-behavior associations. In this case, increasing between- or within-subject variance independently increases the effect size for the between- or within-subject parameter when (9) is the correct model. The parameters can be interpreted separately as trait versus state associations with the outcome. Given this, as one might expect, functional connectivity measures had larger average state associations than structural measures did in the main paper (Fig. S11). A LMM or GEE can be used to model and estimate the parameters separately.

Various parameterizations are possible. They look similar, but differ in their interpretation and mathematical formulation. For example, an alternative to (9) models a separate effect for the first timepoint that influences all

future measurements.

$$Y_{ij} = \beta_0^* + \beta_b^* x_{i1} + \beta_w^* (x_{ij} - x_{i1}) + \epsilon_{ij}, \quad (13)$$

which might make sense if the first measurement in the study represents a particular epoch that is comparable across participants.

2.2.1 Interpretation of between- and within-subject effects

The interpretation of the between- and within-subject parameters will be dependent on the context and the parameterization. The parameters can be better understood by looking at the expected differences between two measurements within (14) and between (15) participants,

$$\mathbb{E}(Y_{ij} - Y_{ik}) = \beta_w (x_{ij} - x_{ik}) \quad (14)$$

$$\mathbb{E}(Y_{ij} - Y_{\ell k}) = \beta_b (\bar{x}_i - \bar{x}_k) + \beta_w \{(x_{ij} - \bar{x}_i) - (x_{\ell k} - \bar{x}_\ell)\}. \quad (15)$$

Comparing measurements for two values of the covariate within a subject are expected to have a different effect than comparing the exact same two covariate values between subjects. From (14), we can see that if we plot $(Y_{ij} - Y_{ik})$ versus $(x_{ij} - x_{ik})$, the slope captures only the within-subject effect, which can be used for visually evaluating the within-subject effect (see e.g., Fig. S12 in the main paper). This can occur if the covariate has high measurement variability, but the outcome variable does not, then we would expect within-subject change in the covariate to be unrelated to within-subject changes in the outcome ($\beta_w = 0$). For psychometric measurements that can vary randomly each time they are measured, \bar{x}_i in (9) represents "trait" value of the covariate and the term $(x_{ij} - \bar{x}_i)$ is the deviation of measurement j from this trait mean. If the brain outcome measure (e.g. GMV) is expected not to vary by state, then it might be reasonable to believe that $\beta_w \approx 0$, whereas with FC data, it's likely that within-subject association varies across regions as the measurements may be differentially affected by state-related changes.

For certain covariates, it does not make sense to consider different between- and within-subject effects, depending on the study design. For example, consider the parameterization in (13) for the GMV-age association in an observational study, it would be odd if the age that a participant entered the study had an additional effect on GMV above the effect of their age when the GMV measurement was taken. In this parameterization, the absence of this

first time point effect implies $\beta_b^* = 0$, or in the parameterization (9), $\beta_b = \beta_w$. Alternatively, if study entry represented the occurrence of a particular event, e.g., diagnosis, then it seems plausible that $\beta_b^* \neq 0$.

If there are potential nonlinearities then each effect may be estimated with multiple degrees of freedom, e.g., by using splines. The interpretation of parameter estimates from model (9) will not be comparable to the estimate from a cross-sectional model because the cross-sectional model cannot differentiate the two effects and is estimating the combination. If the goal is to obtain an estimate that is comparable to the one obtained in a cross-sectional study, then see Section 2.3.

2.3 Modeling the averaged parameter

If the true model has different between- and within-subject parameters, despite that it is incorrect, model (10) can still be used. In this case, we recommend targeting the parameter obtained from a cross-sectional model with one measurement per participant (the weighted average of potential between- and within-subject parameters),

$$Y_{i1} = \beta_0 + \beta x_{i1} + \epsilon_{i1}. \quad (16)$$

When model (16) is fit, but (9) is true, we can use formula (11) to assess the bias of the estimators under different models (Table 1). As can be seen from Table 1, only the GEE with independence working covariance structure targets the same parameter as the cross-sectional model. The commonly used LMM or GEE with another covariance structure is biased for this parameter, where the bias depends on the structure of the working covariance matrix. In the case of LMM, the working covariance matrix is assumed to be the true covariance. Thus, in this setting, only GEE with independence working covariance can be used to estimate β in (16) in a longitudinal model. To increase ES without biasing the parameter estimator between- and within-subject variance must be increased proportionally. This is because the weights for the between- and within-subject effects in $\mathbb{E}\hat{\beta}$ are determined by $\rho_x = \frac{\sigma_b^2}{\sigma_b^2 + \sigma_w^2}$ (as defined in Section 2.4); increasing σ_b^2 or σ_w^2 independently will change the value of the parameter. Thus, a sampling scheme that increases both proportionally must be adopted.

The longitudinal model is less efficient per scan than the cross-sectional model under mild assumptions that are likely to hold in BWAS. To see this,

we compare the ratio of the variances of the two estimators. Assuming that in each study design we collect Nm total brain measurements, then the ratio of the variance of the longitudinal GEE with independence working covariance to the cross-sectional model, (16), is

$$\begin{aligned} \frac{\text{var}(\hat{\beta}_{\text{GEE}})}{\text{var}(\hat{\beta}_{\text{CS}})} &= \frac{\sum_{i=1}^N \left(\sum_{j=1}^m \sigma_y^2 X_{ij}^2 + \sum_{j \neq k}^m \rho \sigma_y^2 X_{ij} X_{ik} \right)}{\left(\sum_{i,j} X_{ij}^2 \right)^2} \times \frac{\sum_i^{Nm} X_{i1}^2}{\sigma_y^2} \\ &\geq \frac{\sum_{i,j} X_{ij}^2 + \sum_{i=1}^N \sum_{j \neq k}^m \rho X_{ij} X_{ik}}{\sum_{i,j} X_{ij}^2} \geq 1, \end{aligned}$$

where the first inequality assumes that the behavioral measurements between subjects are on average larger than measurements within subjects, $\sum_i^{Nm} X_{i1}^2 \geq \sum_{i,j} X_{ij}^2$ and the second inequality assumes that brain and behavioral repeated measurements are each positively correlated within-subject. Practically, this analysis indicates that when targeting an averaged between- and within-subject effect (row 1 of Table 1), it is better to perform a cross-sectional study rather than a longitudinal study. Note, however, collecting more longitudinal measurements in the GEE will have better efficiency than a cross-sectional study with the same number of independent participants. Thus, longitudinal data should be used in the GEE framework if it is available.

Model	Parameters	$\mathbb{E}\hat{\beta}$
Cross-sectional study	$\rho = 0, m = 1$	$\mathbb{E}\hat{\beta} = \beta_b \rho_x + \beta_w (1 - \rho_x)$
GEE independence work covariance	$\rho = 0, m > 1$	$\mathbb{E}\hat{\beta} = \beta_b \rho_x + \beta_w (1 - \rho_x)$
LMM/GEE exchangeable covariance	$\rho \neq 0, m > 1$	Equation (11)

Table 1: Expected values of the parameter estimates from (16), or (10) under different working covariances. If the goal is to estimate the same parameter as the cross-sectional model (16), then LMM/GEE with non-independence covariance structure are biased, because the weights for β_b and β_w are as given in (11). The only longitudinal model that is unbiased for the cross-sectional effect is the GEE with independence working covariance (second row). $\rho_x^2 = \frac{\sigma_x^2}{\sigma^2}$ as defined in Section 2.4.

R 1.4: Oversampling the tails of a distribution is a popular approach, for example in case-control GWAS, and this study argues in its favor more broadly since it boosted effect sizes for age and the six cognitive variables evaluated. Several generic concerns are sometimes raised as potential drawbacks when discussing the benefits of tail oversampling, at least in its more extreme forms. One previously raised concern is that some brain/behavioral phenotype relationships, or a subset of features, may be non-linear, including u- or inverted u-shaped. Could under-sampling the middle of a distribution increase power at the cost of distorting the underlying relationships? For example, if using multivariate methods to predict a disease phenotype, the prediction could be biased towards features that best separate the most severe disease state, at the expense of being able to categorize intermediate severities. In addition, some work has shown that dichotomizing variables increases the risk for confounding (Austin, Brunner Stat. Med., 2004), might the same be true for oversampling the tails?

You are correct, that when there is a U (or inverted U) shape in the covariate outcome relationship, then it will decrease power. Because we recommend sampling schemes that reduce weight in the middle, the estimates here will be unbiased assuming the model is correctly specified, but will be

estimated with lower confidence. This means that a study will be less likely to detect nonlinearities in the middle of the sample, but on average there would be no loss of accuracy of the estimated curves. We've added this point to the discussion (*Increasing between-subject variability to increase effect sizes*):

We show that increasing a variable's SD improves ESs by a median factor of 1.4, even when there is nonlinearity in the association, such as with age and GMV (Fig. S1). When the association is very non-monotonic, e.g., if there is a U-shape relationship between covariate and outcome, sampling the tails more heavily could decrease power, and it also may decrease our ability to detect nonlinearities in the center of the study population. However, increasing between-subject variability was beneficial in more than 72% of the association pairs we studied, despite potential nonlinearities.

We focused on non-multivariate/machine learning models here, but we think you are correct, if there was a particular region that had greater sensitivity to changes in the middle range of a psychometric variable, then sampling the tails would decrease the predictive value of that variable. This kind of curve would look flat at the left and right sides, but be very steep in the middle. Ideally for this type of unusual relationship, a high-powered sampling procedure would sample at the steepest parts of the association curve, but it would require knowing the shape of the curve in advance, which is often not possible.

Sampling the tails is distinct from dichotomization, because dichotomization forces the values of the covariates to be one of two values. Sampling the tails actually tries to obtain measurements that are as different as possible, rather than artificially making them appear different. We attempt to unbiasedly estimate the association between covariate and outcome in their original measurement scales by estimating nonlinear functions.

R1.5: Despite potential drawbacks of tail oversampling, especially when thinking about multivariate methods, the manuscript clearly provides strong examples in its favor, that in most scenarios should outweigh concerns over misrepresenting the underlying physiology. Therefore, if a study is being designed to investigate a single brain-behavior association (e.g., age & brain volume), or perhaps a narrow set of questions, tail oversampling appears to be the winning approach.

We agree that it is advantageous design option. When there are multiple variables to optimize, multivariate sampling schemes can be used to optimize the design to increase effect sizes (Tao et.al, 2015). We've added a reference to this approach in the Discussion, "When there are multiple covariates of interest, multivariate optimal sampling designs can be used to optimize effect sizes."

Reference:

Tao R, Zeng D, Franceschini N, North KE, Boerwinkle E, Lin DY. Analysis of Sequence Data Under Multivariate Trait-Dependent Sampling. *J Am Stat Assoc.* Taylor & Francis; 2015 Apr 3;110(510):560–572.

R1.6: For massive datasets (e.g., ABCD, UKB) that will be used for myriad different research questions, the question of tail oversampling becomes more complicated. Is it possible to oversample the tails of many/most distributions? Furthermore, oversampling requires confidence that the most important metric is known, measured accurately and neurobiologically meaningful. In the case of age all of those are of course true, but it seems to get more complicated for psychopathological questionnaires such as the CBCL, which seem to have little correlation with brain metrics. The current study shows that oversampling does not boost very small brain-behavior associations for some of the cognitive measures. The BWAS effect size for new phenotypic metrics cannot be known prior to conducting a large sample study, thus oversampling the tails carries the risk of designing a study around metrics without neurobiological correlates. Hence, beyond easily accurately measured biological variables (e.g., age, sex) population sampling might be the safest approach, especially if the large-scale dataset is supposed to serve many research groups if a wide variety of questions of interest. If for example a dataset were to be constructed for BWAS of psychiatric disorders, would it be most prudent to oversample patients with severe symptoms in many dimensions, or would it be better to include patients with extreme anxiety, psychosis, depression, but minimal other symptoms? For the other end of the distribution, are there risks of confounding if the sample is enriched for participants without any psychiatric symptoms, such as higher SES, more likely to be cis-gendered?

It is possible to oversample the tails of many/most distributions. In fact, multivariate outcome-dependent sampling designs are highly cost effective and can substantially increase statistical power when compared to a random sample of the same size (Tao et al., 2015). In the early 2010s when DNA sequencing was still prohibitively expensive, multivariate outcome-dependent sampling designs were adopted in large-scale sequencing studies to identify genetic risk factors for multiple phenotypes/diseases of interest. For example, the National Heart, Lung, and Blood Institute Exome Sequencing Project selected ~7000 subjects for whole exome sequencing from six cohort studies (Lin et al., 2013). This project contains several studies, each of which was focused on a particular trait and some of which selected individuals with extreme high and low values of quantitative traits, including low-density lipoprotein, blood pressure, and body mass index. Another example is the Cohorts for Heart and Aging Research in Genomic Epidemiology Targeted Sequencing Study, in which individuals with extreme values from 14 traits, as well as a random sample of individuals, were selected for targeted sequencing (Lin et al., 2014).

We agree with you that high-quality outcome and covariate data are essential for selecting the most informative subjects for the measurement of expensive brain imaging data, because the “magnitude of efficiency gain” depends on the correlation between the data that are used for

subject selection and the research questions of interest. With that being said, we would also like to point out that sampling designs that rely on imperfect data may still gain efficiency over random sampling and may be worth trying, especially in scenarios where the true effect sizes of interest are expected to be small to moderate (as in BWAS). When such cost-effective outcome-dependent sampling designs are used, one should use specific statistical methods that account for the biased sampling designs in the downstream analysis, as standard methods assuming random sampling are inappropriate and will render biased population estimates (Tao et al., 2015; Tao et al., 2017).

We added this paragraph to the *Increasing between-subject variability to increase effect sizes* Section of the discussion:

Because standardized ESs are directly related to study efficiency, design considerations that optimize study efficiency will increase ESs and study replicability. Two-phase, extreme group, and outcome-dependent sampling designs can inform what subjects should be selected for imaging from a larger sample in order to increase the efficiency and ESs of imaging-behavior associations^{38–44}. When there are multiple covariates of interest, multivariate optimal sampling designs can be used to increase effect sizes⁴⁵. As we see here, use of optimal designs increases power, but positively biases estimated ESs relative to their population values due to manipulation of the sample variance⁴². Weighted regression or other statistical methods can be used to compute unbiased ES estimates that account for sampling design^{45–47}. When there are concerns about implementing optimal design strategies, choosing optimal psychometric measurements or interventions (e.g., medications or neuromodulation) within a clinical trial using cross-over or between group comparisons may also be effective for increasing ESs for a target association. The decision to pursue a particular design will depend on the costs and complexity of acquiring the non-brain variables of interest and translational goals that may require population representative ES estimates.

References:

1. Tao R, Zeng D, Franceschini N, North KE, Boerwinkle E, Lin DY. Analysis of Sequence Data Under Multivariate Trait-Dependent Sampling. *J Am Stat Assoc.* Taylor & Francis; 2015 Apr 3;110(510):560–572.
2. Lin DY, Zeng D, Tang ZZ. Quantitative trait analysis in sequencing studies under trait-dependent sampling. *Proc Natl Acad Sci. Proceedings of the National Academy of Sciences;* 2013 Jul 23;110(30):12247–12252.
3. Lin H, Wang M, Brody JA, Bis JC, Dupuis J, Lumley T, McKnight B, Rice KM, Sitlani CM, Reid JG, Bressler J, Liu X, Davis BC, Johnson AD, O'Donnell CJ, Kovar CL, Dinh H, Wu Y, Newsham I, Chen H, Broka A, DeStefano AL, Gupta M, Lunetta KL, Liu CT, White CC, Xing C, Zhou Y, Benjamin EJ, Schnabel RB, Heckbert SR, Psaty BM, Muzny DM, Cupples LA, Morrison AC, Boerwinkle E. Strategies to Design and Analyze Targeted Sequencing Data. *Circ Cardiovasc Genet.* American Heart Association; 2014 Jun;7(3):335–343.
4. Tao R, Zeng D, Lin DY. Efficient Semiparametric Inference Under Two-Phase Sampling, With Applications to Genetic Association Studies. *J Am Stat Assoc.* 2017;112(520):1468–1476. PMID: PMC5823539

Currently, the manuscript does not discuss the potential benefits of perhaps the most extreme forms of brain-behavior (non-BWAS) effect size boosting: interventions and brain lesions. While the data cannot directly speak to the effect sizes of interventions (e.g., medications, neuromodulation, etc.), it might still be worth adding a sentence about RCTs and other within-patient clinical study designs.

We've added a discussion point about this in the *Increasing between-subject variability to increase effect sizes* Section of the discussion. The excerpt is in the response to your comment directly above.

Minor detail points:

The abstract and the first section of the main manuscript overlap quite a bit. Some of the redundancies could likely be trimmed out.

Thank you. We've tried to remove the redundancies.

Lines 44, 45: Age & sex are not complex behavioral phenotypes and therefore not BWAS variables.

We've removed age and sex from that sentence.

Lines 260-63: Could you give more detail as to what types of 'more complex designs' might be needed to increase effect sizes for cognitive and similar metrics? Are there relevant references?

Yes. We've made substantial edits to this section and removed this statement. The final paragraph of that section "Study design considerations", discusses all the design features. Pasted above in response to R1.6.

Lines 265-66: This sentence highlights the 290% effect size increase going from cross-sectional to longitudinal, without additional specifics. It seems that this value is specific to the association between gray matter volume and age, if so, it would be good to include that, perhaps in parentheses?

We've specified this result more clearly and tried to capture the nuance of the subsequent findings more carefully:

In the meta-analysis, longitudinal studies of the total GMV-age effect have, on average, >380% larger standardized ESs than cross-sectional studies. However, we see in subsequent analyses that the benefit of conducting a longitudinal design is highly dependent on the between- and within-subject associations.

Line 292-93: The comment "compared to the primary focus on the ABCD sample in previous papers" is missing references. The word 'papers' sounds a tad colloquial in this sentence.

We've removed this section in the revision.

The use of the RESI, CS-RESI, and L-RESI terminology is slightly confusing. For example, the subheading: "Cross-sectional RESI (CS-RESI) quantifies benefit of longitudinal design," creates cognitive conflict. Why call something that quantifies the benefit of longitudinal designs the Cross-sectional RESI?

We've changed this to "**Comparison of L-RESI and CS-RESI quantifies benefit of longitudinal design**"

In Table 1 the heading titles of "Outcome: RESI or L-RESI" and "Outcome: RESI or CS-RESI" are also confusing. Which one is? The confusion is increased because Table 2 has a column labeled RESI or CS-RESI and one labeled L-RESI, if viewed in isolation Table 2 would suggest that RESI and CS-RESI are equivalent, but not L-RESI.

We've clarified this in the caption of all the tables (Now Tables S3-S10), "For cross-sectional studies the outcome is RESI, for longitudinal studies the outcome is (A) L-RESI or (B) CS-RESI."

Figure 1 is titled "Partial regression plots of a meta-analysis of RESI and L-RESI for [...]," but all the plot y-axes are labeled RESI.

The L-RESI is shown only for the longitudinal studies, we've noted in the figure caption and changed the y-axis labels to (L-)RESI. Hopefully, this helps to make that clearer.

Could Figure 2a,b be compressed by just showing curves of the ES increase for longitudinal designs, instead showing L-RESI minus RESI and CS-RESI minus RESI?

The L-RESI and CS-RESI are both computed in longitudinal studies. The CS-RESI is the estimated effect size if the study had been conducted cross-sectionally and the figure is showing that the CS-RESI accurately estimates the RESI value computed from a cross-sectional study. We've tried to clear this up in the figure caption.

In Figure 2c-f how are the studies sorted vertically? If they are not already, would it be possible to also sort them by sample size and maybe add a single y-axis that gives the readers an idea of the sample size distributions. Panels c-f suggest that the benefits of longitudinal sampling are less apparent in the more lifespan samples (first 3 rows) in contrast to sample that focused on just younger or just older participants. Could that be because those samples are more likely to capture non-linear relationships?

The studies were sorted by the study design and decreasing standard deviation of age. We now clarified this in the caption of Figure 2. The descriptive summaries of each study (including sample size) are put in Tab. S1. The benefit of longitudinal sampling depends on some factors that aren't shown in the figure (between- and within-subject variability of the covariate, as discussed later in the paper and in the supplement). It's possible the large age range reflects the between-subject variance of age more than the within-subject variance and so the between-subject component dominates the ES estimate, causing the CS-RESI and L-RESI to appear similar. This is just an adhoc interpretation and we refrain from discussing differences between cross-sectional and longitudinal designs until the subsequent section.

In Fig. 3b the 30,000 label on the x-axis is partially cut off.

We've completely revised this figure to make the output more interpretable.

Instead of labeling the sampling schemes 0,1,2, perhaps they could be given succinct descriptive labels (e.g., replace 0 with 'flat'), which would make it quicker to parse the findings. And perhaps the same could be done for the within-subject sampling schemes which are currently also labeled scheme 0,1.

Thank you for the suggestion. We have changed the sampling schemes (Fig. S6 & S8) and now name them by their shapes (e.g., bell-shaped, uniform, and U-shaped).

In Fig. 4, it might be helpful to explicitly say in the figure caption text that the data are from the ABCD. For someone familiar with the different data sets, it can be deduced, but it might be easier for readers, to have it re-stated.

Thanks for the suggestion, We've made major revisions to the figures and explicitly state what dataset(s) was used for the results in each figure.

Referee #2 (Remarks to the Author):

R2.1 This study provides a useful contribution to contemporary discussions about effect sizes in neuroimaging and their implications for sample size requirements, the detectability and reproducibility of findings, and the use of neuroimaging for person-level biomarkers. It is based primarily on analyses relating whole-brain structural measures (gray-matter volume [GMV] and several others) to age and sex, with supplementary analyses of selected cognitive variables. The main findings is that larger variation in the outcome (or phenotype; age and sex) is associated with larger effect sizes (ESs) across studies. In addition, longitudinal designs had substantially larger effect sizes than cross-sectional ones, at least for age and to some degree for sex.

Strengths of the paper include (1) the timeliness of the findings, particularly given widespread confusion about and overgeneralization of previous findings showing small BWAS effect sizes, and (2) its ability to provide a more nuanced view and highlight the importance of some variables, particularly outcome variance, that are not widely appreciated.

Thanks for the summary and positive appraisal of our work.

One conceptual limitation is that the relationship between larger ES and higher predictor variance is a well known feature in basic statistics, based on the mechanics of linear models. At an intuitive level, a study that compares 12 year olds to 50 year olds will have much larger ESs than one that compares 49 to 50 year olds. This principle is covered in basic statistics books but may not be widely appreciated by a neuroimaging audience. However, the basics are not discussed here and perhaps should be (one good book on this is "Data analysis" by Judd, McClelland, and Ryan; I'm sure there are many others with good sections). For example, it should be possible to qualitatively predict the degree of ES increase with extreme sampling (fig 3) and possibly other results. Could the authors characterize the degree to which their findings are explained by simple statistical mechanics of increasing predictor variance in the GLM framework?

You are correct that several of the implications of our findings are known statistical principles. The nuance in real data is due to nonlinearity of the underlying associations and unknown model misspecification, which complicates simple statistical results.

We've tried to highlight this theoretical feature throughout, e.g., in the section **Meta-analyses show ESs depend on study population and design**, we've added "This result is expected in linear models based on the well-known relationship between correlation strength and covariate SD." We've also

added a similar statement in the Discussion under *Increasing between-subject variability to increase effect sizes*.

Our results demonstrate that increasing ESs can be as simple as sampling subjects with small and large values of the variable of interest with higher probability. This is well-known in linear models where the effect size is explicitly a function of the SD of the covariate³⁷. We show that increasing a variable's SD improves ESs by a median factor of 1.4, even when there is nonlinearity in the association, such as with age and GMV (Fig. S1).

In response to your and Reviewer 3's interest, we've added a bit more statistical theory to the Supplement in the context of longitudinal models considering state/trait variance of the covariate and outcome. As usual, this theory is a simplification of real life, but aligns with the results we show using the ABCD. For details, please see response to R2.2 below.

The study does offer more nuance, including (1) derivation and use of robust ES estimates, a strength; (2) discussion of differences between between-person and within-person variance and when longitudinal sampling might be expected to increase ESs and when it won't (i.e., when the relationship of interest is strong within-person, and when there are many sources of between-person noise), (3) Some basic large-sample benchmarking of ESs for age and sex, as a reference point, and (4) Some cases where longitudinal sampling isn't helpful, which could provide clues about when a longitudinal study is important (it is for age, but it's less clear what the benefit is for other variables).

Thanks for these positive comments.

Some limitations and things to improve related to these might be: (1) Provide clear mapping from RESI ESs onto power and sample size requirements; (2) Provide more concrete examples and perhaps a mathematical account of between- and within-person error variance and how it relates to the ES benefit with longitudinal designs (can this be mapped onto the data here to explain the pattern of findings better?); (3) Consider a wider range of variables and the implications of the restricted focus on age, at least in discussion.

I elaborate on some of these briefly below.

We respond to these comments item-by-item in your enumeration below.

R2.1. Can you translate the main results into concrete benefit in power and sample size required for 80% power, for broad audience? This may help readers familiar with Cohen's d to understand the implications of a RESI of, e.g., 0.1. What sample sizes are required to detect effects under different conditions studied? (See, e.g., Spisak 2023 Nature for some interpretable plots). How do these compare with effect sizes in, e.g., Marek 2022, Nature?

We now include the effect of resampling on effect sizes, power, and sample size needed to obtain 80% power (Figures 3-6; pasted below for convenience). To avoid conflating "small" and "large" effect sizes across regional GMV and CT, we draw separate mean curves for effect sizes above and below the 25 percentile (solid versus dashed lines).

Fig. 3. Improved effect sizes and statistical power for age associations with different brain measures: total GMV (a-b), regional GMV (c-d), and regional CT (e-f), under three sampling schemes in the UKB study (N = 29,031 for total GMV and N = 29,030 for regional GMV and CT). The sampling schemes target different age distributions to increase the variance of age: Bell-shaped<Uniform<U-shaped (Fig. S6). Using the resampling schemes that increase age variance increases (a) the total GMV-age effect sizes, and (b) statistical power (at significance level of 0.05). The curves represent the average effect size or estimated power at a given sample size and sampling scheme. The shaded areas represent the corresponding 95% confidence bands. (c-f) The bold or dashed curves are the average effect size or power across regions with effect sizes above (solid lines) or below (dashed lines) the 25th percentile for a given sample size and sampling scheme.

Fig. 4. Improved effect sizes and statistical power for age associations with structural brain measures under different longitudinal sampling schemes in ADNI data. Three different sampling schemes (Fig. S8) are

implemented to modify the distributions of baseline age and the age change from baseline and therefore to modify the between- and within-subject variability of age. **(a)** Implementing the sampling schemes results in higher (between- and/or within-subject) variability and increases the total GMV-age effect size, and **(b)** statistical power at significance level of 0.05. The curves represent the average effect size or estimated power and the gray areas are the 95% confidence bands. **(c-d)** Increasing the number of measurements from one to two per subject provides the most benefit on **(c)** the total GMV-age effect size and **(d)** statistical power when using uniform between- and within-subject sampling schemes and $N = 30$. The points represent the mean effect size or estimated power and the whiskers are the 95% confidence intervals. **(e-h)** Improved effect sizes and power for age associations with regional **(e-f)** GMV and **(g-h)** CT, across all brain regions under different sampling schemes. The bold curves are the average effect size or estimated power across all regions with effect sizes above (solid lines) or below (dashed lines) the 25th percentile for a given sample size and sampling scheme. The gray areas are the corresponding 95% confidence bands. Increasing the between- and within-subject variability of age by implementing different sampling schemes can improve the effect size of age, and the associated statistical power, on regional GMV and regional CT.

Fig. 5. Heterogeneous improvement of effect sizes for cognitive, mental health, and demographic associations with structural and functional brain measures in the ABCD study at N = 500. (a) U-shaped between-subject sampling scheme (blue) that increases between-subject variability of the non-brain covariate produces larger effect sizes and (b) reduces the number of participants scanned to obtain 80% power in total GMV. The points and triangles are the average effect sizes across bootstraps and the whiskers are the 95% confidence intervals. Increasing within-subject sampling (triangles) is sometimes detrimental (i.e., reduces effect sizes). A similar pattern holds in (c-d) regional GMV and (e-f) regional CT. In contrast, (g) regional pairwise FC (functional connectivity) effect sizes increase by increasing between- (blue) and within-subject variability (dashed borders) with a corresponding reduction in the (h) number of participants scanned for 80% power.

Fig. 6. Longitudinal study designs can be detrimental to effect sizes and replicability due to heterogeneous between- and within-subject associations between brain and behavior measures. (a-c) Cross-sectional analyses (indicated by “1”s on the x-axes) can have larger effect sizes than the same longitudinal analyses (indicated by “2” or “2+”) for structural brain measures in ABCD. **(d)** The FC measures have a slight benefit of longitudinal modeling. The cross-sectional analyses only use the baseline measures; the longitudinal analyses use the full longitudinal data. **(e)** Most regional GMV associations (Fig. 5c) have larger between-subject parameter estimates ($\hat{\beta}_b$, x-axis) than within-subject parameter estimates ($\hat{\beta}_w$, y-axis; see Supplement: Eqn (13)), whereas **(f)** FC associations (Fig. 5g) show more heterogeneous relationships between the two parameters.

R2.2. The findings illustrate statistical principles in a very large sample, but with a very limited number of brain variables and phenotypic predictors. At a statistical level, it stops short of providing a principled and quantitative account of when precisely effect sizes should be larger as a function of within- and between-person predictor variance. This type of contribution would be expected from a statistical paper of this type, and would enhance the value and generalizability of the paper. Analysis by, e.g., Desmond and Glover actually goes farther in this respect, and the mathematical foundations of variance components in multi-level settings could be gainfully employed here.

We've expanded substantially on the statistical theory which helps to guide when ESs are expected to be larger in longitudinal studies. A detailed description is given here, which is recapitulated more briefly in response to R3.3 below. Your suggestion (and R3.3) helped us to more accurately understand and describe the observed detrimental effects of increasing within-subject variance and we've added to the section, ***Preferred sampling schemes and longitudinal designs depend on state versus trait associations between brain and non-brain measures***, to describe why this occurs. Figures 6 and S12 are included below for convenience.

To investigate why increasing within-subject variance or using longitudinal designs is not beneficial for some associations, we examined an assumption common to widely used LMMs and GEEs in BWAS: that there is consistent association strength of the brain and non-brain measure across between- and within-subject changes in the non-brain measurement. These effects are independent of the random effects parameters and can differ if there are different state or trait associations between the brain and non-brain measure. For example, measures of Crystallized Intelligence are subject to within-individual differences in state more than total GMV, which has low state variance so there is no within-subject association between these variables. In contrast, FC measures are also subject to within-individual differences in state so can have stronger within-subject associations with Crystallized Intelligence. To demonstrate this, we fit models that estimate distinct effects for between- and within-subject associations in ABCD and find that there are large between-subject parameter estimates and small within-subject parameter estimates in total and regional GMV (Tab. S13; Fig 6e), whereas the FC associations are distributed more evenly across between- and within-subject parameters (Fig. 6f). If the between- and within-subject associations are different, fitting these associations separately avoids averaging the larger effect with the smaller effect and can inform our understanding of brain-behavior associations. This approach ameliorates the detrimental effect of the longitudinal design we saw with structural brain measures in the ABCD (Fig. S11). See Supplemental Section 2 for technical details.

Fig. 6. Longitudinal study designs can be detrimental to effect sizes and replicability due to heterogeneous between- and within-subject associations between brain and behavior measures. (a-c) Cross-sectional analyses (indicated by “1”s on the x-axes) can have larger effect sizes than the same longitudinal analyses (indicated by “2” or “2+”) for structural brain measures in ABCD. **(d)** The FC measures have a slight benefit of longitudinal modeling. The cross-sectional analyses only use the baseline measures; the longitudinal analyses use the full longitudinal data. **(e)** Most regional GMV associations (Fig. 5c) have larger between-subject parameter estimates ($\hat{\beta}_b$, x-axis) than within-subject parameter estimates ($\hat{\beta}_w$, y-axis; see Supplement: Eqn (13)), whereas **(f)** FC associations (Fig. 5g) show more heterogeneous relationships between the two parameters.

Fig. S11. The influence of sampling schemes on the effect sizes for between- and within-subject associations, respectively, of cognition, mental health, and demographic covariates with different brain measures in the ABCD study at N = 500. Between-subject effect sizes are predominantly affected by the between-subject variance, whereas within-subject effect sizes are predominantly affected by the within-subject variance. The results for covariates birthweight and handedness, which do not vary within subjects, are not included as the within-subject sampling schemes are not applied to them.

We discuss the modeling strategies in the Discussion, *Longitudinal design considerations*,

Longitudinal models offer the unique ability to separately estimate between- and within-subject associations. When estimating these effects separately, investigators can use

a GEE or LMM with any working covariance structure which will yield unbiased estimates of the between- and within-subject effects if the model is correctly specified. Using sampling strategies to increase between- and within-subject variance will separately increase ESs for the between- and within-subject associations, respectively. Estimates of between- and within-subject associations will not be directly comparable to estimates obtained from a cross-sectional model because they have a different interpretation (See Supplement: Section 2.2)^{49–51}. We discuss several other reasonable modeling approaches for longitudinal BWAS in the Supplement (Section 2).

We also provide more detail in the Supplement (Sections 2.1-2.3). Briefly, we propose three options for modeling longitudinal associations in BWAS. 1) If the between- and within-subject associations are expected to be equal, then the usual linear mixed model (LMM) or GEE will unbiasedly estimate the parameter for the association and sampling schemes that increase the between- and within-subject variance will increase effect sizes for this parameter. 2) When the between- and within-subject effects are different it is probably best to model them separately because it avoids averaging the larger effect with the smaller effect, and because the distinction between these two effects can inform our understanding of brain-behavior associations. In this case, increasing between- or within-subject variance independently increases the effect size for the between- or within-subject parameter. 3) . If between- and within-subject effects are likely to differ, but the goal is to obtain an estimate that is comparable to the one obtained in a cross-sectional study, then a GEE must be used with an independence working covariance structure to obtain unbiased estimates for this parameter (Supplementary Section 2.3).

Finally, with regard to the variance components, we briefly mention them in that excerpt, but the feature driving the difference in cross-sectional and longitudinal designs is due to the fixed effects, not the random effects structure.

R2.3. In the introduction, the authors may want to consider a larger conceptual space of variables that moderate BWAS effect sizes, including the reliability of both phenotypic (outcomes) and brain measures; the univariate vs. multivariate nature of the relationship; and whether phenotypes are more or less likely to relate to type types of brain measures studied (e.g., Spisak et al. 2023 Nature; Rosenberg and Finn 2022 Nat Neuroscience). Here, the point is well taken that effect sizes in Marek et al. reflect a mix of null and non-null phenotypes, but the generalizability of the claims here are limited by a focus on only a few global brain measures (e.g., GMV) and sex/age, which are among the very few phenotypes expected to be related to these brain measures. The effect sizes here may thus represent an upper bound. Age and sex have (1) high phenotypic reliability (low-noise measures), (2) relatively high reliability and stable genetic encoding of brain structure (e.g., Elliot 2022 Nature, Duff et al. 2022), and (3) likely stronger true relationships between brain and phenotype than many other phenotypic measures.

Thus, the paper makes the basic point it intends to make in a large sample, but it's hard to see when the conclusions will generalize more broadly and when they won't.

In response to your and Reviewer 1's comments we've added more outcomes (regional GMV, global and regional cortical thickness, and regional functional connectivity from the Gordon atlas [only available longitudinally in ABCD]) and covariates (in the ABCD, in addition to NIH toolbox, we added CBCL, handedness, birthweight, and BMI). These different covariate-outcome pairs highlight important features of when increasing within-subject variance increases effect sizes (as described in response to comment R2.2). To summarize, the outcome variables with high reliability such as GMV do not benefit from increasing within-subject variance when the covariate has low reliability, such as with psychometric measures from the NIH toolbox measures or the CBCL, because GMV is not strongly associated with state-related changes. In contrast, FC, which we expect to have state-related variability, is associated with within-subject variation in these psychometric measures. This interpretation is detailed in the section ***Preferred sampling schemes and longitudinal designs depend on state versus trait associations between brain and non-brain measures*** (partially copied above in response to R2.2) and in the Discussion section *Study design considerations*.

In the meta-analysis, longitudinal studies of the total GMV-age effect have, on average, >380% larger standardized ESs than cross-sectional studies. However, we see in subsequent analyses that the benefit of conducting a longitudinal design is highly dependent on the between- and within-subject associations. When the between- and within-subject parameters are equal, longitudinal studies offer larger ESs⁴⁸. This is the reason there is a benefit of longitudinal design in the ADNI in the total GMV-age association (Fig. S12). The apparent lack of benefit in the GMV-psychometric measurement association in the ABCD is due to a lack of association between within-subject changes in the psychometric measurements with within-subject changes in GMV (Fig. 6e; Tab. S13). The detrimental effect of the longitudinal design is not due to a reduction in efficiency, but bias in the parameter estimate due to the typical longitudinal model in BWAS conflating between- and within-subject associations.

Here's one more specific example. Longitudinal designs that eliminate between-subject error variance are helpful in some cases, but not others (some of the cognitive variables). Is this because more data per person reduces error variance and increases power, in accordance with the central limit theorem? And/or are cognitive effects not associated because they are unreliable, vary over time between assessments, and/or are simply unrelated to global GMV?

We've expanded the discussion on this and tied it into the statistical theory. See response to comment R2.2 for details. The lack of benefit for the GMV-cognitive variable associations is due to

within-subject variation in the covariate being unrelated to within-subject variation in GMV. Because the model is assuming between- and within-subject changes are the same, it creates bias in the parameter estimate by creating a weighted average of the between- and within-subject effects. This can reduce power in some cases, by heavily weighting the within-subject effect, which is small. Modeling the two effects separately conserves power for the larger effect and allows the between- and within-subject sampling schemes to more accurately target the between- and within-subject effects, respectively.

The findings here may be specifically related to age as well, and may not generalize. For example, the paper does not consider time between assessments, which is key for longitudinal studies because longer times increase variance related to slowly changing processes like age (increasing ESs), but decrease ESs for variables that may fluctuate more rapidly (e.g., cognitive outcomes? e.g., with season, stressors, etc.). Considering effect size per year, and per year during development or during GMV decline with aging, may help in part. But: will the findings on when longitudinal designs are helpful or not generalize beyond age and the few variables tested here?

As discussed, we've expanded this discussion by adding regional measures of GMV, CT, and FC and their relationship with psychometric variables in the ABCD. Indeed, the findings from age variable do not generalize to all brain-behavior associations. The difference in longitudinal design is due to conflation of between- and within-subject effects and the resulting parameter estimate (Supplementary Material Eqn. 11) averages the two effects. The weights for the average depend on the within-subject correlation of the outcome variable and covariate. We describe this as bias in the parameter estimate, which can be removed/reduced by modeling the between- and within-subject effects separately (Fig S11). This is described in details in ***Preferred sampling schemes and longitudinal designs depend on state versus trait associations between brain and non-brain measures***. When modeling the between- and within-subject associations separately there is less bleed over between manipulations of the between- and within-subject effects,

If the between- and within-subject associations are different, fitting these associations separately avoids averaging the larger effect with the smaller effect and can inform our understanding of brain-behavior associations. This approach ameliorates the detrimental effect of the longitudinal design we saw with structural brain measures in the ABCD (Fig. S11). See Supplemental Section 2 for technical details.

Fig. S11. The influence of sampling schemes on the effect sizes for between- and within-subject associations, respectively, of cognition, mental health, and demographic covariates with different brain measures in the ABCD study at N = 500. Between-subject effect sizes are predominantly affected by the between-subject variance, whereas within-subject effect sizes are predominantly affected by the within-subject variance. The results for covariates birthweight and handedness, which do not vary within subjects, are not included as the within-subject sampling schemes are not applied to them.

The authors conclude: “longitudinal studies have larger standardized ESs than cross-sectional studies”. This broad conclusion may not be warranted, and may apply rather specific to age. The

intuitive reason would be that gray matter changes a lot across time, so within-person relationships between age and GMV may be particularly meaningful. Missing is an assessment of whether cognitive variables also have larger ESs in longitudinal studies — e.g., something like Fig 1A for cognitive outcomes.

We've changed this and the heterogeneity of the effect in longitudinal studies is now a highlight of the Discussion under *Longitudinal design considerations*.

In the meta-analysis, longitudinal studies of the total GMV-age effect have, on average, >380% larger standardized ESs than cross-sectional studies. However, we see in subsequent analyses that the benefit of conducting a longitudinal design is highly dependent on the between- and within-subject associations. When the between- and within-subject parameters are equal, longitudinal studies offer larger ESs⁴⁸. This is the reason there is a benefit of longitudinal design in the ADNI in the total GMV-age association (Fig. S12). The apparent lack of benefit in the GMV-psychometric measurement association in the ABCD is due to a lack of association between within-subject changes in the psychometric measurements with within-subject changes in GMV (Fig. 6e; Tab. S13). The detrimental effect of the longitudinal design is not due to a reduction in efficiency, but bias in the parameter estimate due to the typical longitudinal model in BWAS conflating between- and within-subject associations.

Minor comments

“increasing the within-subject SD of the cognitive measurements did not increase CS-RESI or L-RESI”

*Does this imply that brain structure is not related to within-person variation in cognitive scores?
Could this be because the cognitive scores are not reliable, or because brain structure does not vary over time as a function of score?*

Yes, instead of describing this as reliability, we describe it as state/trait variability because, as we see by adding the FC data, the state variability in the cognitive scores is related to the state variability in FC. We avoid the term “reliability” because low reliability implies this is noise, but because this variability is associated with FC, it is not appropriate to call it noise. We provide a short example discussing this in the third paragraph of ***Preferred sampling schemes and longitudinal designs depend on state versus trait associations between brain and non-brain measures:***

For example, measures of Crystallized Intelligence are subject to within-individual differences in state more than total GMV, which has low state variance so there is no within-subject association between these variables (Tab. S12). In contrast, FC measures are also subject to within-individual differences in state so can have stronger within-subject associations with Crystallized Intelligence.

To increase generalizability, the authors might also consider which kinds of outcomes are likely to have large ESs. An interesting result is that composite cognitive measures, which presumably average over distinct kinds of task-specific measures (e.g., the rationale behind factor analysis in traditional psychometrics), may have larger ESs in Fig 4, and some measures that are notoriously unreliable because they are based on reaction time difference measures (e.g., Flanker) have ESs near zero.

We’ve added a sentence to mention that choosing a good psychometric measurement is important to optimize effect sizes, “When there are concerns about implementing optimal design strategies, choosing optimal psychometric measurements or interventions (e.g. medications or neuromodulation) within a clinical trial using cross-over or between group comparisons may also be effective for increasing ESs for a target association.”

“adding a second longitudinal measurement per subject” helps. Because more data are available, or because of a within-subject effect? And what is the time lag?

We are simply describing the result of the bootstrap analysis, here, without inferring the cause of the effect. Technically, the effect size increases at a rate of the square root of the number of measurements (Supplementary Material Eqn. 12), but there other factors contributing, such as that the between- and within-subject effects are approximately equal for age (Fig S13).

Fig 2: Minor questions: Effect sizes are for which brain measure (GMV only)? Why were these two selected for display? Not because they were the largest, as ABCD is quite a bit larger.

Sorry for the confusion. The Fig. 2a-b show the comparability of the CS-RESI to the RESI computed from a cross-sectional study using the effect size for GMV. We now clarified this in the caption of Fig. 2. We selected this two studies (i.e., ADNI and GUSTO) simply because these are two relatively large longitudinal studies, with multiple measurements per subject. The results for all the studies were included as a supplementary figure (Fig. S3).

Fig 2A: Why the small-sample bias towards larger effect sizes? Suggests some kind of model fitting (i.e. overfitting), which I don't think is the case. Is it that RESI is downwardly biased (toward 0) for small samples (e.g., RESI in Fig 4B is smaller for small samples).

According to the results of simulation studies evaluating the bias of RESI estimator in our previous work (<https://link.springer.com/article/10.1007/s11336-022-09899-x>; Fig 2) and L-RESI estimator in the Supplement (Fig. 2 in Supplement), when the true effect size is non-zero the RESI estimator is slightly negatively biased whereas the L-RESI estimator is slightly positively biased in small samples. This is the reason why the difference (=L-RESI - RESI) are positively biased when sample size is small.

Longitudinal designs show larger effects. Calling it a "bias" (e.g., p. 4) may be misleading as there are fewer sources of error variance and more data in longitudinal analyses, reducing error variance and increasing ES. May conflate the statistical properties of averaging more data and assessing within-person effects with biological properties that total GMV may vary within-person with age (during development and aging in particular) ... masking....

But more scans in ADNI don't help much (fig 3E), so it's not likely averaging over noise, but rather avoiding between-person confounds and nuisance variables.

We refer to this now as a systematic difference (instead of bias) throughout. We now emphasize the conflation of between- and within-subject effects in the paper.

Were cognitive measures always collected at the same time as imaging measures? The former may vary as a function of state (within-person), which might explain the lack of advantage of longitudinal designs if they are not collected at the same time. Is it possible that if cognition and brain were collected simultaneously the picture could change? Maybe this is addressed by fig S8?)

To our knowledge, behavioral measurements in the ABCD were collected on the same day (the collection dates are listed as the same in the dataset; please see <https://nda.nih.gov/abcd/>).

*“for the GMV-age association ESs were largest in young and late adulthood when age-related changes in GMV are strongest”. I’m not sure this is the case, as the “Brain scales” 2022 Nature paper shows the most rapid change around 1 year, and little change in early adulthood.
<https://www.nature.com/articles/s41586-022-04554-y/figures/1>. What was the coverage of early life in the present subsample?*

The age range in the data we used is 2.8 to 100 years, which is a subset of the data used by Seidletz & Bethlehem et al. 2022. Demographic details of each study sample are in Tables S1 and S2. Importantly, a large ES does not necessarily relate exactly to fastest change in GMV because it depends on the precision of accuracy of the slope as well, which can depend on whether there are other unmodeled factors contributing to GMV in a given epoch.

“our focus on age effects and broad cerebral tissue classes is likely to find stronger ESs”. From a statistical averaging perspective yes, but not necessarily, as previous studies have shown that some brain areas are particularly sensitive. I agree that Marek et al. and some other studies average over null and non-null findings (an important point!) It may be that if the most relevant brain measures were used, particularly with predictive models (e.g. Marek 2022’s and Spisak 2023’s multivariate models) the effects could be substantially larger still. This may warrant discussion.

We’ve removed this section from the discussion to devote more space to interpreting the heterogeneity of benefit of longitudinal designs (and within-subject sampling). In the new figures, we plot average curves for regional brain variables separating by the percentile of the effect size or power (Fig 4, e & f, for example), to avoid conflating small and large effects, but we no longer focus on this as part of the discussion.

“guarantee larger ESs when the within-subject correlation of the outcome and the within-subject correlation of the independent variable are both positive”. This is confusing — should one be “between”?

We've removed this from the body of the paper and restated it in the Supplementary Material Section 2.1 in the context of the theory outlined in Supplementary Section 2, "When $\beta_b = \beta_w = \beta$ then there is no model misspecification and increasing the between- or within-subject variance of the covariate increases effect sizes for the parameter β . In this case, the parameters β_b and β_w can be distributed out of the sum in (12) and the whole effect size increases linearly with respect σ_w and σ_b ."

I'm familiar with applying ComBat on whole image data before extracting measures of interest, but here it seems that GMV and other measures were first extracted using FreeSurfer, and then ComBat was applied before analysis. It's not clear to me how ComBat isn't redundant with the linear model that adjusts for batch effects in this case. Perhaps it has to do with the multiplicative error model? The error model was not specified in the methods I believe.

More detail on knot point selection, both in terms of number and position, would be helpful in the Methods.

ComBat can be applied to regional or image-level data (see for example Fortin et al. 2018). It is similar to applying a linear model for adjustment except, as you noted, it also adjusts for differences in the scale of the error variance. This could possibly be done in the context of the GEE, as well, but we chose to use this standard analysis pipeline so as not to create additional nuances to the analysis.

Fortin JP, Cullen N, Sheline YI, Taylor WD, Aselcioglu I, Cook PA, Adams P, Cooper C, Fava M, McGrath PJ, McInnis M, Phillips ML, Trivedi MH, Weissman MM, Shinohara RT. Harmonization of cortical thickness measurements across scanners and sites. *NeuroImage*. 2018 Feb 15;167:104–120.

How were datasets acquired and where are they available, if they are publicly available? Which are available from which sources? The data availability statement is vague.

The data are obtained from the various consortia listed in the author list and acknowledgements. Details of acquisition are available at their respective websites. We've added links to the publicly available data in the Data availability section.

sGMV appears to be partially redundant with GMV. Is that the case, and if so, why are they treated as separate outcomes? Why not separate into cortical GMV and subcortical GMV (sGMV)?

We did this to be consistent with the Brain Charts paper. We now also include analyses of all regions separately.

What does it mean to show a negative effect size in this context?

The negative expected effect sizes in some partial regression plots in Fig. 1, these plots show the expected values of the effect size after fixing other covariates at certain constant levels. The scatter points were also adjusted using the estimated residuals and the expected values. Negative values are possible because we did not transform the effect sizes before modeling to aid interpretability. The relative change of the expected outcome (i.e., effect size of age here) over the x-axis is the primary feature we want to highlight.

Referee #3 (Remarks to the Author):

The authors examine the role of non-brain variable distribution in the consideration of power for brain/non-brain associations. Using a robust effect size measure they've developed, they conduct a number of investigations to explore effect size and its relationship to experimental design. In the Life Brain Chart Consortium, they find that a study's mean age and age standard deviation both impact effect size (but not age skew), and longitudinal studies have increased effect size. Using resampling analyses with ADNI, they confirmed the impact of longitudinal analyses, but found reducing number of repeated observations didn't reduce power as long as within SD of age was maintained. With ABCD, however, longitudinal effect sizes for brain-behavior associations were not stronger than cross-sectional ones.

Major issues

*R3.1 While this is a useful contribution I find it find its impact limited as many of the conclusions are tenants of experimental design: Power of a regression increases with linear effect or independent variable variance. (Maybe simplest demonstration of this is with simple linear regression, where $\beta = r_{XY} * SD(Y) / SD(X)$, or in terms of correlation, $r_{XY} = \beta * SD(X) / SD(Y)$. Thus for a given dependent variable Y and linear slope, correlation and thus power increases with SD(X).) Figure 1 column 2 is a direct result of the impact of SD(X) (some studies have narrow age range, others wider). The careful exposition of these results is not without value, but I don't see the general-audience appeal of the findings.*

You are correct and we emphasize this now in the results and discussion; in the third paragraph of the meta-analysis section, "This result is expected in linear models based on the well-known relationship between correlation strength and covariate SD." and in the Discussion subsection *Increasing between-subject variability to increase effect sizes*, "This is well-known in linear models where the effect size is explicitly a function of the SD of the covariate¹⁰."

In general, the relationship between covariate variability and effect sizes are only guaranteed under linearity so an empirical investigation is warranted. One of our primary contributions is the investigation into the effect of longitudinal models. As we show in the revision, for longitudinal models the effect of modifying within-subject variance can be heterogeneous across pairs of brain and non-brain variables (Figures 5 and 6), where increasing within-subject variability of the covariate and implementing a longitudinal design are detrimental to effect sizes for behavioral associations with GMV and cortical thickness. We derive formulas for the expected value of the coefficient and effect size when there are different between- and within-subject effects (Supplementary Material Eqn. 11 and 12). This allows us to describe the effect it has on increasing within-subject variability or conducting a longitudinal design and offer recommendations to preserve power in longitudinal studies. See below for more detailed responses to your concerns. The revision

of the paper discusses two topics, one is the use of modified sampling designs to increase covariate variability and the second is the implications of longitudinal designs and modified sampling schemes in longitudinal designs.

*R3.2 Also, I find the ultimate utility of the work is limited by the authors not directly addressing how to deploy these findings in practice. That is, unlike designing the stimuli timing for a task fMRI study, the setting of large scale studies is limited by a constellation of factors, often cost. What exactly *are* the concrete recommendations of this work? Should researchers *subset* a large-scale dataset to find subsets of phenotypes with large variance? I.e. sacrifice total N to increase $SD(X)$? Or should they find ways to inexpensively test 1000's of potential subjects, to then only recruit for scanning the extreme range of a phenotype? Extreme phenotype designs have been used for some time in genetics, most recently for studying rare variants (see e.g. <https://www.ncbi.nlm.nih.gov/pmc/articles/PMC7564972/>) and is an example of putting these concepts into practice.*

Discarding independent subjects will not increase power so these suggestions will not help retrospectively. To increase sampling variance of the covariate, it would be necessary to inexpensively collect one or more covariates of interest and choose a subset of individuals to scan based on their collected covariate values to target a specific distributional shape that increase the covariate variability. We provide more explicit recommendations of sampling strategies that could be employed to increase ESs in the second paragraph of the subsection, *Increasing between-subject variability to increase effect sizes*, in the Discussion,

Because standardized ESs are directly related to study efficiency, design considerations that optimize study efficiency will increase ESs and study replicability. Two-phase, extreme group, and outcome-dependent sampling designs can inform what subjects should be selected for imaging from a larger sample in order to increase the efficiency and ESs of imaging-behavior associations³⁸⁻⁴⁴. When there are multiple covariates of interest, multivariate optimal sampling designs can be used to increase effect sizes⁴⁵. As we see here, use of optimal designs increases power, but positively biases estimated ESs relative to their population values due to manipulation of the sample variance⁴². Weighted regression or other statistical methods can be used to compute unbiased ES estimates that account for sampling design⁴⁵⁻⁴⁷. When there are concerns about implementing optimal design strategies, choosing optimal psychometric measurements or interventions (e.g., medications or neuromodulation) within a clinical trial using cross-over or between group comparisons may also be effective for increasing ESs for a target association. The decision to pursue a particular design will depend on the costs and complexity of acquiring the non-brain variables of interest and translational goals that may require population representative ES estimates.

We now also offer practical recommendations with regard to longitudinal study design in Supplementary Section 2, which we summarize in the second paragraph of *Longitudinal design considerations* subsection of the Discussion,

Longitudinal models offer the unique ability to separately estimate between- and within-subject associations. When estimating these effects separately, investigators can use a GEE or LMM with any working covariance structure which will yield unbiased estimates of the between- and within-subject effects if the model is correctly specified. Using sampling strategies to increase between- and within-subject variance will separately increase ESs for the between- and within-subject associations, respectively. Estimates of between- and within-subject associations will not be directly comparable to estimates obtained from a cross-sectional model because they have a different interpretation (See Supplement: Section 2.2)^{49–51}. We discuss several other reasonable modeling approaches for longitudinal BWAS in the Supplement (Section 2).

*R3.3 Related, are the practical implications of the findings on cross-sectional vs. longitudinal studies. In the context of brain imaging, a dominant cost factor can be scanning time, and then a reasonable reference is *not* comparing longitudinal with N subjects to a cross-sectional study, but rather a longitudinal study with $n = \sum_i n_i$ measurements to a cross-sectional study with n subjects. Even if a resampling approach isn't clear, perhaps even a simple adjustment (e.g. using a factor n/N) could easily address this question and enrich the results.*

We have now expanded the theory to clarify the results of the comparison between cross-sectional and longitudinal designs, and modifications of the variability of the covariates in longitudinal designs. A detailed response is given to comment R2.2. To briefly recapitulate, here, our sampling schemes applied to longitudinal data (Section ***Preferred sampling schemes and longitudinal designs depend on state versus trait associations between brain and non-brain measures***; Fig. 5) showed that increasing within-subject variability was detrimental for some associations. To investigate this further, we compared cross-sectional and longitudinal models in the full ABCD cohort and show that many ESs are smaller using longitudinal designs (Fig 6). This is due to the typical linear model incorrectly assuming the between- and within-subject effects are the equal. Figure 6e shows that this is not true for structural associations with the psychometric variables in the ABCD. The theory in the Supplement (Eqn. 11) shows that when the between- and within-subject effects differ, the parameter estimate that assumes they are equal is a weighted combination of the two and the weight depends on the *assumed* within-subject correlation of the outcome as well as the true within-subject correlation of the covariate. Finally, we show that when modeling the between- and within-subject effect separately in longitudinal models, we can separately control the between- and within-subject effect sizes by manipulating the between- and within-subject variance, respectively (Fig. S11).

Understanding these empirical results in this theoretical context allows us to offer some concrete recommendations for cross-sectional and longitudinal modeling in the subsection *Longitudinal design considerations* of the Discussion and Supplement. Longitudinal designs are advantageous in that they offer the unique ability to estimate between- and within-subject effects separately. When not separately estimating between- and within-subject effects in a longitudinal model recommended in Supplementary Section 2.3, we show that with minimal assumptions it is better to conduct a cross-sectional design in the sense that you describe with regards to the number of total scans.

Another concern is the use of the variable age, which has various special aspects, but notably explains a terrific amount of variance. I see a similar body of work has just appeared <https://www.nature.com/articles/s41562-023-01642-5> using different measures of effect size, and the added value relative to that work should be discussed. Also, it seems notable that longitudinal effect sizes were dramatically stronger for age, but not for behavior variables in ABCD... this seems like a paradox but it was never discussed or unpacked.

Thanks for pointing us to this work. In the work of Liu et. al. (2023), they focused on exploring the replicability functions over sample sizes for various brain-phenotype associations and found that age is among the covariates that has the largest effect sizes and best replicability on many brain measures in the cross-sectional UKB study. They also empirically found that replicability is positively related to effect sizes and the associations with larger effect sizes require smaller sample sizes to achieve ideal power, which both have been well indicated by the statistical theory. We now have added this reference in the introduction.

They also showed a strategy of “preselection of individuals based on extreme phenotypes” could decrease the required sample size for a pre-specified power level. Such a strategy (as well as the extreme phenotype designs you mentioned before) is similar to our sampling schemes but simpler. Their investigation did not investigate longitudinal designs or provide explicit recommendations on sampling and modeling that we have added in the revision.

In the revision, we made the sampling schemes more feasible to researchers in practice (by letting the covariates distribution follow pre-specified shapes), and also extended the implementation to longitudinal setting using ADNI and ABCD. More importantly, after showing the benefits of longitudinal designs and increasing within-subject variability to effect sizes are highly dependent on the brain-behavior associations (as it works for age but not for many other non-brain variables in ABCD), we now unpack more about the theoretical reasons behind it – the between- and within-subject association strengths are equal for age (Figure S13) whereas not equal for many cognitive measures. In the end, we also provided concrete recommendations for cross-sectional and

longitudinal modeling in the subsection *Longitudinal design considerations* of the Discussion and Supplementary Section 2 (please see our responses to your comment R3.3).

Regarding the exposition, given that the core of this work concerns effect size, there a surprising lack of clarity about what exactly is meant by "standardized effect size". The issue is that there is no one single measure of effect size, as the exact form of "standardized effect size" varies depending on whether one is working with a one-sample, two-sample or regression setting, no less considering complexities of mixed models [see e.g. Lakens (2013)]. The working assumption of the authors seems to be that the reader is well-familiar with RESI; notably, despite being a centerpiece of the work, RESI is never defined in the body or methods (it is in the supplementary materials). While it is understood that the audience is not statistical, some sort of exposition on RESI should be given in the methods of this work. (E.g. the connection to $d/2$ seems arbitrary; less arbitrary, it seems, is to simply say that RESI is robust version of Cohen's f).

By standardized effect size we mean that it is an effect size that has been rendered unitless by scaling out an estimate of the units (e.g., the standard deviation). We've added a sentence describing this in the second paragraph of the introduction, "Standardized ESs (Pearson's correlation, Cohen's d , Cohen's f) quantify the strength of an association on a unitless scale."

We've added a description of the RESI in the Methods section *Robust effect size index (RESI) for association strength*,

The RESI is a recently developed ES index that has consistent interpretation across many model types, encompassing all types of test statistics in most regression models^{30,31}. Briefly, the RESI is a standardized parameter describing the deviation of the true parameter value β from the reference value β_0 from the statistical null hypothesis $H_0: \beta = \beta_0$,

$$S = \sqrt{(\beta - \beta_0)^T \Sigma_{\beta}^{-1} (\beta - \beta_0)},$$

where Σ_{β} is the covariance matrix for $\hat{\beta}$ the estimator for β . More details about the definition can be found in Supplement.

We provided a consistent estimator for RESI

$$\hat{S} = \left\{ \max\left[0, \frac{T^2 - m}{N}\right] \right\}^{1/2},$$

Where T^2 is the Chi-square test statistics testing for the null hypothesis $H_0: \beta = \beta_0$, m is the number of parameters being tested (i.e., the length of β) and N is the number of subjects.

As RESI is generally applicable across different model and data types, it's also applicable to the situation where Cohen's d was defined. In this scenario, the RESI is equal to $\frac{1}{2}$ Cohen's d ³⁰, so Cohen's suggested thresholds for ES can be adopted for RESI: small

(RESI=0.1); medium (RESI=0.25); and large (RESI=0.4). Because RESI is robust, when the assumptions of Cohen's d are not satisfied, such as when the variances between the groups are not equal, RESI is still a consistent estimator, but Cohen's d is not. The confidence intervals (CIs) for RESI were constructed using 1,000 non-parametric bootstraps³¹.

RESI is not explicitly model dependent because it is defined from the noncentrality parameter of the test statistic which can be computed from most model types, in contrast to common effect sizes (e.g. Cohen's d , f , R -squared). We relate RESI to the scale of Cohen's d because it is a common standardized effect size whose scale most researchers are familiar with, whereas Cohen's f is less commonly reported.

Regarding the choice to focus on RESI, it isn't clear whether RESI be viewed as a convenience, or a unique and special element of this work. If the latter, will these findings generalize should the user choose other measures of effect size? E.g. what if the analyst used linear mixed effects models instead of a GEE? In this univariate (not voxelwise) setting, LME would seem to be the natural choice, requiring comment on the applicability of CS-RESI/L-RESI comparisons when LME is used with longitudinal data.

RESI is used for convenience so that we can report the same effect size across different types of parameters (e.g. for age-GMV associations we are testing nonlinear splines terms using a Chi-squared statistic on 3 degrees of freedom, for sex differences we are performing a Z-test, and for the psychometric variables we are performing a Z-test). The RESI allows us to use a single effect size while incorporating the robust covariance estimator typical of GEEs.

GEEs were chosen for convenience, robustness, and computational efficiency. First, we have rigorously evaluated the performance of the RESI bootstrap confidence interval procedure for GEEs and reported these results in the supplement. Second, GEEs use robust standard error estimates and we did not want our effect sizes to be biased by potential heteroskedasticity. Lastly, GEEs have a vastly faster computational time, especially for large datasets such as ABCD; the analyses in the ABCD take approximately >5 times longer time to run for LMM than GEEs without bootstrapping. In the context we discuss here, GEEs and LMM are equivalent models because we are using an identity link function for the GEE, so point estimates should be equivalent within the numerical error of the different estimation procedures. Because linear mixed models are the most common approach for BWAS, we now report the point estimates for the findings in the ABCD in Figure S10 to show that the results are similar to those shown in Figure 6. Both are copied below for convenience.

Fig. S10. Replication of Fig. 6a-d using linear mixed models (LMMs) with random intercepts. The results are similar using both methods and show the detrimental effect of longitudinal study designs for most structural brain associations. Only the point estimates of L-RESI are obtained as the bootstrap CIs in LMMs have not been evaluated.

Fig. 6. Longitudinal study designs can be detrimental to effect sizes and replicability due to heterogeneous between- and within-subject associations between brain and behavior measures. (a-c) Cross-sectional analyses (indicated by “1”s on the x-axes) can have larger effect sizes than the same longitudinal analyses (indicated by “2” or “2+”) for structural brain measures in ABCD. **(d)** The FC measures have a slight benefit of longitudinal modeling. The cross-sectional analyses only use the baseline measures; the longitudinal analyses use the full longitudinal data. **(e)** Most regional GMV associations (Fig. 5c) have larger between-subject parameter estimates ($\hat{\beta}_b$, x-axis) than within-subject parameter estimates ($\hat{\beta}_w$, y-axis; see Supplement: Eqn (13)), whereas **(f)** FC associations (Fig. 5g) show more heterogeneous relationships between the two parameters.

Minor Issues.

Line 158: There is a cryptic reference to bias: "The bias we note..." This is the first mention of any bias in effect size measures; a careful study of the referenced Fig. 2 does not explain what is being referred to here.

We were referring to the difference in the L-RESI and the RESI estimated in a cross-sectional study. We've changed it to the work "difference" instead of "bias".

Line 687: Related to Line 158, is the caption of Fig. 2, which refers to a "modified CS-RESI", which doesn't really make any sense... i.e. if Fig. 2A plots CS-RESI - RESI, how can that be "removed" without giving a zero value? Or, more simply, what is the value of Fig 2B? Note also there is no other reference to "modified CS-RESI" anywhere else in the submission.

We've removed the term "modified." Figure 2b is showing the difference in the CS-RESI, estimated from the longitudinal study, to the RESI estimated from the cross-sectional study to show that they are estimating the same value (the line goes to zero).

References

Lakens, D. (2013). Calculating and reporting effect sizes to facilitate cumulative science: A practical primer for t-tests and ANOVAs. *Frontiers in Psychology*, 4(NOV), 1, 1-12.
<https://doi.org/10.3389/fpsyg.2013.00863>

Reviewer Reports on the First Revision:

Editorial note: Due to the technicality of the response and the fact that Reviewer 3 could not re-review the paper, we engaged two more referees, Reviewers 4 and 5, to comment on the methodological and technical aspects of the paper.

Referee #1 (Remarks to the Author):

General points:

In the revised version of their article, Kang et al. now include regional structural metrics and functional connectivity metrics. The first half of the article (Figures 1-4) convincingly makes the point that when you are studying the effects of age on structural brain metrics (including regional ones), you should spread out your longitudinal timepoints (within and between participant) and employ a u-shaped sampling scheme. In addition, it appears as if two timepoints per participant already give you most of the benefit of using a longitudinal design. These points are relevant for designing lifespan, developmental and aging studies, clearly made and unlikely to be controversial.

One question related to this first section would be whether the authors recommend abandoning trying to get more than two timepoints per participant. For example, the ABCD and HBCD studies plan on getting five timepoints per participant; is that a suboptimal distribution of datapoints? Should more participants be enrolled, but with fewer timepoints per participant?

In the second part of the manuscript (Figures 5,6), focus switches from age to other variables. For a collection of non-age variables, Kang et al. also show that a u-shaped sampling scheme (vs. a bell-shaped) increases the effect sizes for cross-sectional designs. The same is true for longitudinal designs when the u-shaped sampling increases the between-participant variability. However, in longitudinal designs, if the u-shaped sampling increases within-participant variability it mostly seems to decrease the effect sizes. In the ABCD study, effect sizes for most cognitive testing scores are higher when only using the baseline data (cross-sectional), than when using the longitudinal data. For measures that do not change longitudinally (birthweight) or those that are not expected to change, have little measurement error, or state dependence (handedness) this effect is not seen. The effect is also absent for functional connectivity measures. The authors then conduct follow-up analyses showing that structural metrics have much larger between-participant parameter estimates, than within participant (Fig. 6e). Whereas functional connectivity metrics have more equivalent within- and between-participant parameter estimates (Fig. 6f). The authors then argue that in longitudinal designs one should separately estimate within- and between-participant associations (e.g., Fig. S11). At least when comparing measures with high state dependence (e.g., crystallized intelligence) to those with minimal state dependence (gray matter volume). Otherwise, the much larger between-participant associations are reduced by inclusion of the much smaller within-participant associations.

In this second section, the data related to u-shaped sampling for cross-sectional designs, for variables other than age are also convincing. The finding that u-shaped sampling aiming to increase between-participant variability for non-age variables also increases effect sizes for longitudinal designs is similarly convincing. Although the question, raised in the previous round of reviews, as to how studies should best screen variable distributions across many dimensions, remains. Should study investigators get access to existing standardized testing scores (i.e. from schools) and then preferentially recruit from the tails of the distributions? Is that feasible? Should studies conduct

some form of cost-effective abbreviated online testing to guide recruitment? It seems that these approaches would be most beneficial for hypothesis driven studies. If the goal is to collect information about populations, and to generate population-representative samples that can be utilized to investigate a broad set of hypotheses, it is hard to argue against uniform sampling. It seems that if a very large effort is meant to generate a relatively future-proof, multi-purpose data set to be used by all interested researchers, uniform sampling might still be the way to go. Given that in a real-world US sample, many variables are correlated, for example, with very tricky sociological issues, one could accidentally end up with a sample that pits rural vs. urban or private schools vs. public schools at the ends of some set of distributions; these are just examples, there are many more. Uniform sampling also seems like a reasonable means for avoiding societal/political outcry over not sampling certain subpopulations, intentionally or unintentionally. In terms of the discussion of the u-shaped sampling benefits, which are undeniable, for a given variable, or even collection of variables, it still seems like the ultimate benefits depend on how broad or narrow (hypothesis-driven) the study goals are. For a hypothesis driven sample, u-shaped sampling should be the norm. For studies, like UKB or ABCD/HBCD that could be considered society-wide efforts unlikely to be duplicated/replicated anytime soon, uniform sampling might still be the safest choice for a series of reasons.

The newly added findings related to the differences between structural and functional metrics, likely at least in part state-related (Fig. 6e,f), are interesting. Here, my biggest questions are about interpretation. For the associations between cognitive and structural brain measures, having longitudinal data reduces the effect sizes, something the authors refer to as 'detrimental'. Given our history of falsely preferring larger over smaller effect sizes, that word seems too loaded. It seems more desirable that the effect sizes are replicable and generalizable. Bigger is not necessarily better when it comes to effect sizes.

In the specific scenario of structural brain metrics vs. cognitive measures, adding a longitudinal measurement reduces the effect size. Typically adding measurements, brings one closer to the truth, so then the question becomes why should one think that the larger, cross-sectional effects would be closer to the truth? If the 'longitudinal' measure was a repeated measure in that it happened only a few days later, would it also be expected to reduce the effect size? Only a few days later, one would not expect a structural brain metric to have changed. So if the effect size was still reduced, would that suggest that the original cross-sectional effect size was inflated due to sampling variability, and that repeated measurement increased the accuracy of the non-brain metric, which reduced the effect size? Associating structural brain metrics with non-brain variables that are not state dependent (e.g. handedness) does not show this effect, neither does comparing functional brain metrics (state dependent) to state dependent cognitive measures. Is it possible/likely that functional brain and cognitive measures collected close together in time, as in most studies, are subject to similar state effects, so that anxious during scanning and anxious during cognitive testing at timepoint 1 and happy during scanning and happy during cognitive testing at timepoint 2 allow for larger longitudinal BWAS effects? Is the issue with forcing a longitudinal analysis onto an association that is devoid of a true longitudinal relationship? For example, in the case of cognitive testing scores, which are supposed to be stable traits, but are also variable day-to-day based on whether the child had their Wheaties in the morning, etc, should we refrain from correlating them against structural metrics that are not variable on such rapid timescales. Or put differently, is it advisable to correlate

structural brain metrics which are state invariant against cognitive measures that are state dependent. For example, total gray matter volume is independent of whether someone has a 10/10 tension headache, but testing performance is not. If we now sample someone across the state space, either rapidly (repeated sampling) or with a delay (longitudinal), the state-driven variance in the non-brain measure can drive down the effect size. But which effect size is closer to the truth? Maybe longitudinal sampling with a u-shaped sampling scheme is always superior, but it does not always enlarge the effect size.

Overall, Kang et al., convincingly argue several points important for optimizing study designs, but the framing of the arguments could perhaps be refined further. The results themselves seem to demonstrate that there are fundamental differences between studying the effects of age on the brain, which this group of authors has done extremely successfully and relating other variables to the brain. Similarly, there seem to be differences between structural and functional brain metrics and state-dependent/independent non-brain metrics. Instead of trying to throw these different types of comparisons into the same pot, it might be more informative to highlight these differences and to then present the results in terms of recommendations that are specific to certain analyses vs. recommendations that always apply. For example, it seems that increasing between-participant variability by u-shaped sampling is always a good idea unless your goal is to represent the general population?

The manuscript makes important and valid points about study designs. Therefore, it does not need to employ rhetorical constructions that could be taken as encouragement to circumvent sample-size requirements (last sentence of the abstract). Instead, the manuscript would be further strengthened, if it focused on highlighting the additional power to be gained from pairing larger sample sizes with optimized (u-shaped) sampling schemes. In contrast to sampling schemes, which seem to have some specific dependencies, larger samples universally improve replicability and generalizability of BWAS results.

Specific points:

Title:

The term 'longitudinal BWAS' clashes with the original definition of BWAS as cross-sectional.

Abstract:

The word 'detrimental' is very strong.

Intro:

Setting up the importance of BWAS as relevant to diagnostics and prognostics, seems slightly off the mark. It seems more relevant to epidemiology since diagnostics and prognostics require high sensitivity and specificity that allow clinicians to alter care for a specific patient.

Is a whole brain metric still a brain-wide association since it cannot be expressed as a map. Is it more or a brain-average association?

Figures: It would be helpful to also embed them in the text for easier reviewing

Fig. 5. Why does decreased effect size with u-shaped within-subject sampling suggest a decrease in power, not a correction of inflated effect sizes?

Fig. 6d. The functional connectivity effect sizes seem extremely small. The thick black bar is at zero for all of them, is that correct?

For supplemental figure captions, please repeat the key information each time. The captions should be sensical, even if read out of order. For example Fig. S10 caption does not spell out what 1, 2+ on the x-axis means.

Referee #2 (Remarks to the Author):

The authors have provided a comprehensive response to the reviewers, with new material and substantial revision. Overall, I am persuaded that the paper makes a useful contribution to the broad discussion on BWAS effect sizes and what brain imaging can and cannot reveal about behavior.

A particularly interesting feature that has come out of this analysis is that between-person associations and within-person associations between brain and behavior have different effect sizes, and that common modeling strategies (e.g., LMMs, GEEs) average over them.

Thus, longitudinal designs are less efficient in many cases. In fact, they are beneficial mainly in tracking age, but not for other behavioral variables (like cognitive performance scores on measures from the NIH toolbox). This implies that within-person variations in structure (gray matter) track age, but do not track changes in other performance measures well. Within-person variations in functional connectivity track performance a bit better, which is suggestive, though it's unclear if this difference is reliable.

Pointing this out, and developing the rationale and explanation more completely, allays my previous concern that the basic findings on effects of sampling and predictor variance are well known from basic statistical theory. The authors argue that (1) real-world data may deviate from predictions made by normative theory for a number of reasons, and there is value in presenting results from actual datasets; and (2) the distinction between within- and between-person effect sizes and its implications for study design are not well understood. I am persuaded on both of these points. I also think that the idea of separating between- and within-person effects is important and under appreciated, and this paper will inform the next round of large-scale studies of brain function and structure.

There are some remaining features of the presentation that make it difficult to interpret the results in some cases. This list is not exhaustive, but may help in tightening all the descriptions so that the results are interpretable and usable by future readers.

For example:

Fig 5: Boxplots reflect bootstrapped associations, but how was region handled in regional GMV and CT analyses? Are these effect sizes over randomly selected regions?

Tab. S13: What are the intra-subject correlations and how are they calculated? (They are not mentioned in the Note)

When RESI is explained, there is a clear expression for S , but the expression for \hat{S} uses T^2 , without clearly connecting T^2 to S . Thus, many readers will not be able to follow this without outside references.

RESI is not explicitly defined with respect to \hat{S} , though it is implied that $RESI = \hat{S}$ (and this is only made explicit in a different section of the methods). \hat{S} is also multivariate and depends on all the beta-hats in a model, and it's not made explicit how the effect sizes for separate terms (or sets of terms) in the model are calculated.

The paper (Fig. 6) refers to “methods for estimating between-subject and within-subject associations separately”, but no reference or document explaining what these are is mentioned here. Presumably this is explained in the new technical supplement Section 2.2 and 2.4, but it’s not clear.

Referee #4 (Remarks to the Author):

The manuscript used a meta-analysis of robust effect size index (RESI) to demonstrate that enlarge the covariate variation and/or switching a cross-sectional study to longitudinal can improve the standardized effect size. First, I am not convinced that “standardized effect size” is a quantity that can be improved via study design. Second, “standardized effect size” is not the only and major concern of deciding whether a study should be longitudinal. Third, the conclusions were mainly derived from studying the association between imaging features and age which is not generalizable. The following are my comments.

Major Concerns:

1. My understanding of effect size should be an underlying feature of a biological process, rather than a quantity that can be “improved”. I am confused about the authors’ study objective here. Perhaps, “effect size” is a wrong term of the quantity the authors tried to improve here.
2. In general, the purpose of conducting a cross-sectional study and a longitudinal study is different. Compared to a cross-sectional study, for a longitudinal study, the study interest may focus more on longitudinal/temporal (intrasubject) variation of the outcomes. Thus, the determination of running a cross-sectional or longitudinal study does not due to the concern of power or replicability, but more of the research question. Switching from a cross-sectional to longitudinal study, just due to the concern of improving standardized effect size, may not be feasible or applicable for many studies. On the other hand, running a longitudinal study takes longer time, may cost more and lose subjects at follow-up visits. These are also factors that should be taken into consideration during study design. The current study seems to ignore these aspects and only considers the improvement of one feature.
3. “Meta-analyses shows ESs depend on study population and design” section.
 - (a) When getting RESI and L-RESI, the authors used data from multiple cross-sectional and longitudinal studies, respectively. From my understanding, the authors did not take the variation in study protocol or imaging scanner effect into consideration. This has been demonstrated as a concern when analyzing imaging data collected from multiple sites.
 - (b) It is not very surprising of the results that standardized ESs depend on age as the two high RESI periods are ages of neurodevelopment and neurodegeneration. And if one aims to study the association with age, for sure, one should recruit subjects on a reasonably wide age range (for example, one cannot study the association with age when the recruited subjects are all about the same age). Therefore, I am not sure how the current analysis results provide more information to a study design.
4. “Improved sampling schemes can increase ESs” section
 - (a) Not sure what the “bell- and U-shaped” sampling distributions look like.
 - (b) What is the age range of the UKB data? If it is across lifespan, I am not very surprised that the U-shape “performs” better since my understanding of the U-shape puts higher sampling weights on the age period of neurodevelopment and neurodegeneration (the two ends of the age range).
 - (c) The current conclusion of the association with age may not be generalizable to the study of other factors (covariates).
5. “Preferred sampling schemes and longitudinal design XXX” section

- (a) The ABCD study subjects at baseline are mainly between 8 to 12 years old, not a wide age range. Not sure how this small age range will impact the findings. On the other hand, there are many other features that may have an impact on brain development. For example, not all subjects are typically developing children. It seems such information or individual features were not taken into consideration into the analysis.

Referee #5 (Remarks to the Author):

Summary of the manuscript:

The abbreviations used here are ES - effect size, CD - cross-sectional design or cross-sectional study, and LD - longitudinal design.

The authors have leveraged data from the Lifespan Brain Chart Consortium to explore the impact of study design on effect sizes (ES) in both cross-sectional (CD) and longitudinal (LD) designs within Brain-wide Association Studies (BWAS). Their comprehensive analysis, bolstered by a meta-analysis, seeks to clarify how ES correlates with replicability across BWAS. The manuscript's strengths are manifold, including (1) the use of a rich dataset enabling detailed examination of design elements such as sample size, frequency of imaging scans per participant, and the derivation of global and regional brain metrics, while also including covariates like age, sex, cognitive abilities, psychopathology, and demographics; (2) R codes with workflows to implement similar analysis for other BWAS studies on optimizing ES.

The manuscript's current analysis and findings present a compelling case for publication in Nature, contingent upon a detailed revision of the workflow and result descriptions. It would be particularly beneficial for the authors to outline their recommendations step-wise in the Conclusion section, especially when deciding between CD and LD for Brain-wide Association Studies (BWAS) based on specific responses and covariates. Clarification on the choice between CD and LD should be accompanied by a discussion on design aspects such as the number of participants, the frequency of sample collection per participant, and the total number of samples collected, with a particular focus on the effect size (ES) concerning one or multiple covariates. An analysis of the time intervals between repeated samples for each subject should also be included to provide insights into the optimal study design. However, I understand it is not reasonable to ask it now. I'd suggest caution be exercised in making definitive statements about the lesser superiority of longitudinal designs in enhancing ES, a concern previously highlighted by reviewer R2 without including time interval analysis. Addressing these recommendations can significantly improve the manuscript's contribution to BWAS studies.

Major concerns:

1. The findings presented in the abstract appear to be misaligned with the datasets and variables detailed in the manuscript, necessitating a thorough revision of the abstract to accurately reflect the manuscript's content. Specifically, the manuscript illustrates how ES can be improved by adding an additional longitudinal measurement per subject within the ADNI dataset (line 247), a detail generalized in the abstract to encompass the LBCC dataset ((line 51)). To enhance clarity and coherence, I recommend revising the abstract to accurately represent these findings, ensuring it aligns with

the specific datasets and variables analyzed in the study. This adjustment will not only improve the abstract's precision but also ensure readers gain an accurate overview of the study's conclusions.

2. The manuscript needs clarification regarding the categorization of univariate and multivariate analyses in the context of Brain-wide Association Studies (BWAS). Specifically, the study designates analyses involving only age as a covariate as univariate, aligning with simple linear regression principles. In contrast, analyses incorporating multiple covariates are labeled as multivariate models akin to multiple linear regression. The confusion is compounded in line 377, where the manuscript describes multivariate BWAS by inverting the roles of response and covariates, treating brain measures as imaging variables while seemingly assuming psychometric or demographic measurements as the response in univariate analyses.

The multivariate approach could be considered when predicting multiple responses, allowing for the decomposition of effect sizes (ES) of various brain measures on multiple responses.

Clarification is needed to differentiate between univariate and multivariate BWAS accurately, ensuring a clear understanding of when and how each analysis type is applied, particularly in terms of response and covariate roles. This distinction is crucial for accurately interpreting the study's methodology and findings.

3. The organization and clarity regarding the dataset and RESI descriptions within the manuscript require enhancement for better comprehension and flow. Notably, there are instances of redundancy and fragmentation in presenting information about the datasets and RESI, which could potentially confuse readers. For example, identical details about the datasets are mentioned in both lines 91 and 125, as well as in lines 123-127 and 136-138, leading to unnecessary repetition. Furthermore, the explanation of CS-RESI is split between lines 102 and 182, with the latter providing crucial understanding that could be more effectively communicated if introduced in a consolidated manner earlier in the manuscript. To resolve these issues, I recommend a thorough reorganization of the manuscript to eliminate repetitive information and ensure that concepts like CS-RESI are fully explained at their initial mention. This will not only streamline the content but also facilitate a smoother, more logical progression of ideas for the reader.
4. Lines 730-731 and other places: My concern centers on the methodology used for calculating statistical power throughout the manuscript. The authors describe power calculation as the proportion of bootstrap replicates yielding p-values less than or equal to 5%. However, statistical power is traditionally defined as the probability of correctly rejecting the null hypothesis when there is, in fact, a true effect (i.e., when the alternative hypothesis is true). The critical issue with the described approach is that it assumes all observed data inherently exhibit a non-null effect, which may not be the case. Without confirmation that the bootstrap samples are drawn from a distribution under the alternative hypothesis (i.e., one where a non-null effect is present), equating the proportion of bootstrap replicates with p-values ≤ 0.05 to statistical power is problematic. This is because such a proportion could potentially include false discoveries, given the lack of certainty regarding the presence of a true effect in the observed data. A more nuanced approach or additional clarification on how the authors account for this potential discrepancy in their power analysis would enhance the credibility and interpretability of their findings.

Some of the minor suggestions for improving the text are as follows.

1. Lines 41 - 43 and 50 -52: I assume both lines are about the Lifespan Brain Chart Consortium (LBCC), so why don't you mention this in lines 41-43 instead of specifying the data source in line 50?

2. Lines 41-42: Is it CS RESI? or just RESI?
3. Line 99: independent variable - I suggest either covariate or predictors instead of independent variable. In line 109, the authors used covariates. So, there needs to be more consistency in terminologies.
4. Lines 102 - 105: mentioned that CS-RESI improves the comparability of ESs between LD and CD, but there is no reference to this.
5. Lines 105 - 107: This might be for non-biologists - what is the sampling scheme in the UKB and ADNI? Are they related to how to sample MRI scans or sampling individuals based on predictors?
6. Lines 109 - 111: Is ABCD a subset of the LBCC? If not, why do you choose ABCD to generalize the findings on GMV-age association? Moreover, generalization is many folds. Is it here for each response (brain measure) and one covariate?
7. Inconsistent usage of “brain outcome measure”, “brain measure”, and “brain outcome”. I’d use one of the terms throughout the manuscript. For example, lines 111, 50, 54, etc....
8. Lines 113- 114: What are the common longitudinal models conflating between and within-subject effect? Did you consider modeling both cross-sectional and longitudinal effect (adjusted for cross-sectional) in the common model?
9. Lines 119 - 123: To fit each study level analysis, “.... spline function of age and sex in 63 neuroimaging datasets from the LBCC....”. This line needs to be clarified. Did you regress each response (total GMV, total sGMV,...) on each covariate (age, sex) in each of the 63 studies separately? What is the exponential family used in the generalized linear regression? What are the model assumptions for GEE? Does the response follow any particular exponential family?
10. I’d suggest describing the dataset used in the sub-analysis with respect to LBCC. I understand there are 63 neuroimaging datasets from the LBCC. Then, I get lost in the subsets. For example, lines 121 - 125 states there are 47 CDs and 16 LDs containing 77,695 MRI scans from 60,900 CN individuals. Then, lines 126 - 127 mention 43 datasets (30 CD and 13 LD) for the mean CT and regional measures. Are these dataset subsets of 63 datasets, and are there any overlaps? Tables S1 and S2 give answers to the questions, but the reader will benefit if you refine the dataset description in lines 119 - 12. Then, write about the RESE in lines 128 - 135 in the next paragraph.
11. Lines 102-104 and 128 - 135 Aren’t they about RESI and its modification to LD? I think describing the RESI in lines 102 - 104 is better. Again, lines 181-182 are about defining and evaluating CS-RESI, and we need to see the details in the Supp Section 1. Why can’t all these be summarized in a section and refer to the details in the Supplement?
12. Line 15: Is it study population SD and skewness of age covariate?
13. Line 159: What does “respectively” mean?
14. Another inconsistency in using study design, study population, study population features, study features, and study design features. I’d use terms consistently throughout the manuscript.
15. Line 188: Is it 1000 bootstraps per dataset (16 LD)?

16. Line 189: Why the bootstrap samples are longitudinal subsamples? I thought they were bootstrap resamples. Aren't they with the same number of subjects and possibly a different number of total samples?
17. Lines 204: Fig. 2b doesn't describe N. Is this asymptotic related to the increasing number of individuals, the number of repeated samples per individual, or both?
18. Line 206: ESs from all studies using the CS-RESIs in longitudinal studies.. can it be ESs from all longitudinal studies using the CS-RESIs?
19. Line 221: "...schemes that sequentially increase..." is it consequently increase?
20. Line 226 & Fig 3 b.: How did you calculate the power in detecting age ES?
21. Line 237: the study, not "a study".
22. Lines 240 - 241: Is there a reference to show L-RESI and power relationship?
23. Lines 246 -250: Is this investigation on the ADNI dataset?
24. Line 260: is FC the functional?
25. Line 266: ...non-brain covariate? In line 262, the authors used "non-brain covariates". Then, why, in line 266, did they change the same to "non-brain measure"?
26. Line 275: It is unclear what is "for each data type"?
27. Lines 279-280: "...than conducting longitudinal analyses using the full longitudinal data". Here, what are the longitudinal analyses? Is it ES analysis using the longitudinal data?
28. Line 282: Are these random intercepts individual-specific?
29. Lines 283 - 285: I'd not conclude that the longitudinal designs can be detrimental to ESs because the frequency of time points would affect ESs. So I'd remove "and, counter-intuitively, longitudinal design can be detrimental to ESs".
30. Lines 300-301: fitting these associations separately,... is separate indicating CD on the baseline samples and LD on the repeated samples or CD on all repeated samples but assuming all repeated samples are independent?
31. Discussion section: Most of the above comments apply to the discussion section. In addition, why it is, in lines 375-386, imaging variables instead of brain measures?
32. Figures labels: Why are the labels a light grey color (very hard to read)? Can't they be black? I haven't gone through the captions for the figures and tables, but there might be revisions needed based on the answers to the above concerns.
33. Line 555: repeated based on twice
34. Line 557: is ADNI part of the LBCC? It needs to be clarified in this sentence.
35. Methods: Statistical analysis:

- (a) In all model specifications, I can't see the model assumptions. What is the distribution of Y_{ij} ?
- (b) Line 632: with two degrees of freedom?
- (c) Line 658: Where is the supplement? I have figures and tables in the supplement.
- (d) Line 659: We provided means – is it introduced in this paper?
- (e) Line 660: It must be

$$\hat{S} = \left(\max\left\{0, \frac{T^2 - m}{N}\right\} \right)^{1/2}$$

note that $\{\}$ used for set notation.

- (f) Line 661: Where - start with a lowercase letter for “where” ... and why is C uppercase in Chi? What is the chi-square test statistic to test the hypothesis on the location parameter β ?
- (g) Line 669: ...bootstrap samples. I'm afraid that the sampling subject doesn't account for an appropriate dependence structure within the individual for a longitudinal design.
- (h) I have concern throughout about the GEE. GEE is one of the ways to estimate the parameters in the model. So, I wonder what model the longitudinal data analysis used. Did authors assume the response follows any exponential family? Gaussian (possibly no for a non-negative response, log-normal? or gamma?
- (i) Line 676: Is it a model for CD? If it is for LD, we need two indices for individuals and repeated samples, y_{ij} .
- (j) Line 677: Where - start with a lowercase letter for “where”
- (k) Line 679: The number of knots and degree of freedom are different. Clarify this in lines 679 and 676.
- (l) : Lines 679-680: For GEE models, we used an exchangeable correlation structure- that can be written as an active sentence.
- (m) Line 687: it is confusing when i is used for the individual and study. I'd use k for study.
- (n) Line 688 and 692: weights are the inverse of the standard error of each RESI estimate is repeated.
- (o) Line 695: Why do we need a spline for sex?
- (p) Line 702: Remove “respectively”
- (q) Line 703: Remove “ the partial regression plots”
- (r) Line 728: 95% confidence intervals (CIs) for effect size estimate - remove estimate. CI is constructed for the parameter, not for the estimator.
- (s) Line 814 and wherever it appears: In the GEE models - GEE is a method of estimation, not a model. The GEE is applicable across a wide range of models and settings.
- (t) Line 820: As a statistician, I can't understand the “ frame count of each scan”.
- (u) Line 820-822 and wherever it appears: authors can specify only one time the packages used for GEE estimation and RESI CI calculation. The same statement is repeated many times in the Methods section.
- (v) Line 836: how do you calculate the F-statistic?
- (w) Line 838: What is df here?

- (x) Line 848: Subscripts are not presented well. It might be helpful to specify the model similar to (2.2.4, page 23 of Diggle (2002)).

$$GMV_{ij} \sim \text{ns}(\text{age}, \text{df} = 2) + \text{sex} + \beta_C x_{i1} + \beta_L (x_{ij} - x_{i1}),$$

where β_C and β_L are the cross-section and longitudinal covariate effects, respectively.

References

Diggle, P. (2002). *Analysis of longitudinal data*. Oxford university press.

Author Rebuttals to First Revision:

Referee #1 (Remarks to the Author):

General points:

R1.1: In the revised version of their article, Kang et al. now include regional structural metrics and functional connectivity metrics. The first half of the article (Figures 1-4) convincingly makes the point that when you are studying the effects of age on structural brain metrics (including regional ones), you should spread out your longitudinal timepoints (within and between participant) and employ a u-shaped sampling scheme. In addition, it appears as if two timepoints per participant already give you most of the benefit of using a longitudinal design. These points are relevant for designing lifespan, developmental and aging studies, clearly made and unlikely to be controversial.

One question related to this first section would be whether the authors recommend abandoning trying to get more than two timepoints per participant. For example, the ABCD and HBCD studies plan on getting five timepoints per participant; is that a suboptimal distribution of datapoints? Should more participants be enrolled, but with fewer timepoints per participant?

Response: As you describe below, large consortia like ABCD have broader aims than a hypothesis-driven study of age effects and there is a benefit to obtaining more time points to learn about the nature of within-subject brain-behavior associations. Moreover, non-imaging and imaging measurements that have large state variability and vary together may have stronger within-subject effects than between-subject effects and we expect having more time points per participant would allow greater precision in the estimates of within-subject effects if they are estimated separately. We now include a decision tree delineating when particular designs are preferred (please see our response to editor above).

R1.2: In the second part of the manuscript (Figures 5,6), focus switches from age to other variables. For a collection of non-age variables, Kang et al. also show that a u-shaped sampling scheme (vs. a bell-shaped) increases the effect sizes for cross-sectional designs. The same is true for longitudinal designs when the u-shaped sampling increases the between-participant variability. However, in longitudinal designs, if the u-shaped sampling increases within-participant variability it mostly seems to decrease the effect sizes. In the ABCD study, effect sizes for most cognitive testing scores are higher when only using the baseline data (cross-sectional), than when using the longitudinal data. For measures that do not change longitudinally (birthweight) or those that are not expected to change, have little measurement error, or state dependence (handedness) this effect is not seen. The effect is also absent for functional connectivity measures. The authors then conduct follow-up analyses showing that structural metrics have much larger between-participant parameter estimates, than within participant (Fig. 6e). Whereas functional connectivity metrics have more equivalent within- and between-participant parameter estimates (Fig. 6f). The authors then argue that in longitudinal designs one should separately estimate within- and between-participant associations (e.g., Fig. S11). At least when comparing measures with high state dependence (e.g., crystallized intelligence) to those with minimal state dependence (gray matter volume). Otherwise, the much

larger between-participant associations are reduced by inclusion of the much smaller within-participant associations.

In this second section, the data related to u-shaped sampling for cross-sectional designs, for variables other than age are also convincing. The finding that u-shaped sampling aiming to increase between-participant variability for non-age variables also increases effect sizes for longitudinal designs is similarly convincing. Although the question, raised in the previous round of reviews, as to how studies should best screen variable distributions across many dimensions, remains. Should study investigators get access to existing standardized testing scores (i.e. from schools) and then preferentially recruit from the tails of the distributions? Is that feasible? Should studies conduct some form of cost-effective abbreviated online testing to guide recruitment? It seems that these approaches would be most beneficial for hypothesis driven studies. If the goal is to collect information about populations, and to generate population-representative samples that can be utilized to investigate a broad set of hypotheses, it is hard to argue against uniform sampling. It seems that if a very large effort is meant to generate a relatively future-proof, multi-purpose data set to be used by all interested researchers, uniform sampling might still be the way to go. Given that in a real-world US sample, many variables are correlated, for example, with very tricky sociological issues, one could accidentally end up with a sample that pits rural vs. urban or private schools vs. public schools at the ends of some set of distributions; these are just examples, there are many more. Uniform sampling also seems like a reasonable means for avoiding societal/political outcry over not sampling certain subpopulations, intentionally or unintentionally. In terms of the discussion of the u-shaped sampling benefits, which are undeniable, for a given variable, or even collection of variables, it still seems like the ultimate benefits depend on how broad or narrow (hypothesis-driven) the study goals are. For a hypothesis driven sample, u-shaped sampling should be the norm. For studies, like UKB or ABCD/HBCD that could be considered society-wide efforts unlikely to be duplicated/replicated anytime soon, uniform sampling might still be the safest choice for a series of reasons.

Response: One clarification: the sampling procedures we used target a particular shape (U, uniform, or bell). The distribution of the variable in the population might not be uniform when using random sampling, it could be bell-shaped – in this case, targeting a uniform shape will increase the variance of the covariate and thus may increase the effect size while still obtaining a good representation of the domain of the covariate distribution. A goal of society-wide efforts is, in part, to unbiasedly estimate population distributions, so initiatives like the ABCD might not want to use a modified sampling strategy for the purpose of accurately estimating the distribution of the covariates. That said, trying to target a uniform distribution in sampling for a large number of variables could still be useful albeit at a higher cost for the nonimaging variables. We've added a clause in the discussion detailing how sampling can increase variance from a bell-shaped distribution without obtaining a highly unusual distribution, such as a U-shaped.

When the association is very non-monotonic, e.g., if there is a U-shape relationship between covariate and outcome, sampling the tails more heavily could decrease power,

and it also may decrease our ability to detect nonlinearities in the center of the study population. **In this case, sampling to obtain a uniform distribution of the covariate balances power across the range of the covariate and can still increase power relative to random sampling when the covariate has a bell-shaped or normal distribution in the population.**

Multivariate sampling when using traits as the outcome has been developed by a coauthor, which we refer to in the paper, but we are not aware of existing methods to perform multivariate covariate dependent sampling. As you described, the ethical and societal consequences of the proposed sampling procedures are an important consideration, and balancing covariates to ensure primary covariates are not confounded by their societal correlates will be important and again highlights the importance of using multivariate sampling strategies that could be used to ensure balance. We've added a statement touching on this in the Discussion:

Because standardized ESs are directly related to study efficiency, design considerations that optimize study efficiency will increase ESs and study replicability. Two-phase, extreme group, and outcome-dependent sampling designs can inform what subjects should be selected for imaging from a larger sample in order to increase the efficiency and ESs of brain-behavior associations⁴²⁻⁴⁸. When there are multiple covariates of interest, multivariate optimal sampling designs can be used to increase ESs⁴⁹. **Multivariate designs are also needed to stratify sampling to avoid confounding by sociological variables.**

Because of the complexity of resources at different study sites, it is hard to provide specific details of how to execute the modified sampling without knowing more details about the covariate and how the imaging data are collected. For example, if the covariate varies slowly over time, the measurement can be ascertained in an initial screening and then select a subset for imaging at a later date based on their measured values. If the covariate varies with cognitive state, then it might be necessary to administer cognitive testing and then scan a subset of participants. The most general recommendation is careful consideration of study design. We provide some mathematical details to guide what total sample size is necessary from a normally distributed covariate to obtain a modified distribution that is uniform (Supplement: Section 2). Code for these analyses is provided in the Supplement: Section 4. Large consortia like the ABCD/HBCD or UKB can be used to estimate population distributions of similar covariates to determine optimal sampling strategies that balance study costs and bias of effect size estimates caused by the sampling strategy. We've added a statement describing this in Supplement: Section 2:

Besides optimizing study efficiency, there are non-statistical factors that should be considered when choosing a study design, such as the practical challenges in recruiting the study population. Because of the complexity of resources at different study sites, it is hard to provide specific details of how to execute the modified sampling without information about the covariate and how the imaging data are collected. For example, if the covariate varies slowly over time, the measurement can be ascertained in an initial

screening and then a subset can be selected for imaging on a later date based on their measured values. If the covariate varies with cognitive state, then it might be necessary to administer cognitive testing and immediately scan a subset or to schedule the imaging and re-administer the cognitive test at scanning. Such complex design issues should be determined on a case-by-case basis.

R1.3: The newly added findings related to the differences between structural and functional metrics, likely at least in part state-related (Fig. 6e,f), are interesting. Here, my biggest questions are about interpretation. For the associations between cognitive and structural brain measures, having longitudinal data reduces the effect sizes, something the authors refer to as 'detrimental'. Given our history of falsely preferring larger over smaller effect sizes, that word seems too loaded. It seems more desirable that the effect sizes are replicable and generalizable. Bigger is not necessarily better when it comes to effect sizes.

Response: The literature on effect sizes often conflates effect size parameters and their estimates. In neuroimaging, effect size estimates tend to be large because they are biased by the selection process of multiple testing (reporting effect sizes that have small p -values) and, in some cases, biased study reporting such as p -hacking. Effect size parameters are unknown fixed values quantifying the strength of an association. Because they are directly related to power, larger effect size parameters imply greater replicability with regards to the consistency of conclusions across hypothesis tests – bigger is always better. As you describe, bigger effect size estimates are not necessarily better for replicability and generalizability, because they may be subjected to the bias of the study procedures. We now have included a statement clarifying this distinction in the Discussion:

It is important to note that, in this study, we endeavor to increase ES *parameters*, which are unknown values quantifying the strength of associations, by intentionally modifying design features, such as the covariate variability to increase the efficiency of the unstandardized effect estimate (see Supplement: Section 1). Larger ES parameters are always preferable in hypothesis testing as they have higher replicability. In contrast, larger ES *estimates* do not necessarily imply greater generalizability or replicability, as they can be positively biased due to a small sample size or by the analysis procedures (such as by multiple testing, biased reporting, or p -hacking). Unstandardized effects (such as regression coefficients), which are often related to the biological interpretation of the population, are not affected by the sampling modifications used here and can be unbiasedly estimated using correctly specified models⁴¹.

R1.3.2: In the specific scenario of structural brain metrics vs. cognitive measures, adding a longitudinal measurement reduces the effect size. Typically adding measurements, brings one closer to the truth, so then the question becomes why should one think that the larger, cross-sectional effects would be closer to the truth?

Response: Our findings are meant to demonstrate that, when using a single parameter to model the different between- and within-subject effects, the “truth” (i.e., the standardized ES

parameter; see our response to your comment R1.3 above) changes with the study design (e.g., adding longitudinal measurements or increasing within-subject variability) and it may decrease in specific cases similar to structural brain measures vs. cognitive measures. The convergence you are referring to is implied by the law of large numbers that says adding independent observations to an estimator leads to convergence to the true value. This does not apply when comparing the results of the two models because the second model is a completely different model with different assumptions (not simply adding observations). The hypothesis that the two models have the same effect size is not supported by the data because it would imply that the first, cross-sectional confidence interval (Fig. 6a) completely missed the true effect size parameter and the second one captured it – this would happen less than 5% of the time by chance because the two confidence intervals are nonoverlapping, so by conventional statistical reporting we would reject this hypothesis.

We added another analysis here to further illustrate that the two models are estimating different ES parameters. We restrict to the participants who have both the baseline and the 2nd measurements, and compare the estimated ESs from two cross-sectional analyses, the first one only uses the baseline measurements, and the second one only uses the 2nd measurements. In this case the trend of decreasing ES disappears. This implies that the effect sizes are more similar between the two time points, further emphasizing that the conventional longitudinal model is estimating a different effect size parameter than the cross-sectional analyses and not a result of the convergence via the law of large numbers.

Additionally, we showed mathematically that when there are different between- and within-subject effects (such as the specific scenario of structural brain measures vs. cognitive measures), but we still use one single regression coefficient to model these two effects, the resulting unstandardized effect for this regression coefficient will be a strange, weighted average of the two effects.

$$\mathbb{E}\hat{\beta} = \beta_b \left[\frac{(1 - \rho)\rho_x^2}{(1 + (m - 1)\rho)(1 - \rho_x^2) + (1 - \rho)\rho_x^2} \right] + \beta_w \left[\frac{(1 + (m - 1)\rho)(1 - \rho_x^2)}{(1 + (m - 1)\rho)(1 - \rho_x^2) + (1 - \rho)\rho_x^2} \right] \quad (13)$$

The weights depend on study design features so it will change while adding longitudinal measurements or increasing between- and/or within-subject variability. And in the specific scenario of structural brain measures and cognitive measures, we provide evidence that the within-subject effects are very minimal for this type of association (Fig. 6e). Therefore, the ES parameter for the average effect can decrease while adding longitudinal measurements as it increases the weight of the minimal within-subject effect in such “average” ES parameter.

The standardized ES is also affected, as shown in Fig 6a, and mathematically:

$$S_{\text{sgn}} = \frac{\mathbb{E}\hat{\beta}}{\sqrt{\text{Var}(\sqrt{n}\hat{\beta})}} = m^{1/2}\{(1 + (m - 1)\rho)\sigma_w^2 + (1 - \rho)\sigma_b^2\}^{-1/2}\sigma_y^{-1/2} \\ \times \left\{ \beta_b\sigma_b^2 \left[\frac{(1 - \rho)}{1 + (m - 1)\rho} \right]^{1/2} + \beta_w\sigma_w^2 \left[\frac{(1 + (m - 1)\rho)}{(1 - \rho)} \right]^{1/2} \right\}. \quad (14)$$

So it's not that the ES estimate approaches the constant underlying truth, but reflects how the underlying truth changes with study designs.

We have added an emphasis on the changing ES parameters in the Discussion:

When the between- and within-subject effects differ but we still fit them with a single effect, the coefficient for the covariate does not have the same interpretation (as when the effects are equal) because it is a weighted average of the between- and within-subject effects whose weights are determined by the study design features (see Supplement: Section 3). This difference in interpretation affects both the unstandardized effects and standardized ESs. The apparent lack of benefit in the GMV-psychometric measurement association in the ABCD is due to a lack of association between within-subject changes in the psychometric measurements with within-subject changes in GMV (Fig. 6e; Tab. S13). The smaller ESs we find in longitudinal analyses are due to the larger weight on the smaller within-subject effect for the weighted average effect in the longitudinal model (see Supplement: Section 3 Eqn. (14)). As shown in our analyses, fitting the between- and within-subject effects separately prevents conflating the two effects.

If the 'longitudinal' measure was a repeated measure in that it happened only a few days later, would it also be expected to reduce the effect size? Only a few days later, one would not expect a structural brain metric to have changed. So if the effect size was still reduced, would that suggest that the original cross-sectional effect size was inflated due to sampling variability, and that repeated measurement increased the accuracy of the non-brain metric, which reduced the effect size?

Response: As described above, the reduction in effect size is due to a change in the parameter, as in equation (14) from the Supplement included above. The reduction is due to variability in the covariate that is not reflected with variability in the outcome (i.e., $\beta_w = 0$). We don't expect a structural brain measure to change over a few days, but the cognitive measures may change as they may be affected by the participants' states at the visits. In this situation, the within-subject change in the brain measure (which is zero) is not related to the within-subject change in the cognitive measure, which means the within-subject effect of the covariate has to be zero because the brain is not changing. Based on our statistical theory (included in our response to your previous comment), we would expect the effect size parameter to be reduced in this case as well: this way of operating the design, i.e., repeatedly measuring a structural brain measure that is not expected to change in a short time and correlating it to the within-

subject change of a cognitive measure that may change quickly within a short time, forces the within-subject effect to be zero and increases its weight in the average effect size parameter.

Associating structural brain metrics with non-brain variables that are not state dependent (e.g. handedness) does not show this effect, neither does comparing functional brain metrics (state dependent) to state dependent cognitive measures. Is it possible/likely that functional brain and cognitive measures collected close together in time, as in most studies, are subject to similar state effects, so that anxious during scanning and anxious during cognitive testing at timepoint 1 and happy during scanning and happy during cognitive testing at timepoint 2 allow for larger longitudinal BWAS effects? Is the issue with forcing a longitudinal analysis onto an association that is devoid of a true longitudinal relationship?

Response: Yes, this is a great example. On a short time scale, structural brain measures are effectively stable. If the structural brain measures and cognitive measurements were taken over a longer time scale, we might expect within-subject changes in cognitive testing to be associated with within-subject changes in gray matter. We've reviewed the terminology in the last paragraph of Section "**Preferred sampling schemes and longitudinal designs depend on state versus trait associations between brain and non-brain measures**" to make sure that the conclusion you describe here was clear in the text.

For example, in the case of cognitive testing scores, which are supposed to be stable traits, but are also variable day-to-day based on whether the child had their Wheaties in the morning, etc, should we refrain from correlating them against structural metrics that are not variable on such rapid timescales. Or put differently, is it advisable to correlate structural brain metrics which are state invariant against cognitive measures that are state dependent. For example, total gray matter volume is independent of whether someone has a 10/10 tension headache, but testing performance is not. If we now sample someone across the state space, either rapidly (repeated sampling) or with a delay (longitudinal), the state-driven variance in the non-brain measure can drive down the effect size. But which effect size is closer to the truth? Maybe longitudinal sampling with a u-shaped sampling scheme is always superior, but it does not always enlarge the effect size.

Response: That is a good conceptualization of what is happening – within-subject variability in the covariate is not related to the outcome. In practice, we might not know, in general, when that is the case or not. For example, over larger time scales, GMV and cognitive tests may be sensitive to slowly varying trait-related changes. Instead of refraining from modeling associations between particular imaging and non-imaging variables, we instead recommend explicitly modeling between- and within-subject effects in longitudinal data. This can be done by modeling the baseline cognitive measure as well as the change from the baseline value at each time point as covariates (an example of how to do this is now added to Supplement: Section 3.2). This approach retains power for the between-subject effect, but still allows exploration of the within-subject effect. Similar to the response to comment R1.3, when there are different between- and within-subject effects but we incorrectly assume they are equal, the cross-sectional and longitudinal models are targeting different combinations of the between- and

within-subject effects, but neither is “correct”. In practice, if investigators choose to estimate a single parameter by assuming between- and within-subject associations are equal, then we recommend targeting the averaged parameter that the cross-sectional model is targeting, so that there is consistency between cross-sectional and longitudinal models (see responses to R1.3 and Editor’s comments and Supplement: Section 3.3).

R 1.4: Overall, Kang et al., convincingly argue several points important for optimizing study designs, but the framing of the arguments could perhaps be refined further. The results themselves seem to demonstrate that there are fundamental differences between studying the effects of age on the brain, which this group of authors has done extremely successfully and relating other variables to the brain. Similarly, there seem to be differences between structural and functional brain metrics and state-dependent/independent non-brain metrics. Instead of trying to throw these different types of comparisons into the same pot, it might be more informative to highlight these differences and to then present the results in terms of recommendations that are specific to certain analyses vs. recommendations that always apply. For example, it seems that increasing between-participant variability by u-shaped sampling is always a good idea unless your goal is to represent the general population?

Response: We’ve added two flow charts that delineate design and analysis choices based on particular features of the data. The first chart provides recommendations for modified sampling based on the goals of the study (e.g. unbiased estimation of population effect sizes) and the second provides design and analysis recommendations based on features of the data. See our response to the editor’s comments above for details. In the *Conclusion* subsection of the Discussion, we refer to general recommendations:

It is difficult to provide general recommendations for design and analysis that apply to all studies, and study design must consider other features in addition to optimizing ESs, such as practical challenges in collecting longitudinal data. Nevertheless, the present study provides guidelines for designing and analyzing BWAS for optimal ESs (Fig. S13 and S14), based on our empirical and theoretical results. Although the decision for a particular design or analysis strategy will depend on unknown features of the brain and non-brain measures and their association, these characteristics can be evaluated in pilot data or the analysis dataset (Fig. S12; Supplement: Section 3.1). Broadly, increasing the covariate SD in the study population has a deterministic increase in standardized ESs for most associations so is a practical approach, but the benefit of a longitudinal design is dependent on the form of the association between the covariate and outcome. Our findings highlight the importance of rigorous evaluation of statistical designs and analysis methods to improve power and replicability of BWAS.

Fig. S13 and S14 provide the recommendations with references to supporting information throughout the paper and Supplement. As you describe, increasing between-subject variability seems to be beneficial in the majority of associations unless the goal is to estimate population distributions with high accuracy, so we included a sentence to emphasize this as well in the

Conclusion subsection of the Discussion, “Broadly, increasing the covariate SD in the study population has a deterministic increase in standardized ESs for most associations so is a practical approach, but the benefit of a longitudinal design is dependent on the form of the association between the covariate and outcome.”

We also describe in the Discussion subsection “*Increasing between-subject variability to increase effect sizes*,” and in response to comment R1.2, that sampling to obtain a U-shaped distribution is not required to increase variance as random sampling can sometimes generate bell-shaped or skewed distributions. In these cases, sampling to obtain a uniform distribution will still increase covariate variance and will be lower risk with regards to balancing bias of effect size parameters, estimation of population distributions, and concerns about societal consequences due to sensitive confounders. Finally, we realized that we had not drawn a distinction between unstandardized effects used for the biological interpretation of population effects and standardized ES. The ESs we talk about in this study are standardized ES parameters, which are unitless, directly related to replicability, and affected by our sample schemes. When people interpret the population effects, they often mean the unstandardized effects (e.g., regression coefficient, beta), these are not affected by our sampling schemes under correctly specified mean models. We now have added a statement in the Discussion to emphasize this.

Unstandardized effects (such as regression coefficients), which are often related to the biological interpretation of the population, are not affected by the sampling modifications used here and can be unbiasedly estimated using correctly specified models⁴¹.

R 1.5: The manuscript makes important and valid points about study designs. Therefore, it does not need to employ rhetorical constructions that could be taken as encouragement to circumvent sample-size requirements (last sentence of the abstract). Instead, the manuscript would be further strengthened, if it focused on highlighting the additional power to be gained from pairing larger sample sizes with optimized (u-shaped) sampling schemes. In contrast to sampling schemes, which seem to have some specific dependencies, larger samples universally improve replicability and generalizability of BWAS results.

Response: We’ve changed the last sentence to focus on the benefit of sampling schemes.

Specific points:

Title:

The term ‘longitudinal BWAS’ clashes with the original definition of BWAS as cross-sectional.

Response: To keep the context of our paper within the recent discussion on effect sizes we’ve retained this terminology, but now explicitly define longitudinal BWAS in the introduction

In particular, modifications to the sampling scheme and implementing longitudinal designs might improve statistical power for associations of brain and cognitive or

biological measurements^{1,16–18}. Further, longitudinal BWAS, which utilize repeated measurements from the same individuals, can study intra-individual associations.

Abstract:

The word 'detrimental' is very strong.

Response: We've changed this to "have smaller" in the abstract and throughout the paper.

Intro:

Setting up the importance of BWAS as relevant to diagnostics and prognostics, seems slightly off the mark. It seems more relevant to epidemiology since diagnostics and prognostics require high sensitivity and specificity that allow clinicians to alter care for a specific patient.

Response: We've changed this sentence to "In brain disorder research, the fundamental goal of BWAS is to identify individual differences that improve our understanding of how brain organization and function may be altered in clinical conditions."

Is a whole brain metric still a brain-wide association since it cannot be expressed as a map. Is it more or a brain-average association?

Response: This variable was included in Marek et al. (2022) and is consistent with their definition of BWAS "as 'studies of the associations between common inter-individual variability in human brain structure/function and cognition or psychiatric symptomatology'."

Figures: It would be helpful to also embed them in the text for easier reviewing

Response: We've inserted the main figures into the manuscript text.

Fig. 5. Why does decreased effect size with u-shaped within-subject sampling suggest a decrease in power, not a correction of inflated effect sizes?

Response: As described in our response to R1.3 and R1.3.2, the effect size literature often conflates effect size parameters and estimates. Throughout the paper, we are primarily describing effect size parameters, and for this part of our study, we are demonstrating the effect size parameters change with study design features, and the effect size parameters are directly related to power. For this reason, no effect size estimate is inflated in the sense of bias, because it represents an effect size parameter for a regression coefficient given a data and analysis model (regardless of whether the model is correctly specified for the data or not).

Fig. 6d. The functional connectivity effect sizes seem extremely small. The thick black bar is at zero for all of them, is that correct?

Response: Yes, it is correct and aligns with the results from Marek et al. (2022) as it is a mixture of null and non-null associations.

For supplemental figure captions, please repeat the key information each time. The captions should be sensical, even if read out of order. For example Fig. S10 caption does not spell out what 1, 2+ on the x-axis means.

Response: We've improved the captions of the supplementary figures accordingly.

Referee #2 (Remarks to the Author):

The authors have provided a comprehensive response to the reviewers, with new material and substantial revision. Overall, I am persuaded that the paper makes a useful contribution to the broad discussion on BWAS effect sizes and what brain imaging can and cannot reveal about behavior.

A particularly interesting feature that has come out of this analysis is that between-person associations and within-person associations between brain and behavior have different effect sizes, and that common modeling strategies (e.g., LMMs, GEEs) average over them.

Thus, longitudinal designs are less efficient in many cases. In fact, they are beneficial mainly in tracking age, but not for other behavioral variables (like cognitive performance scores on measures from the NIH toolbox). This implies that within-person variations in structure (gray matter) track age, but do not track changes in other performance measures well. Within-person variations in functional connectivity track performance a bit better, which is suggestive, though it's unclear if this difference is reliable.

Pointing this out, and developing the rationale and explanation more completely, allays my previous concern that the basic findings on effects of sampling and predictor variance are well known from basic statistical theory. The authors argue that (1) real-world data may deviate from predictions made by normative theory for a number of reasons, and there is value in presenting results from actual datasets; and (2) the distinction between within- and between-person effect sizes and its implications for study design are not well understood. I am persuaded on both of these points. I also think that the idea of separating between- and within-person effects is important and under appreciated, and this paper will inform the next round of large-scale studies of brain function and structure.

Response: Thanks for this positive appraisal of this work.

R 2.1: There are some remaining features of the presentation that make it difficult to interpret the results in some cases. This list is not exhaustive, but may help in tightening all the descriptions so that the results are interpretable and usable by future readers.

For example:

Fig 5: Boxplots reflect bootstrapped associations, but how was region handled in regional GMV and CT analyses? Are these effect sizes over randomly selected regions?

Response: Sorry for the confusion. For each association between a regional outcome and a covariate, the *mean* effect size or power estimates across the 1,000 bootstraps were calculated under given sampling scheme(s), and each boxplot represents the distribution of these mean effect size estimates of the covariate across *all the regions*. We've added descriptions to the figure captions.

Tab. S13: What are the intra-subject correlations and how are they calculated? (They are not mentioned in the Note)

Response: We've added this description to the table caption. They are estimated from GEEs with exchangeable correlation structure, which are used for the analyses.

When RESI is explained, there is a clear expression for S, but the expression for S-hat uses T^2 , without clearly connecting T^2 to S. Thus, many readers will not be able to follow this without outside references.

RESI is not explicitly defined with respect to S-hat, though it is implied that $RESI = S-hat$ (and this is only made explicit in a different section of the methods). S-hat is also multivariate and depends on all the beta-hats in a model, and it's not made explicit how the effect sizes for separate terms (or sets of terms) in the model are calculated.

Response: We've clarified these points by explaining what the letter S denotes and adding the equation for T^2 .

The paper (Fig. 6) refers to "methods for estimating between-subject and within-subject associations separately", but no reference or document explaining what these are is mentioned here. Presumably this is explained in the new technical supplement Section 2.2 and 2.4, but it's not clear.

Response: This can be done in several ways such as including the covariate for each time point as well as the mean value of the covariate across all time points, which will be identical within-subject. We've added references to Methods and Supplement, and example R code as

well in the Supplement: Section 3.2:

the correct model. To fit this model, we include the original covariate x_{ij} which varies over measurements from the same individual, and also include a covariate that is the mean of the original covariate $\bar{x}_i = n_i^{-1} \sum_{j=1}^n x_{ij}$, as in equation (11). The mean covariate will be the same value for all measurements within participants. A LMM or GEE can be used to model and estimate the parameters separately. The parameters can be interpreted separately as trait versus state associations with the outcome. Given this, as one might expect, functional connectivity measures had larger average state associations than structural measures did in the main paper (Fig. S11).

Referee #4 (Remarks to the Author):

The manuscript used a meta-analysis of robust effect size index (RESI) to demonstrate that enlarge the covariate variation and/or switching a cross-sectional study to longitudinal can improve the standardized effect size. First, I am not convinced that “standardized effect size” is a quantity that can be improved via study design. Second, “standardized effect size” is not the only and major concern of deciding whether a study should be longitudinal. Third, the conclusions were mainly derived from studying the association between imaging features and age which is not generalizable. The following are my comments.

Major Concerns:

- 1) *My understanding of effect size should be an underlying feature of a biological process, rather than a quantity that can be “improved”. I am confused about the authors’ study objective here. Perhaps, “effect size” is a wrong term of the quantity the authors tried to improve here.*

Response: Generally speaking, effect size indices are a quantitative reflection of the magnitude of an association’s strength – they are parameters. Many standardized effect sizes rely on the sample/study features. For example, a common standardized effect size index is Pearson’s correlation coefficient between an outcome Y and a covariate X is $\rho_{X,Y} = \beta \frac{\sigma_X}{\sigma_Y}$, where σ_X and σ_Y represent the standard deviation of X and Y. Although β is fixed, σ_X depends on how the sample is collected and therefore can be modified by sampling in such a way that increases the variance of X and thus, the (standardized) effect size. This can be proven mathematically when the association is known to be linear, but may not hold under nonlinear relationships or if the model is misspecified, thus the justification for us to investigate this property empirically in BWAS. Often, the unstandardized effect (such as the regression coefficient beta) is used to describe the biological interpretation. This is not affected by the sampling strategies when the model is correctly specified and is also different from the standardized effect size we are focusing on in this paper. We focus on standardized effect sizes because they are directly

related to power and replicability. We now have added emphasis on this distinction in the Introduction and Discussion.

In the Introduction:

Standardized ESs (such as Pearson's correlation r , Cohen's d , and Cohen's f) are different from unstandardized effects (such as regression coefficients) because they quantify the strength of an association on a unitless scale. Low replicability in BWAS has been attributed to a combination of small standardized ESs, small sample sizes, p -hacking, and publication bias^{1,2,9-13}. Throughout this paper, we focus on standardized ESs as these are directly related to power.

In the Discussion:

Unstandardized effects (such as regression coefficients), which are often related to the biological interpretation of the population, are not affected by the sampling modifications used here and can be unbiasedly estimated using correctly specified models⁴¹.

2) *In general, the purpose of conducting a cross-sectional study and a longitudinal study is different. Compared to a cross-sectional study, for a longitudinal study, the study interest may focus more on longitudinal/temporal (intrasubject) variation of the outcomes. Thus, the determination of running a cross-sectional or longitudinal study does not due to the concern of power or replicability, but more of the research question. Switching from a cross-sectional to longitudinal study, just due to the concern of improving standardized effect size, may not be feasible or applicable for many studies. On the other hand, running a longitudinal study takes longer time, may cost more and lose subjects at follow-up visits. These are also factors that should be taken into consideration during study design. The current study seems to ignore these aspects and only considers the improvement of one feature.*

Response: We agree that there are many considerations that should be taken while designing and conducting a longitudinal study and quantifying intra-individual change is an important benefit of longitudinal designs. The fixed effects parameters in typical random effects models used in longitudinal BWAS assume between- and within-subject effects are equal. This implies that intra-subject changes are not different from between-subject changes except through the random effects terms. Under this assumption, cross-sectional and longitudinal models are estimating the same parameter, thus we can evaluate their power (and effect size) for estimating this parameter as we do throughout the paper. When this assumption is violated and the longitudinal model is incorrectly specified, we show that this can lead to reductions in power due to differences in the intra-subject association between the outcome and covariate (Section "**Preferred sampling schemes and longitudinal designs depend on state versus trait associations between brain and non-brain measures**"). As requested by reviewers, we have revised the paper to provide general recommendations for designing and analyzing data to

optimize effect sizes and power. Of course, as you describe, there are many other features to consider that are likely study-specific, e.g. budget and practical challenge in retaining participants, so we emphasize this more clearly in our recommendations and that, ultimately, the decision on study design will depend on these other features as well. We've added a statement to emphasize your point in the manuscript (Section "***Increasing between-subject variability to increase effect sizes,***" in the Discussion):

The decision to pursue a particular design will depend on the costs and complexity of acquiring the non-brain covariates of interest and translational goals that may require population representative ES estimates.

and the *Conclusion* section of the Discussion:

It is difficult to provide general recommendations for design and analysis that apply to all studies, and study design must consider other features in addition to optimizing ESs, such as practical challenges in collecting longitudinal data. Nevertheless, the present study provides guidelines for designing and analyzing BWAS for optimal ESs (Fig. S13 and S14), based on our empirical and theoretical results.

- 3) *"Meta-analyses shows ESs depend on study population and design" section.*
- a) *When getting RESI and L-RESI, the authors used data from multiple cross-sectional and longitudinal studies, respectively. From my understanding, the authors did not take the variation in study protocol or imaging scanner effect into consideration. This has been demonstrated as a concern when analyzing imaging data collected from multiple sites.*

Response: We removed the site effects by implementing ComBat and LongComBat. Please see the Methods section – Statistical analysis – Removal of site effects starting on line 621 of revision 1. We've now also added a statement to the body of the paper.

To fit each study-level analysis, we regress each of the global brain measures (total GMV, total sGMV, total WMV, mean CT) and regional brain measures (regional GMV and CT, based on Desikan-Killiany parcellation³⁴) on sex and a nonlinear spline function of age in each of the 63 neuroimaging datasets from the LBCC using linear regression models for the cross-sectional datasets and generalized estimating equations (GEEs) for the longitudinal datasets (Methods). **Site effects are removed before the regressions using ComBat (Methods).**

- b) *It is not very surprising of the results that standardized ESs depend on age as the two high RESI periods are ages of neurodevelopment and neurodegeneration. And if one aims to study the association with age, for sure, one should recruit subjects on a reasonably wide age range (for example, one cannot study the association with age when the recruited subjects are all about the same age). Therefore, I am not sure how the current analysis results provide more information to a study design.*

Response: We agree that this result is to be expected given that the brain changes differentially through the lifespan as shown in our previous work as well (Bethlehem et al. 2022). The primary important result of the meta-analysis is that the standard deviation of the study is strongly associated with the effect size – studies with higher variance of age have larger effect sizes after controlling for the mean and the skewness of age in the study population (as we show in subsequent analyses, controlling the age range as well). This means, for example, if two studies are investigating age-related changes in adolescence, the study with a larger standard deviation of age will have a larger effect size and greater power (SD age | others in **Figure 1**), although we do not evaluate this claim causally until the following section, but this is still strong empirical evidence that the study design feature is associated with effect sizes.

4) *“Improved sampling schemes can increase ESs” section*

a) *Not sure what the “bell- and U-shaped” sampling distributions look like.*

Response: The sampling distributions were plotted in Fig. S6 in the previous version of the revised manuscript, and in the revised manuscript we ensure that we refer to this figure when making first reference to the sampling schemes. Fig. S6 is included below for convenience.

Fig. S6. The sampling scheme implemented in UKB. The sampling schemes adjust the variance of age in the samples by assigning heavier or lighter weights to the subjects with age at the two tails of the population. The U-shaped scheme produces the largest variability of age in the samples, followed by uniform and bell-shaped sampling schemes.

b) *What is the age range of the UKB data? If it is across lifespan, I am not very surprised that the U-shape “performs” better since my understanding of the U-shape puts higher sampling weights on the age period of neurodevelopment and neurodegeneration (the two ends of the age range).*

Response: The age range in UKB data is ~45 to ~80 years (top left panel of Fig. S1), so the U-shaped sampling scheme places the highest weights around 50 and 78 (as shown in Fig S6 pasted above).

c) *The current conclusion of the association with age may not be generalizable to the study of other factors (covariates).*

Response: You are correct that this result does not generalize. For this reason, we included the following section to investigate the effect of the different sampling schemes for other brain-covariate associations. Please see the section in the revised manuscript beginning with “To explore if the proposed sampling schemes are effective on a variety of non-brain covariates and their associations with structural and functional brain measures, we implement the sampling schemes on subjects (cognitively normal and atypical) with cross-sectional and longitudinal measurements from the ABCD dataset.”

5) *“Preferred sampling schemes and longitudinal design XXX” section*

- a) *The ABCD study subjects at baseline are mainly between 8 to 12 years old, not a wide age range. Not sure how this small age range will impact the findings. On the other hand, there are many other features that may have an impact on brain development. For example, not all subjects are typically developing children. It seems such information or individual features were not taken into consideration into the analysis.*

Response: In this section, we moved from investigating age to other non-brain measures (e.g., cognition, psychopathology, etc) to improve the generalizability of our findings. Some of the variables we considered measured various aspects of cognition and development (e.g., intelligence, CBCL). In this case, instead of controlling the variance of age in the sampling procedure, we controlled the variance of the *covariate* of interest (i.e., the *non-brain measures* here) to see how this changes the effect size. As shown, increasing the within-subject variance sometimes reduces effect sizes for volumetric and thickness outcomes.

Referee #5 (Remarks to the Author):

Summary of the manuscript:

The abbreviations used here are ES - effect size, CD - cross-sectional design or cross-sectional study, and LD - longitudinal design.

The authors have leveraged data from the Lifespan Brain Chart Consortium to explore the impact of study design on effect sizes (ES) in both cross-sectional (CD) and longitudinal (LD) designs within Brain-wide Association Studies (BWAS). Their comprehensive analysis, bolstered by a meta-analysis, seeks to clarify how ES correlates with replicability across BWAS. The manuscript’s strengths are manifold, including (1) the use of a rich dataset enabling detailed examination of design elements such as sample size, frequency of imaging scans per participant, and the derivation of global and regional brain metrics, while also including covariates like age, sex, cognitive abilities, psychopathology, and demographics; (2) R codes with workflows to implement similar analysis for other BWAS studies on optimizing ES.

The manuscript’s current analysis and findings present a compelling case for publication in Nature, contingent upon a detailed revision of the workflow and result descriptions. It would be particularly beneficial for the authors to outline their recommendations step-wise in the

Conclusion section, especially when deciding between CD and LD for Brain-wide Association Studies (BWAS) based on specific responses and covariates. Clarification on the choice between CD and LD should be accompanied by a discussion on design aspects such as the number of participants, the frequency of sample collection per participant, and the total number of samples collected, with a particular focus on the effect size (ES) concerning one or multiple covariates. An analysis of the time intervals between repeated samples for each subject should also be included to provide insights into the optimal study design. However, I understand it is not reasonable to ask it now. I'd suggest caution be exercised in making definitive statements about the lesser superiority of longitudinal designs in enhancing ES, a concern previously highlighted by reviewer R2 without including time interval analysis. Addressing these recommendations can significantly improve the manuscript's contribution to BWAS studies.

Response: Thank you for these suggested edits. We've now included two decision trees to guide designing and analyzing BWAS (Fig. S13 and S14) and provide recommendations in the body of the paper. Please see our response to the editor above. Briefly, we provide recommendations on whether to use a random or modified sampling scheme based on the goals of the study. We also provide another decision tree to assist in choosing what type of sampling scheme and analysis model to use based on the nature of the true association. As the nature of the true association is often unknown, we describe procedures to evaluate these characteristics using pilot data (for study design) or visualizations in the analysis dataset.

For the age analyses, the time interval between measurements is controlled by the within-subject sampling variance – larger within-subject sampling variance will have measurements that are further apart on average. These analyses indicate that longer intra-subject periods will increase effect sizes for age. We've included a comment describing these results with respect to the time interval between measurements, "Together, these results suggest having larger spacing in between- and within-subject age measurements increases ES and power."

Considering the time between psychometric measurements is also an interesting feature to consider. The difference in between- and within-subject associations observed for the structural brain outcomes (e.g., volumetric and thickness measures) is likely dependent on the time frame of the repeated within-subject measurements. Because structural brain outcomes are relatively stable over the short time span (2 years) but the psychometric measurements are relatively noisy, we saw small within-subject effects. Over a longer time scale, it is possible that changes in both structural brain outcomes and psychometric measurements are due to long-term "trait"-related effects and the between- and within-subject effects will be more similar. Unfortunately, because of the tight age range of within-subject measurements in the ABCD, we are not able to investigate how the within-subject associations change over longer periods of time, but this highlights the importance of the continued acquisition of the longitudinal ABCD data.

Major concerns:

1. *The findings presented in the abstract appear to be misaligned with the datasets and variables detailed in the manuscript, necessitating a thorough revision of the abstract to*

accurately reflect the manuscript's content. Specifically, the manuscript illustrates how ES can be improved by adding an additional longitudinal measurement per subject within the ADNI dataset (line 247), a detail generalized in the abstract to encompass the LBCC dataset ((line 51)). To enhance clarity and coherence, I recommend revising the abstract to accurately represent these findings, ensuring it aligns with the specific datasets and variables analyzed in the study. This adjustment will not only improve the abstract's precision but also ensure readers gain an accurate overview of the study's conclusions.

Response: Thank you for pointing out this inconsistency; we've made your recommended edits to correct the abstract.

...Here, we perform analyses and meta-analyses of a robust effect size index (RESI) using 63 longitudinal and cross-sectional magnetic resonance imaging studies from **the Lifespan Brain Chart Consortium** (77,695 total scans) to demonstrate that optimizing study design is critical for improving standardized effect sizes and replicability in BWAS. A meta-analysis of brain volume associations with age indicates that BWAS with larger covariate variance have larger effect size estimates and that the longitudinal studies we examined have systematically larger standardized effect sizes than cross-sectional studies. We propose a cross-sectional RESI to adjust for the systematic difference in effect sizes between cross-sectional and longitudinal studies that allows investigators to quantify the benefit of conducting their study longitudinally. Analyzing age effects on global and regional brain measures **from the United Kingdom Biobank and the Alzheimer's Disease Neuroimaging Initiative**, we show that modifying longitudinal study design through sampling schemes to increase between-subject variability and adding a single additional longitudinal measurement per subject can improve effect sizes....

- 2. The manuscript needs clarification regarding the categorization of univariate and multivariate analyses in the context of Brain-wide Association Studies (BWAS). Specifically, the study designates analyses involving only age as a covariate as univariate, aligning with simple linear regression principles. In contrast, analyses incorporating multiple covariates are labeled as multivariate models akin to multiple linear regression. The confusion is compounded in line 377, where the manuscript describes multivariate BWAS by inverting the roles of response and covariates, treating brain measures as imaging variables while seemingly assuming psychometric or demographic measurements as the response in univariate analyses. The multivariate approach could be considered when predicting multiple responses, allowing for the decomposition of effect sizes (ES) of various brain measures on multiple responses. Clarification is needed to differentiate between univariate and multivariate BWAS accurately, ensuring a clear understanding of when and how each analysis type is applied, particularly in terms of response and covariate roles. This distinction is crucial for accurately interpreting the study's methodology and findings.*

Response: We thank the reviewer and believe this confusion was caused by differences in conventions of terminology in neuroscience and Statistics. The term “Multivariate BWAS” has been used in neuroscience to refer to BWAS that use machine learning with high-dimensional imaging variables as predictors, whereas multivariate models in Statistics refer to those where there are multiple outcome variables (e.g., different brain regions). We now refer to multivariate or univariate BWAS and avoid saying multivariate/univariate methods throughout the entire paper (i.e., we avoid using the convention from Statistics).

3. The organization and clarity regarding the dataset and RESI descriptions within the manuscript require enhancement for better comprehension and flow. Notably, there are instances of redundancy and fragmentation in presenting information about the datasets and RESI, which could potentially confuse readers. For example, identical details about the datasets are mentioned in both lines 91 and 125, as well as in lines 123-127 and 136-138, leading to unnecessary repetition. Furthermore, the explanation of CS-RESI is split between lines 102 and 182, with the latter providing crucial understanding that could be more effectively communicated if introduced in a consolidated manner earlier in the manuscript. To resolve these issues, I recommend a thorough reorganization of the manuscript to eliminate repetitive information and ensure that concepts like CS-RESI are fully explained at their initial mention. This will not only streamline the content but also facilitate a smoother, more logical progression of ideas for the reader.

Response: Thanks for these recommendations that improve the clarity of the paper. We’ve made these edits to reduce the repetitive and scattered information about datasets and explain the CS-RESI more clearly in the introduction where we introduce the RESI:

Systematic differences in ESs between cross-sectional and longitudinal studies that compromise their comparability arise because the ES is dependent on study design. Throughout the present study, we use the robust effect size index (RESI)^{31–33} as a measure of standardized ES, a recently developed ES index encompassing many types of test statistics which is equal to $\frac{1}{2}$ Cohen’s *d* under some assumptions (Methods; Supplement)³¹. Because standardized ESs are dependent on whether the study design was cross-sectional or longitudinal (Supplement: Section 1.1.1), we refer to the RESI estimate in longitudinal studies as the “longitudinal RESI (L-RESI)”. We also propose the “cross-sectional RESI (CS-RESI) for longitudinal datasets” that improves the comparability of ESs between longitudinal and cross-sectional studies and allows us to quantify the benefit of using a longitudinal study design in a single dataset. The CS-RESI represents the RESI in the same study population if the longitudinal study had been conducted cross-sectionally. Comparing L-RESI to the newly developed CS-RESI allows us to quantify the benefit of using a longitudinal study design in a single dataset (see Supplement: Section 1).

4. Lines 730-731 and other places: My concern centers on the methodology used for calculating statistical power throughout the manuscript. The authors describe power calculation as the proportion of bootstrap replicates yielding p-values less than or equal to 5%. However, statistical power is traditionally defined as the probability of correctly

rejecting the null hypothesis when there is, in fact, a true effect (i.e., when the alternative hypothesis is true). The critical issue with the described approach is that it assumes all observed data inherently exhibit a non-null effect, which may not be the case. Without confirmation that the bootstrap samples are drawn from a distribution under the alternative hypothesis (i.e., one where a non-null effect is present), equating the proportion of bootstrap replicates with p -values ≤ 0.05 to statistical power is problematic. This is because such a proportion could potentially include false discoveries, given the lack of certainty regarding the presence of a true effect in the observed data. A more nuanced approach or additional clarification on how the authors account for this potential discrepancy in their power analysis would enhance the credibility and interpretability of their findings.

Response: The bootstrap analyses we perform treat the full dataset as the population, so the observed effect in these full datasets represents the true effect in the simulated population where the bootstraps represent random samples from this population. Even though the true effect size in the actual population may be zero, the “population” effect sizes in our bootstrap simulation are not exactly zero because the coefficients are nonzero in the full dataset, although they can be arbitrarily close to zero. However, in response to your comment, in the revised manuscript, we used the approach of Marek et al. (2022) and defined non-null effects as those that are significant in the full sample for the UKB and ADNI. For the UKB, this included all regional GMV-age associations except one and all CT-age associations except for one. For the ADNI data, it included all GMV-age and CT-age associations except for 1 and 4, respectively. We’ve now adjusted Figures 4 and 5 to reflect this change (previously, we had defined “null” effects as those with smaller effect sizes below the 25 percentile). The new figures are included below for convenience. We’ve also updated the figure captions and methods to reflect this change.

Fig. 3. Improved effect sizes and statistical power for age associations with different brain measures: total gray matter volume (GMV) (a-b), regional GMV (c-d), and regional cortical thickness (CT) (e-f), under three sampling schemes in the UKB study ($N = 29,031$ for total GMV and $N = 29,030$ for regional GMV and CT). The sampling schemes target different age distributions to increase the variance of age: Bell-shaped < Uniform < U-shaped (Fig. S6). Using the resampling schemes that increase age variance increases (a) the total GMV-age effect sizes, and (b) statistical power (at significance level of 0.05). The same result holds for (c-d) regional GMV and (e-f) regional CT. The curves represent the average effect size or estimated power at a given sample size and sampling scheme. The shaded areas represent the corresponding 95% confidence bands. (c-f) The bold curves are the average effect size or power across all regions with significant uncorrected effect sizes using the full UKB data.

Fig. 4. Improved effect sizes and statistical power for age associations with structural brain measures under different longitudinal sampling schemes in the ADNI data. Three different sampling schemes (Fig. S8) are implemented in bootstrap analyses to modify the distributions of baseline age and the age change from baseline and therefore to modify the between- and within-subject variability of age. **(a)** Implementing the sampling schemes results in higher (between- and/or within-subject) variability and increases the total gray matter volume (GMV)-age effect size and **(b)** statistical power at significance level of 0.05. The curves represent the average effect size or estimated power and the gray areas are the 95% confidence bands across the 1,000 bootstraps. **(c-d)** Increasing the number of measurements from one to two per subject provides the most benefit on **(c)** the total GMV-age effect size and **(d)** statistical power when using uniform between- and within-subject sampling schemes and $N = 30$. The points represent the mean effect size or estimated power and the whiskers are the 95% confidence intervals. **(e-h)** Improved effect sizes and power for age associations with regional **(e-f)** GMV and **(g-h)** cortical thickness (CT), across all brain regions under different sampling schemes. The bold curves are the average effect size or estimated power **across all regions with significant uncorrected effect sizes using the full ADNI data**. The gray areas are the corresponding 95% confidence bands. Increasing the between- and within-subject variability of age by implementing different sampling schemes can increase the effect size of age, and the associated statistical power, on regional GMV and regional CT.

Some of the minor suggestions for improving the text are as follows.

- 1. Lines 41 - 43 and 50 -52: I assume both lines are about the Lifespan Brain Chart Consortium (LBCC), so why don't you mention this in lines 41-43 instead of specifying the data source in line 50?*

We have made the change accordingly.

- 2. Lines 41-42: Is it CS RESI? or just RESI?*

The RESI here collectively means all versions proposed in the paper: the RESI in cross-sectional studies and the CS-RESI and L-RESI applied longitudinal studies.

- 3. Line 99: independent variable - I suggest either covariate or predictors instead of independent variable. In line 109, the authors used covariates. So, there needs to be more consistency in terminologies.*

We have made the edits and now consistently use “covariate(s)” throughout the manuscript.

- 4. Lines102-105: mentioned that CS-RESI improves the comparability of ESs between LD and CD, but there is no reference to this.*

We have moved the description of the comparability of CS-RESI to the introduction as you suggested above, and added the reference (to Supplement: Section 1).

- 5. Lines 105 - 107: This might be for non-biologists - what is the sampling scheme in the UKB and ADNI? Are they related to how to sample MRI scans or sampling individuals based on predictors?*

We've edited this sentence to be clear we mean our sampling schemes based on the covariate to increase variance.

- 6. Lines 109 - 111: Is ABCD a subset of the LBCC? If not, why do you choose ABCD to generalize the findings on GMV-age association? Moreover, generalization is many folds. Is it here for each response (brain measure) and one covariate?*

ABCD is part of LBCC but it contains many other brain and non-brain measurements that are not available in other studies of LBCC, such as functional connectivity (a functional brain measure), NIH Toolbox (a cognitive measure), and CBCL (a psychopathological measure). As we needed to generalize our findings on other types of associations between brain and non-brain variables, we chose ABCD. Here we looked at the combinations of each response and a covariate.

7. *Inconsistent usage of “brain outcome measure”, “brain measure”, and “brain outcome”. I’d use one of the terms throughout the manuscript. For example, lines 111, 50, 54, etc....*

Brain measures are the outcomes throughout our manuscript. We now have made sure we use “brain measures(s)” throughout the manuscript.

8. *Lines 113- 114: What are the common longitudinal models conflating between and within-subject effect? Did you consider modeling both cross-sectional and longitudinal effect (adjusted for cross-sectional) in the common model?*

The common longitudinal models use one parameter to model the between- and within-subject effects. The concluding sentence of the introduction describes the approach of modeling the between- and within-subject effects separately, as you describe in your second question. Details of how we fit these models are now included in Supplement: Section 3.2.

3.2 Modeling different between- and within-subject effects

When the between- and within-subject effects are different it is probably best to model them separately because it avoids averaging the larger effect with the smaller effect, and because the distinction between these two effects can inform our understanding of brain-behavior associations. In this case, increasing between- or within-subject variance independently increases the effect size for the between- or within-subject parameter when (11) is the correct model. To fit this model, we include the original covariate x_{ij} which varies over measurements from the same individual, and also include a covariate that is the mean of the original covariate $\bar{x}_i = n_i^{-1} \sum_{j=1}^n x_{ij}$, as in equation (11). The mean covariate will be the same value for all measurements within participants. A LMM or GEE can be used to model and estimate the parameters separately. The parameters can be interpreted separately as trait versus state associations with the outcome. Given this, as one might expect, functional connectivity measures had larger average state associations than structural measures did in the main paper (Fig. S11).

Various parameterizations are possible. They look similar, but differ in their interpretation and mathematical formulation. For example, an alternative to (11) models a separate effect for the baseline measurements that influences all future measurements.

$$Y_{ij} = \beta_0^* + \beta_b^* x_{i1} + \beta_w^* (x_{ij} - x_{i1}) + \epsilon_{ij}, \quad (16)$$

which might make sense if the first measurement in the study represents a particular epoch that is comparable across participants.

9. *Lines 119-123: To fit each study level analysis, “....spline function of age and sex in 63 neuroimaging datasets from the LBCC....”. This line needs to be clarified. Did you regress each response (total GMV, total sGMV,...) on each covariate (age, sex) in each of the 63 studies separately? What is the exponential family used in the generalized linear regression? What are the model assumptions for GEE? Does the response follow any particular exponential family?*

We regressed each response on age *and* sex in each of the 63 studies. We edited the sentences to make them clearer. We used the identify linkage function in the GEEs, and assumed the mean model is correctly specified. It makes no assumptions about the error nor the outcome distributions, and we assumed the correlation structure within clusters/subjects to be exchangeable. Further details are provided in the Methods section and we've added a parenthetical reference to that section:

To fit each study-level analysis, we regress each of the global brain measures (total GMV, total sGMV, total WMV, mean CT) and regional brain measures (regional GMV and CT, based on Desikan-Killiany parcellation³⁴) on sex and a nonlinear spline function of age in each of the 63 neuroimaging datasets from the LBCC using linear regression models for the cross-sectional datasets and generalized estimating equations (GEEs) for the longitudinal datasets (Methods).

The Methods section reads:

The mean model was specified as below after ComBat/LongComBat:

$$y_{ij} \sim ns(age_{ij}, df = 2) + sex_i,$$

where y_{ij} was taken to be a global brain measure (i.e., total GMV, WMV, sGMV or mean CT) or regional brain measures (i.e., regional GMV or CT) at the j -th visit from the subject i , and $j = 1$ for cross-sectional datasets. The age effect was estimated with natural cubic splines with two degrees of freedom, which means that there were two boundary knots and one interval knot placed at the median of the covariate age. **For the GEEs, we used an exchangeable correlation structure as the working structure and identity linkage function. The model assumes the mean is correctly specified, but makes no assumption about the error distribution.** The GEEs were fit with the geepack package⁶³ in R.

10. *I'd suggest describing the dataset used in the sub-analysis with respect to LBCC. I understand there are 63 neuroimaging datasets from the LBCC. Then, I get lost in the subsets. For example, lines 121 - 125 states there are 47 CDs and 16 LDs containing 77,695 MRI scans from 60,900 CN individuals. Then, lines 126 - 127 mention 43 datasets (30 CD and 13 LD) for the mean CT and regional measures. Are these dataset subsets of 63 datasets, and are there any overlaps? Tables S1 and S2 give answers to the questions, but the reader will benefit if you refine the dataset description in lines 119 - 12. Then, write about the RESE in lines 128 - 135 in the next paragraph.*

Thanks for this suggestion. We have made edits to the paragraphs to improve the clarity and moved the description about RESI and its modification to longitudinal design (i.e., CS-RESI) to the introduction, as you suggested in your comments below and above.

11. Lines 102-104 and 128 - 135 Aren't they about RESI and its modification to LD? I think describing the RESI in lines 102 - 104 is better. Again, lines 181-182 are about defining and evaluating CS-RESI, and we need to see the details in the Supp Section 1. Why can't all these be summarized in a section and refer to the details in the Supplement?

We have moved all of the descriptions about RESI and CS-RESI (i.e., its modification to longitudinal designs) to the introduction, and also referred to the Supplement. Please see the introduction with track changes.

12. Line 15: Is it study population SD and skewness of age covariate?

We believe you meant line 157. Yes, and we have clarified we meant SD and skewness of age covariate.

13. Line 159: What does "respectively" mean?

As there are 4 global brain measures (i.e., total GMV, sGMV, WMV, and CT), we mean we showed an effect of study designs on the effect sizes of age on each of these 4 global brain measures, respectively. We have changed the sentence to the following:

"Finally, the meta-analyses also show a moderate effect of study design on the RESI of age on **each of** the global brain measures (Fig. 1a-d; Tab. S3A; Tab. S4A, Tab. S5A, and Tab. S6A)"

14. Another inconsistency in using study design, study population, study population features, study features, and study design features. I'd use terms consistently throughout the manuscript.

We have made sure we now use "study design features" throughout the manuscript.

15. Line 188: Is it 1000 bootstraps per dataset (16 LD)?

Yes. Now we clarified that it's 1000 bootstraps per sample size in each of the longitudinal datasets:

To demonstrate the difference in ES by study design and validate the CS-RESI estimator, we use non-parametric bootstrapping to create longitudinal bootstrap samples by randomly sampling subjects with replacement (1,000 bootstraps per sample size) in **each of** the 16 longitudinal neuroimaging datasets.

16. Line 189: Why the bootstrap samples are longitudinal subsamples? I thought they were bootstrap resamples. Aren't they with the same number of subjects and possibly a different number of total samples?

They are bootstrap resamples and their number of subjects is lower than the original sample size (e.g., the x-axis of Fig 4a), so we called them subsamples. We realized this is not necessary and have changed it to “longitudinal bootstrap resamples” to avoid possible confusion.

17. Lines 204: Fig. 2b doesn't describe N. Is this asymptotic related to the increasing number of individuals, the number of repeated samples per individual, or both?

We have added a description of sample size N to the caption of Fig 2. The asymptotic property is related to increasing the sample size (i.e., the number of individuals). The number of repeated samples (bootstraps) is 1,000 for each value of N .

18. Line 206: ESs from all studies using the CS-RESIs in longitudinal studies.. can it be ESs from all longitudinal studies using the CS-RESIs?

We meant we repeated the meta-analysis and used CS-RESI to represent the effect sizes in longitudinal studies. We made the following changes in hopes to make it clearer:

“As a further comparison, we repeat the meta-analyses above on the ESs from all studies **but now use** the CS-RESIs **to represent the ESs** in the longitudinal studies, demonstrating a reduction in the design effect (Tab. S3B, S4B, S5B, and S6B).”

19. Line 221: “...schemes that sequentially increase...” is it consequently increase?

We meant the three sample schemes (bell-shaped, uniform and U-shaped schemes) increase the covariate variability in sequential order, with the bell-shaped one providing the lowest variability among these three. We have removed the word “sequentially”.

20. Line 226 & Fig 3 b.: How did you calculate the power in detecting age ES?

Please see our response to your major concern #4.

21. Line 237: the study, not “a study”.

We left this as a study – referring to an arbitrary study collecting similar data.

22. Lines 240 - 241: Is there a reference to show L-RESI and power relationship?

Power for the L-RESI is based on the test statistic from the GEE. L-RESI within the GEE is a special case of the RESI, except where multiple measurements are taken on each individual.

Power curves were given in Figure 3 Vandekar , Tao, Blume (2020)⁷. We've added this reference in the methods section so that readers can view the power curves there.

Assuming the statistic from the GEE approximately follows an F-distribution with cumulative distribution function $F()$, the power can be computed $F(F^{-1}(1 - \alpha; df, rdf); df, rdf, \lambda = N \times S)$, where df is the numerator degrees of freedom, rdf is the denominator degrees of freedom, S is the L-RESI and N is the number of independent samples. This is also indicated by the equation in Methods: *Sample size calculation for a target power with a given effect size*, but that function is an estimator for power with the mean L-RESI across the bootstrap samples plugged in.

23. Lines 246 -250: Is this investigation on the ADNI dataset?

Yes, we now have clarified this in the text.

24. Line 260: is FC the functional?

Yes, FC means functional connectivity. We have now added the definition of this acronym on its first use.

25. Line 266: ...non-brain covariate? In line 262, the authors used "non-brain covariates". Then, why, in line 266, did they change the same to "non-brain measure"?

We used the non-brain measures as the covariates. We now have made sure that we call them non-brain covariates throughout to emphasize we are using them as the independent variable in the regressions, except where we refer to "brain and non-brain measures" together.

26. Line 275: It is unclear what is "for each data type"?

The different brain outcomes were considered to be different types of brain measures, for example, total/regional GMV and the mean/regional cortical thickness are considered structural brain measures, and functional connectivity is considered functional brain measures. We have changed the sentence as follows:

To confirm this, we compare the ESs of the non-brain covariates **on each type of brain measure** estimated using only the baseline measurements to the ESs estimated using the full longitudinal data (Fig. 6a-d).

27. Lines 279-280: "...than conducting longitudinal analyses using the full longitudinal data". Here, what are the longitudinal analyses? Is it ES analysis using the longitudinal data?

Yes, here we are comparing the ES estimated in cross-sectional analyses (which only use the baseline of ABCD data) with the ES estimated in longitudinal analyses (which use the full longitudinal ABCD data).

28. Line 282: Are these random intercepts individual-specific?

Yes, we have added this point to make it clearer in the text.

"Identical results are shown using linear mixed models (LMMs) with **individual-specific** random intercepts, which are commonly used in BWAS (Fig. S10)."

29. Lines 283 - 285: I'd not conclude that the longitudinal designs can be detrimental to ESs because the frequency of time points would affect ESs. So I'd remove "and, counter-intuitively, longitudinal design can be detrimental to ESs".

We have changed the word "detrimental" and now describe it as it reduces the ESs.

30. Lines 300-301: fitting these associations separately,... is separate indicating CD on the baseline samples and LD on the repeated samples or CD on all repeated samples but assuming all repeated samples are independent?

Separately refers to using two covariates, one for the mean value of the covariate and one for the time-specific value of the covariate. The mean value captures between-subject effects and the time-specific variable captures within-subject effects. We've now added some R code and output in the Supplement: Section 3.2 to help clarify how to fit this model.

31. Discussion section: Most of the above comments apply to the discussion section. In addition, why it is, in lines 375-386, imaging variables instead of brain measures?

We've corrected these inconsistencies throughout the discussion.

32. Figures labels: Why are the labels a light grey color (very hard to read)? Can't they be black? I haven't gone through the captions for the figures and tables, but there might be revisions needed based on the answers to the above concerns.

We have changed the color of the labels to black and made sure the content in the captions matches our revision.

33. Line 555: repeated based on twice

We have it corrected.

34. Line 557: is ADNI part of the LBCC? It needs to be clarified in this sentence.

Yes, it is. Now we have edited the text to make it clearer.

35. Methods: Statistical analysis:

(a) In all model specifications, I can't see the model assumptions. What is the distribution of Y_{ij} ?

We used the identity linkage function in all models. We've added this in the description as noted in response to comment R5.9.

(b) Line 632: with two degrees of freedom?

As you suggested below as well, we have clarified the specification of natural splines in terms of the number and placements of knots, and degrees of freedom throughout the manuscript.

(c) Line 658: Where is the supplement? I have figures and tables in the supplement.

Those are supplementary figures and tables. There is a separate file whose title is "Supplement to ...(the title of the main manuscript)", which might be merged into the main document for review with the main manuscript, supplementary figures and tables, etc.

(d) Line 659: We provided means – is it introduced in this paper?

We've changed the wording to "In previous work, we defined ..."

(e) Line 660: It must be

$$\hat{S} = \left(\max\left\{0, \frac{T^2 - m}{N}\right\} \right)^{1/2}$$

note that {} used for set notation.

Thank you, we have corrected it.

(f) Line 661: Where - start with a lowercase letter for “where” ... and why is C uppercase in Chi? What is the chi-square test statistic to test the hypothesis on the location parameter β ?

We have corrected them and now added a formula for the chi-square test statistics.

(g) Line 669: ...bootstrap samples. I’m afraid that the sampling subject doesn’t account for an appropriate dependence structure within the individual for a longitudinal design.

We conducted simulation studies to show that the bootstrap CIs (by sampling the subjects) can provide nominal coverage. Please see the Section 1.2 in the Supplement.

(h) I have concern throughout about the GEE. GEE is one of the ways to estimate the parameters in the model. So, I wonder what model the longitudinal data analysis used. Did authors assume the response follows any exponential family? Gaussian (possibly no for a non-negative response, log-normal? or gamma?

We specified the mean models as linear regression models and used GEE to estimate them with the longitudinal data. We used the identity linkage function. This does not assume the errors are Gaussian, because the error distribution is nonparametric, using the CLT for approximately accurate confidence intervals in large sample sizes, as in Liang and Zeger (1986). Please also see our response and excerpt to Comment R5.9 for details.

(i) Line 676: Is it a model for CD? If it is for LD, we need two indices for individuals and repeated samples, y_{ij} .

This is the mean model for CD and LD and we used OLS and GEE to estimate this model for CD and LD, respectively. We now added the indices and added a sentence to clarify that the j 's in all cross-sectional data are 1. We also made similar edits to the model equation for the FC analysis in ABCD.

(j) Line 677: Where - start with a lowercase letter for “where”

Corrected.

(k) Line 679: The number of knots and degree of freedom are different. Clarify this in lines 679 and 676.

Thanks for pointing this out. We have corrected it and clarified the specification of natural splines in terms of the placements of knots and degrees of freedom, and made sure they are consistent throughout the manuscript.

(l) : Lines 679-680: For GEE models, we used an exchangeable correlation structure-

that can be written as an active sentence.

Corrected.

(m) Line 687: it is confusing when i is used for the individual and study. I'd use k for study.

Thanks. We have edited it.

(n) Line 688 and 692: weights are the inverse of the standard error of each RESI estimate is repeated.

Thanks. We have corrected it.

(o) Line 695: Why do we need a spline for sex?

That is the proportion of males in a study, which was included as a covariate to measure the study features. We applied splines to such proportions.

(p) Line 702: Remove "respectively"

Edited.

(q) Line 703: Remove "the partial regression plots"

Edited.

(r) Line 728: 95% confidence intervals (CIs) for effect size estimate - remove estimate. CI is constructed for the parameter, not for the estimator.

Thank you. We have corrected it and made sure to use the correct description throughout the manuscript.

(s) Line 814 and wherever it appears: In the GEE models - GEE is a method of estimation, not a model. The GEE is applicable across a wide range of models and settings.

Thank you, we have corrected it throughout the manuscript.

(t) Line 820: As a statistician, I can't understand the "frame count of each scan".

This is a neuroimaging term. Each participant's scan consists of a series of images (frames), and the functional connectivity measures for a participant were derived from these images. The

frame count is often used as a quality metric as it is the number of frames used to derive the functional connectivity measures, therefore, we used it as the weights in the analyses.

(u) Line 820-822 and wherever it appears: authors can specify only one time the packages used for GEE estimation and RESI CI calculation. The same statement is repeated many times in the Methods section.

We have removed the redundancy.

(v) Line 836: how do you calculate the F-statistic?

We used the F distribution to calculate the required sample sizes for 80% power after estimating the effect size.

(w) Line 838: What is df here?

The degrees of freedom for this F -distribution is 1 and $N-df$, where N is the total sample size and df is the total degree of freedom of the analysis model. We described it when introducing the cumulative density function for this F -distribution:

“Let df denote the total degree of freedom of the analysis model, $F(x; \lambda)$ denote the cumulative density function for the random variable X , which follows the (non-central) F -distribution with degrees of freedoms being 1 and $N - df$ and non-centrality parameter λ .”

(x) Line 848: Subscripts are not presented well. It might be helpful to specify the model similar to (2.2.4, page 23 of Diggle (2002)).

$$GMV_{ij} \sim ns(\text{age}, df = 2) + \text{sex} + \beta_C x_{i1} + \beta_L (x_{ij} - x_{i1}),$$

where β_C and β_L are the cross-section and longitudinal covariate effects, respectively.

Thank you. We present this using R notation so that readers can see how we specified the models. More details on how the covariates are computed are provided in Supplement: Section 3.2 and we add a reference to that supplementary section there now.

References

Diggle, P. (2002). Analysis of longitudinal data. Oxford university press.

Reviewer Reports on the Second Revision:

Editorial note: Reviewer 2 was unable to return a re-review but as there was overlapping expertise with the other referees, the author's responses to R2 were evaluated by the other referees and deemed sufficient.

Referee #1 (Remarks to the Author):

Kang et al., have included additional analyses and added clarifying text. However, for a more general science audience, beyond statisticians, the current organization of the text and certain terms and phrases run a very high risk of being misunderstood (or not understood). The potential impact of this work does not lie in the statistics per se, but their applicability to BWAS, which represent a sizable enterprise at this point. Thus, this work needs to be presented in a manner that is digestible by many and puts the findings in relation to de facto practices.

BWAS have been reporting effect size estimates and recent work highlighting poor replicability of small N BWAS results, have also been about effect size estimates. Thus, the predominant association with the term 'effect size' amongst BWAS practitioners is 'effect size estimate'. The revised manuscript now includes a clarification, that the authors are referring to 'effect size parameters', when writing about 'effect sizes', but it is buried deep in the Discussion (lines 444-453). The term 'parameter' isn't used until line 371 for any reason or in any context (main text). It seems that effect size parameters correspond to what the authors refer to 'actual effect sizes' in the abstract. This distinction is critically important and must be made at the very beginning of the story. Since BWAS researchers deal with effect size estimates it must then be made crystal clear and explicit how these effect size parameters relate to effect size estimates, from the outset. This is needed to justify the focus on effect size parameters (actual effect sizes) over effect size estimates.

Throughout the manuscript, the dominant phrasing talks about 'increasing' and 'improving' effect sizes through various maneuvers. This terminology is misleading and given the history of publishing inflated effect size estimates, also dangerous. An association, for example that of height and weight in humans in 2024 (cross-sectional), has a correct answer that could be obtained by measuring the height and weight of every human. If this true, global, height/weight correlation is an effect size, then effect sizes cannot be increased or improved by sampling schemes. As the sample size approaches all 8 billion humans, the u-shaped and flat sampling schemes will include the same information and must return the same effect size. So it would be helpful to clarify how the findings described in this manuscript relate to the true biological answers out there in the world. The authors themselves write in their response to reviewers that the effect sizes (parameters) when switching from cross-sectional to longitudinal, are different effects. Thus, instead of including text that makes it sound as if a sampling scheme can change a biological fact, it must be made much clearer that the longitudinal effect size parameters and cross-sectional effect size parameters are the correct answers to slightly different scientific questions. It seems important to use very precise language. Some of the text (especially in the most prominent spots) could currently be understood as the authors advertising a transmogrification that turns a Chihuahua into a Rottweiler. Instead, it should be clear that the study shows that if someone wants a bigger dog, they should get a Rottweiler, not a Chihuahua.

The strong preference for publishing larger effect size estimates has fueled the replicability crises. The true/actual effect sizes are still only theoretical at this point, and we're still stuck with estimates, since BWAS samples are still on the smaller side (for real-world reasons). The results described in this manuscript provide an opportunity to drive home that the goal of BWAS is increasing our understanding the brain in the world, which requires replicability. Instead of advertising larger effect size (parameters) in every subheading, it might be more beneficial to take it a step further and highlight the greater replicability instead. Smaller true effect sizes can still be of great societal/biological relevance, as long as they're not bogus. Something that has a tiny negative effect on the brain, but applies to everyone, may still be worth remedying. Bigger effect size parameters are better because they can achieve replicability in smaller samples, not because smaller true effect sizes are always scientifically useless. Statistically larger effect size parameters are superior, but scientifically it still matters what the question is.

To me the points most important for BWAS researchers are: On average longitudinal designs have superior replicability, but not necessarily when asking a cross-sectional question. If you're only going longitudinal for statistical, not scientific reasons, maybe stop at two timepoints. Picking the right sampling scheme can boost replicability; strongly consider oversampling the tails. Explicitly model within- and between-participant effects or pay the price.

Referee #4 (Remarks to the Author):

I thank the authors for taking the effort to address my comments. I have no further concerns.

Referee #4 (Remarks on code availability):

The authors provided the code. Since the authors used data from public databases, my understanding is that one shall be able to reproduce the findings if they have the access to the databases.

Referee #5 (Remarks to the Author):

Thank you for accommodating some of the recommendations in your revision. I was happy with the revision until I saw your definition of the effect size parameter. Here is my explanation.

C. Data & methodology: validity of approach, quality of data, quality of presentation

Unfortunately, I am confused about the definition of effect size after reading your response to other reviewers. You now define an effect size parameter and an effect size estimate. Based on the previous version of the manuscript, I assumed the authors were discussing the association between BWAS features in a regression setting. They estimate and infer the association using effect size to understand the impact of study design in both cross-sectional (CD) and longitudinal (LD) designs within BWAS.

Unfortunately, the authors introduced the effect size parameter in the current revision, which is my main concern. Let me illustrate my point with a simple linear regression example.

Suppose we have a response variable y and a predictor variable x . I want to study the association between y and x . There are many model families, but I assume the association is linear, so I chose a linear model family. Next, I write the model as $y = \beta_0 + \beta_1 x + \epsilon$. Here, the association between y and x is measured by β_1 . This is the true strength of the association.

Next, I want to infer the true strength of the association. I define a test statistic, which is $\frac{\hat{\beta}_1}{SD(\hat{\beta}_1)}$. We can call this test statistic the effect size. The test statistic is a function of random variables, so we study its properties under different sampling schemes (or study designs).

If we have longitudinal data, the model can be changed to account for the within-subject variation: $y = \beta_0 + \beta_1 x + \tau Z + \epsilon$. Note that the true association β_1 doesn't change.

I am sorry, but I cannot comment on the revision without understanding the effect size (standardized parameter) terminology. I am unaware of the true association being defined using its standard deviation. In the frequentist approach, the parameter is fixed, so there is no variation. Therefore, how do we define the effect size parameter?

More specifically, the authors' response states: "So it's not that the ES estimate approaches the constant underlying truth, but reflects how the underlying truth changes with study designs." I argue that the underlying truth never changes because it is fixed, whereas the estimate of the underlying truth can definitely change based on the test statistic. In the regression example, we test $H_0 : \beta_1 = 0$; we don't test either $H_0 : \frac{\beta_1}{SD(\beta_1)} = 0$ or $H_0 : \frac{\hat{\beta}_1}{SD(\hat{\beta}_1)} = 0$.

Please clarify this terminology so I can provide more comprehensive feedback.

Author Rebuttals to Second Revision:

Referee #1 (Remarks to the Author):

R1.1 Kang et al., have included additional analyses and added clarifying text. However, for a more general science audience, beyond statisticians, the current organization of the text and certain terms and phrases run a very high risk of being misunderstood (or not understood). The potential impact of this work does not lie in the statistics per se, but their applicability to BWAS, which represent a sizable enterprise at this point. Thus, this work needs to be presented in a manner that is digestible by many and puts the findings in relation to de facto practices.

BWAS have been reporting effect size estimates and recent work highlighting poor replicability of small N BWAS results, have also been about effect size estimates. Thus, the predominant association with the term 'effect size' amongst BWAS practitioners is 'effect size estimate'. The revised manuscript now includes a clarification, that the authors are referring to 'effect size parameters', when writing about 'effect sizes', but it is buried deep in the Discussion (lines 444-453). The term 'parameter' isn't used until line 371 for any reason or in any context (main text). It seems that effect size parameters correspond to what the authors refer to 'actual effect sizes' in the abstract. This distinction is critically important and must be made at the very beginning of the story. Since BWAS researchers deal with effect size estimates it must then be made crystal clear and explicit how these effect size parameters relate to effect size estimates, from the outset. This is needed to justify the focus on effect size parameters (actual effect sizes) over effect size estimates.

Throughout the manuscript, the dominant phrasing talks about 'increasing' and 'improving' effect sizes through various maneuvers. This terminology is misleading and given the history of publishing inflated effect size estimates, also dangerous. An association, for example that of height and weight in humans in 2024 (cross-sectional), has a correct answer that could be obtained by measuring the height and weight of every human. If this true, global, height/weight correlation is an effect size, then effect sizes cannot be increased or improved by sampling schemes. As the sample size approaches all 8 billion humans, the u-shaped and flat sampling schemes will include the same information and must return the same effect size. So it would be helpful to clarify how the findings described in this manuscript relate to the true biological answers out there in the world. The authors themselves write in their response to reviewers that the effect sizes (parameters) when switching from cross-sectional to longitudinal, are different effects. Thus, instead of including text that makes it sound as if a sampling scheme can change a biological fact, it must be made much clearer that the longitudinal effect size parameters and cross-sectional effect size parameters are the correct answers to slightly different scientific questions. It seems important to use very precise language. Some of the text (especially in the most prominent spots) could currently be understood as the authors advertising a transmogrification that turns a Chihuahua into a Rottweiler. Instead, it should be clear that the study shows that if someone wants a bigger dog, they should get a Rottweiler, not a Chihuahua.

Thank you for mentioning these risks of being misunderstood by the readers. We agree that bringing these distinctions up earlier in the paper is essential. Furthermore, we agree that it is important to be very precise with language, while avoiding confusing statistical terminology. To do this, we have made significant revisions throughout the Introduction and Discussion. Broadly, we consulted with our coauthors, who made substantial edits to the Introduction and Discussion to improve the clarity. As described above, we also move the section “Comparison of L-RESI and CS-RESI quantifies benefit of longitudinal design” to the Supplementary Information to reduce statistical content.

Previously, as you described, we did not include a discussion of standardized ESs and the true biological effects until the discussion. We now explicitly state the relationship between standardized effect sizes (ESs) and underlying biological effects in the Introduction to contextualize our entire study. We now clarify that standardized ESs depend on both the underlying biological effects and study designs, and two studies of the same biological effect with different study designs will have different standardized ESs. This is to emphasize your point, that the underlying biological effects in the population are not affected by our modifications to the study designs, but standardized ESs can still be affected. This is supported, now, with an example illustrating how a standardized ES can be affected by the study design. In addition, throughout the paper, we now refer to “standardized effect size” or “standardized ES” instead of “effect size” to make it clear we are focusing on standardized ESs instead of effect sizes in general. We explain how standardized effect sizes are dependent on study design in the third paragraph of the Introduction:

Standardized ESs (such as Pearson’s correlation and Cohen’s d) are statistical values that not only depend on the underlying biological association in the population, but also on the study design. Two studies of the same biological effect with different study designs will have different standardized ESs. For example, contrasting brain function of depressed versus non-depressed groups will have a different Cohen’s d ES if the study design measures more extreme depressed states contemporaneously with measures of brain function, as opposed to less extreme depressed states, even if the underlying biological effect is the same. While researchers cannot increase the magnitude of the underlying biological association, its standardized ES – and thus its replicability – can be increased by critical features of study design.

To clarify the distinction between *parameters* and *estimates*, we do not explicitly define these terms now, to avoid burdening the readers with statistical jargon. We point out that choosing a study design *prior to data collection* can lead to larger standardized ESs without biasing the biological effect. We contrast this with analysis procedures, such as p -hacking, that produce biased/inflated effect size estimates during data analysis. In the fourth paragraph of the Introduction:

In this paper, we focus on identifying modifiable study design features that can be used to improve the replicability of BWAS by increasing standardized ESs. Increasing standardized ESs through study design prior to data collection stands in sharp contrast to bad research practices that can artificially inflate reported ESs, such as p -hacking and publication bias. Surprisingly, there has been very little research regarding how modifications to the study design might improve BWAS replicability. Specifically, we focus on two major design features that directly influence standardized ESs: variation in sampling scheme and longitudinal designs^{1,12-14}. Notably, these design features can be implemented without inflating the sample estimate of the underlying biological effect when using correctly specified models¹⁵. By increasing the replicability of BWAS through study design, we can more efficiently utilize the National Institutes of Health's \$1.8 billion average annual investment in neuroimaging research in the past decade¹⁶.

Finally, we also move the whole section "Comparison of L-RESI and CS-RESI quantifies benefit of longitudinal design" to the Supplement to reduce the statistical content and the risk of being misunderstood by the audience. We believe this further reduces the burden of statistical content and highlights the key information of this work to make it more digestible.

***R1.2** The strong preference for publishing larger effect size estimates has fueled the replicability crises. The true/actual effect sizes are still only theoretical at this point, and we're still stuck with estimates, since BWAS samples are still on the smaller side (for real-world reasons). The results described in this manuscript provide an opportunity to drive home that the goal of BWAS is increasing our understanding the brain in the world, which requires replicability. Instead of advertising larger effect size (parameters) in every subheading, it might be more beneficial to take it a step further and highlight the greater replicability instead. Smaller true effect sizes can still be of great societal/biological relevance, as long as they're not bogus. Something that has a tiny negative effect on the brain, but applies to everyone, may still be worth remedying. Bigger effect size parameters are better because they can achieve replicability in smaller samples, not because smaller true effect sizes are always scientifically useless. Statistically larger effect size parameters are superior, but scientifically it still matters what the question is.*

We fully agree that small biological effects are very important. As we responded to your comment above, we now have emphasized that the study of underlying biological effects is not affected by our increasing the standardized effect sizes (ESs) through study designs. We do not explicitly contrast small and large biological effects in this paper, as our focus is on standardized ESs and replicability. We previously focused on standardized ES as they (along with the sample size) directly control replicability, and as such are critical in the broad effort to improve replicability independent

of the sample size. In response to your comments, we have thoroughly revised the manuscript to refocus our narrative to replicability. Specifically, we have:

1. We changed our title to “Study design features increase replicability in cross-sectional and longitudinal brain-wide association studies”.
2. We defined replicability more precisely: “Statistical replicability is typically defined as the probability of obtaining consistent results from hypothesis tests across different studies”;
3. We explicitly emphasized the implication of higher replicability from larger standardized effect sizes (ESs) as appropriate;
4. We modified Figures 2, 3, and 4 to directly illustrate the impact of sampling schemes on standardized ESs and replicability (they previously showed power). Moreover, in Figures 4 and 5, we reduced the number of covariates shown to highlight the key information which makes the figure more readable. The remaining covariates are shown in Extended Data Figures (Figures S6 and S9), which were also shown in the previous version of the paper. The new Figures are pasted below for convenience.
5. We thoroughly revised the subsection of the Discussion that previously focused on increasing standardized effect sizes to instead focus on replicability, as detailed below:

Sampling strategies can increase replicability

Our results demonstrate that standardized ES and replicability can be increased by enriched sampling of subjects with small and large values of the covariate of interest. This is well-known in linear models where the standardized ES is explicitly a function of the SD of the covariate²⁴. We show that designing a study to have larger covariate SD increases standardized ESs by a median factor of 1.4, even when there is nonlinearity in the association, such as with age and GMV (Fig. S1). When the association is very non-monotonic – as in the case of a U-shape relationship between covariate and outcome – sampling the tails more heavily could decrease replicability, and diminish our ability to detect nonlinearities in the center of the study population. In such a case, sampling to obtain a uniform distribution of the covariate balances power across the range of the covariate and can increase replicability relative to random sampling when the covariate has a normal distribution in the population. Increasing between-subject variability is beneficial in more than 72% of the association pairs we studied, despite the presence of such nonlinearities (Fig. S7).

Because standardized ESs are dependent on study design, careful design choices can simultaneously increase standardized ESs and study replicability. Two-phase, extreme group, and outcome-dependent sampling designs can inform which subjects should be selected for imaging from a larger sample in order to increase the efficiency and standardized ESs of brain-behavior associations²⁸⁻³⁴. For example, given the high degree of accessibility of cognitive and behavioral testing (e.g., to be performed virtually or electronically), individuals scoring at the extremes on a testing scale/battery (“phase I”) could be prioritized for subsequent brain

scanning (“phase II”). When there are multiple covariates of interest, multivariate two-phase designs can be used to increase standardized ESs and replicability³⁵. Multivariate designs are also needed to stratify sampling to avoid confounding by other socio-demographic variables. Together, the use of optimal designs can increase both standardized ESs and replicability relative to a design that uses random sampling³². If desired, weighted regression (such as inverse probability weighting) can be combined with optimized designs to estimate a standardized ES that is consistent with the standardized ES if the study had been conducted in the full population³⁵⁻³⁷. Choosing highly reliable psychometric measurements or interventions (e.g., medications or neuromodulation within a clinical trial)³⁸⁻⁴⁰ may also be effective for increasing replicability. The decision to pursue an optimized design will also depend on other practical factors, such as the cost and complexity of acquiring other (non-imaging) measures of interest and the specific translational goals of the research.

Fig. 2. Increased standardized effect sizes (ESs) and replicability for age associations with different brain measures: (a-b) total gray matter volume (GMV), (c-d) regional GMV, and (e-f) regional cortical thickness (CT), under three sampling schemes in the UKB study ($N = 29,031$ for total GMV and $N = 29,030$ for regional GMV and CT). The sampling schemes target different age distributions to increase the variability of age: Bell-shaped < Uniform < U-shaped (Fig. S3). Using the resampling schemes that increase age variability increases (a) the standardized ESs and (b) replicability (at significance level of 0.05) for total GMV-age association. The same result holds for (c-d) regional GMV and (e-f) regional CT. The curves represent the average standardized ES or estimated replicability at a given sample size and sampling scheme. The shaded areas represent the corresponding 95% confidence bands. (c-f) The bold curves are the average standardized ES or replicability across all regions with significant uncorrected effects using the full UKB data.

Fig. 3. Increased standardized effect sizes (ESs) and replicability for age associations with structural brain measures under different longitudinal sampling schemes in the ADNI data. Three different sampling schemes (Fig. S5) are implemented in bootstrap analyses to modify the between- and within-subject variability of age, respectively. **(a-b)** Implementing the sampling schemes results in higher (between- and/or within-subject) variability and increases the **(a)** standardized ES and **(b)** replicability (at significance level of 0.05) for the total gray matter volume (GMV)-age association. The curves represent the average standardized ES or estimated replicability and the gray areas are the 95% confidence bands across the 1,000 bootstraps. **(c-d)** Increasing the number of measurements from one to two per subject provides the most benefit on **(c)** standardized ES and **(d)** replicability for the total GMV-age association when using uniform between- and within-subject sampling schemes and $N = 30$. The points represent the mean standardized ESs or estimated replicability and the whiskers are the 95% confidence intervals. **(e-h)** Increased standardized ESs and replicability for the associations of age with regional **(e-f)** GMV and **(g-h)** cortical thickness (CT), across all brain regions under different sampling schemes. The bold curves are the average standardized ES or estimated replicability across all regions with significant uncorrected effects using the full ADNI data. The gray areas are the corresponding 95% confidence bands. Increasing the between- and within-subject variability of age by implementing different sampling schemes can increase the standardized ESs of age, and the associated replicability, on regional GMV and regional CT.

Fig. 4. Heterogeneous improvement of standardized effect sizes (ESs) for select cognitive, mental health, and demographic associations with structural and functional brain measures in the ABCD study with bootstrapped samples of $N = 500$. (a) U-shaped between-subject sampling scheme (blue) that increases between-subject variability of the non-brain covariate produces larger standardized ESs and (b) reduces the number of participants scanned to obtain 80% replicability in total gray matter volume (GMV). The points and triangles are the average standardized ESs across bootstraps and the whiskers are the 95% confidence intervals. Increasing within-subject sampling (triangles) can reduce standardized ESs. A similar pattern holds in (c-d) regional GMV and (e-f) regional cortical thickness (CT); boxplots show the distributions of the standardized ESs across regions (or region pairs for functional connectivity (FC)). In contrast, (g) regional pairwise FC standardized ESs improve by increasing between- (blue) and within-subject variability (dashed borders) with a corresponding reduction in the (h) number of participants scanned for 80% replicability. See Fig. S6 for the results for all non-brain covariates examined.

R1.3 *To me the points most important for BWAS researchers are: On average longitudinal designs have superior replicability, but not necessarily when asking a cross-sectional question. If you're only going longitudinal for statistical, not scientific reasons, maybe stop at two timepoints. Picking*

*the right sampling scheme can boost replicability; strongly consider oversampling the tails.
Explicitly model within- and between-participant effects or pay the price.*

This is a great summary of the majority of the key points in our work. We have made sure these points are appropriately emphasized in the Conclusions part of our Discussion.

...One general principle that increases standardized ESs for most associations is to increase the covariate SD (through, e.g., two-phase, extreme group, and outcome-dependent sampling), which is practically applicable to a wide range of BWAS contexts. Moreover, longitudinal designs can provide larger standardized ES and higher replicability, but the benefit is dependent on the form of the association between the covariate and outcome. Longitudinal BWAS enable us to study between- and within-subject effects, and they should be used when the two effects are hypothesized to be different. Together, our findings highlight the importance of rigorous evaluation of study designs and statistical analysis methods that can improve the replicability of BWAS.

Referee #4 (Remarks to the Author):

I thank the authors for taking the effort to address my comments. I have no further concerns.

Referee #4 (Remarks on code availability):

The authors provided the code. Since the authors used data from public databases, my understanding is that one shall be able to reproduce the findings if they have the access to the databases.

Thank you for reviewing our work. Yes, we provided all the code we used for the analyses in this work and the results of our analyses (i.e., the model fitting objects) on our github repository (https://github.com/KaidiK/RESI_BWAS). Others can reproduce the tables and figures shown in the manuscript based on our analysis results. Although we cannot directly share the subject-level data from the studies, many of them are available through the original consortia. Once others have the same datasets and applied the same pre-processing, they can reproduce our findings using our code.

Referee #5

Thank you for accommodating some of the recommendations in your revision. I was happy with the revision until I saw your definition of the effect size parameter. Here is my explanation.

C. Data & methodology: validity of approach, quality of data, quality of presentation

Unfortunately, I am confused about the definition of effect size after reading your response to other reviewers. You now define an effect size parameter and an effect size estimate. Based on the previous version of the manuscript, I assumed the authors were discussing the association between BWAS features in a regression setting. They estimate and infer the association using effect size to understand the impact of study design in both cross-sectional (CD) and longitudinal (LD) designs within BWAS.

Unfortunately, the authors introduced the effect size parameter in the current revision, which is my main concern. Let me illustrate my point with a simple linear regression example.

Suppose we have a response variable y and a predictor variable x . I want to study the association between y and x . There are many model families, but I assume the association is linear, so I chose a linear model family. Next, I write the model as $y = \beta_0 + \beta_1 x + \epsilon$. Here, the association between y and x is measured by β_1 . This is the true strength of the association.

Next, I want to infer the true strength of the association. I define a test statistic, which is $\frac{\hat{\beta}_1}{SD(\hat{\beta}_1)}$. We can call this test statistic the effect size. The test statistic is a function of random variables, so we study its properties under different sampling schemes (or study designs).

If we have longitudinal data, the model can be changed to account for the within-subject variation: $y = \beta_0 + \beta_1 x + \tau Z + \epsilon$. Note that the true association β_1 doesn't change.

I am sorry, but I cannot comment on the revision without understanding the effect size (standardized parameter) terminology. I am unaware of the true association being defined using its standard deviation. In the frequentist approach, the parameter is fixed, so there is no variation. Therefore, how do we define the effect size parameter?

More specifically, the authors' response states: "So it's not that the ES estimate approaches the constant underlying truth, but reflects how the underlying truth changes with study designs." I argue that the underlying truth never changes because it is fixed, whereas the estimate of the underlying truth can definitely change based on the test statistic. In the regression example, we test $H_0 : \beta_1 = 0$; we don't test either $H_0 : \frac{\beta_1}{SD(\beta_1)} = 0$ or $H_0 : \frac{\hat{\beta}_1}{SD(\hat{\beta}_1)} = 0$.

Please clarify this terminology so I can provide more comprehensive feedback.

As discussed in response to R1 above, in the revised manuscript we make efforts to distinguish the standardized effect sizes (ESs) and the underlying biological effects, and briefly explain why we can increase the standardized ESs through study designs *prior to data collection*. We do not rely on technical definitions of these to avoid burdening nonstatistical readers with jargon. We instead give the distinction of how choosing a study design prior to data collection can lead to larger standardized ESs without biasing the estimation of biological effects, and contrast this with analysis procedures, such as p -hacking, that produce biased reported effect sizes in the analysis stage. Please see our R1.1-R1.3 above.

Regarding the technical definition of the standardized effect size index, following your regression example, let's say we want to test the null hypothesis $H_0 : \beta_1 = 0$, using the test statistic $\sqrt{N} \times \hat{\beta}_1 / \widehat{SD}(\sqrt{N}\hat{\beta}_1)$, where $\widehat{SD}(\sqrt{N}\hat{\beta}_1)$ denotes an estimator of the standard deviation of $\sqrt{N}\hat{\beta}_1$. If the null is false, the expected value of the test statistic is asymptotically equivalent to $\sqrt{N} \times \beta_1 / SD(\sqrt{N}\hat{\beta}_1)$. This consists of a part $\beta_1 / SD(\sqrt{N}\hat{\beta}_1)$ that is independent of sample size and a

part that is the square root of sample size. A signed version of the RESI is defined as $\beta_1 / SD(\sqrt{N}\hat{\beta}_1)$ (see Jones, Kang, Vandekar, 2023). Its numerator, β_1 , captures the biological effect and the denominator, $SD(\sqrt{N}\hat{\beta}_1)$, captures all features of the procedures used to estimate β_1 , including study design features such as how the data is collected and the study design. We defined the RESI in the Methods section using this notation and terminology.

The RESI is a recently developed standardized effect size (ES) index that has consistent interpretation across many model types, encompassing all types of test statistics in most regression models^{21,22}. Briefly, the RESI is a standardized ES parameter describing the deviation of the true parameter value(s) β from the reference value(s) β_0 from the statistical null hypothesis $H_0: \beta = \beta_0$,

$$S = \sqrt{(\beta - \beta_0)^T \Sigma_\beta^{-1} (\beta - \beta_0)},$$

where S denotes the parameter RESI, β and β_0 can be vectors, Σ_β is the covariance matrix for $\sqrt{N}\hat{\beta}$ (where $\hat{\beta}$ is the estimator for β) (Supplementary Information: Section 1).

In your example case, the aim of our study would be to explore how we can leverage study design features to increase the statistical efficiency (i.e., to decrease the $SD(\sqrt{N}\hat{\beta}_1)$) to improve the standardized effect size parameter $\beta_1 / SD(\sqrt{N}\hat{\beta}_1)$, and ultimately, the replicability. The RESI will change with studies with different sampling schemes and designs, but the biological parameter, β_1 , is not affected. For example, using the example you describe, imagine another study that collects two measurements on each subject, and models the difference in $y_{i2} - y_{i1}$ with $x_{i2} - x_{i1}$,

$$y_{i2} - y_{i1} = 0 + \beta_1(x_{i2} - x_{i1}) + \epsilon_{i2} - \epsilon_{i1}.$$

This regression will target the same parameter as the cross-sectional version of the study, because the biological effect will be the same; but the standard deviation of $\hat{\beta}_1$ will be different, because it will depend on $Var(\epsilon_{i2} - \epsilon_{i1}) = Var(\epsilon_{i2}) + Var(\epsilon_{i1}) - 2Cov(\epsilon_{i2}, \epsilon_{i1})$ instead of $Var(\epsilon_i)$, therefore, the standardized ESs will differ. Reviewer 1 noted similar areas for misunderstanding and we've made substantial edits to the Introduction and Discussion to emphasize this (See response to R1.1 and R1.2).

We've changed the terminology so that we don't refer to a particular study design as "the underlying truth." Specifically, in the section "*Sampling strategies can increase replicability*" of the discussion, we previously referred to the bias of the standardized ES relative to random sampling, but instead described this simply as being a different standardized ES:

Because standardized ESs are dependent on study design, careful design choices can simultaneously increase standardized ESs and study replicability. Two-phase, extreme

group, and outcome-dependent sampling designs can inform which subjects should be selected for imaging from a larger sample in order to increase the efficiency and standardized ESs of brain-behavior associations²⁸⁻³⁴. For example, given the high degree of accessibility of cognitive and behavioral testing (e.g., to be performed virtually or electronically), individuals scoring at the extremes on a testing scale/battery (“phase I”) could be prioritized for subsequent brain scanning (“phase II”). When there are multiple covariates of interest, multivariate two-phase designs can be used to increase standardized ESs and replicability³⁵. Multivariate designs are also needed to stratify sampling to avoid confounding by other socio-demographic variables. Together, the use of optimal designs can increase both standardized ESs and replicability relative to a design that uses random sampling³². If desired, weighted regression (such as inverse probability weighting) can be combined with optimized designs to estimate a standardized ES that is consistent with the standardized ES if the study had been conducted in the full population³⁵⁻³⁷. Choosing highly reliable psychometric measurements or interventions (e.g., medications or neuromodulation within a clinical trial)³⁸⁻⁴⁰ may also be effective for increasing replicability. The decision to pursue an optimized design will also depend on other practical factors, such as the cost and complexity of acquiring other (non-imaging) measures of interest and the specific translational goals of the research.

Reference

1. Jones, M., Kang, K. & Vandekar, S. RESI: An R Package for Robust Effect Sizes. Preprint at <https://doi.org/10.48550/arXiv.2302.12345> (2023)

Reviewer Reports on the Third Revision:

Referee #1 (Remarks to the Author):

Overall, this latest version of the manuscript is greatly improved in clarity, and its relevance for BWAS more obvious.

The title and abstract and text are all improved over the prior version.

Abstract comments:

The sentence “We demonstrate that commonly used longitudinal models can, counterintuitively, reduce standardized ESs and replicability” could be sharpened by using a more informative qualifier than ‘commonly used’. What types of longitudinal models have this property? What are the critical factors of these types of models? The next sentence: “The benefit of conducting longitudinal studies depends on the strengths of the between- versus within-subject associations of the brain and non-brain measures,” provides some, but only partial clarification. One possibility could be to delete the first sentence. Alternatively, the logical link between these two sentences could be made stronger. Overall, this two sentence segment feels a little imbalanced given that longitudinal designs are great for looking at age, but there isn’t a sentence dedicated to highlighting their benefits, while there is a full sentence highlighting that they can make things worse. So this section could be more balanced by presenting the results more in the fashion of: longitudinal designs are preferable if X, and cross-sectional ones if Y.

Minor:

Grammar mismatch “BWAS with larger variability in covariate” ... should that be plural ‘covariates’

Consider replacing subject with participant in the entire document.

Intro:

Need a citation in the sentence that first mentions RESI.

Replace use of the word ‘paper’ with ‘study’ or some other more formal synonym. ‘Paper’ is too colloquial.

Consider ending the first section (de facto intro) with the sentence that ends in “that allows us to demonstrate how longitudinal study design directly impacts standardized ESs.” The text after that feels too much like foreshadowing without any evidence (yet). The content of the last 5 sentences of this first section should be in the abstract, not at the end of the intro. It’s good text, so some of it might fit into the abstract, but it feels out of place at the end of the introduction. It violates formatting conventions, and some readers might get annoyed that there’s too much assertion without evidence prior to the results, but outside of the abstract.

Under which assumptions is $RESI = \frac{1}{2}$ Cohen's d ? is there a short enough way to explain that in the main text. Currently the readers are just sent to the Methods.

Second paragraph under first subheading "Meta-analyses show ..."

It would be helpful to give a one clause motivation for why you're using the RESI, other than coming up with it. What is its benefit over Cohen's d ? Do all the results hold with another measure of standardized ES? I'm not suggesting ditching RESI, but without a brief motivational clause, it seems somewhat arbitrary.

In the section "Improved sampling schemes ..." it would be appropriate to add a clause that age effects are very large compared to much more subtle cognitive and/or psychopathology effects, given the endless debates over a handful of examples where in retrospect a sample of $< 1,000$ would've led to replicable findings.

For Figure 2 the caption title (in bold) seems rather too lengthy. Could maybe end it at the colon and then have the other info under the respective sections of the caption text.

First sentence of the second paragraph, should the verb be past tense?

"Preferred sampling schemes ..." section

Use past tense to make style consistent with prior sections.

Second to last sentence, break up into two sentences for improved clarity. Could probably delete 'to scan' from the first of those two sentences.

Last sentence of the section, consider using passive voice for the sentence's second clause. In this study, which relies on resampling to simulate various designs, the authors are actively increasing or decreasing factors like the between-participant variability of the covariates, so the active voice seems appropriate. Yet, in the actual conduct of BWAS, researchers cannot and should not change these variables, once the data have been collected. In a real study, we're not changing the inter or intra-participant variability, we're only passively receiving them. They're already facts of the available data sets. It's purely stylistic but using the active voice too often in such sentences creates the feeling of something actively nefarious, such as targeted post-hoc changes to a sample and other means of p-hacking. In the conduct of an actual study, not the simulation of one, you're stuck with the inter-participant variability, you can't increase or decrease it anymore.

Second paragraph "This finding is consistent when fitting ..." sounds unidiomatic. Consider replacing 'is consistent' with something like 'holds'.

Last sentence: Text in parentheses and foreshadowing (see below) are stylistically unusual. Omit the parentheses and either delete (see below) or reorganize text so that it isn't required.

Consider giving the third paragraph in this section its own subheading.

3rd paragraph: GEEs hasn't been introduced yet as an abbreviation.

For consistency with Nature style: Remove 'Discussion' subheading and first paragraph underneath it which is a repetitive summary. Distribute any unique content to corresponding paragraphs.

Final paragraph: replace 'Conclusions' with subheading that carries specific information.

Questions I was left wanting more discussion of: Why does the benefit of longitudinal designs in the analyses presented taper off after 2 measurements? Are there scenarios where one would predict this not to be the case?

Are there scenarios/reasons why one might not want to estimate within & between-participant effects separately, or can you make a strong recommendation to always estimate them separately?

Can these findings be reduced to more generalized recommendations. ... in detail the results suggest that the answer is often 'it depends', but is there a deeper principle that can be highlighted in the last paragraph?

General comments:

Replace 'subject' with 'participant' throughout.

Consistently use past tense when describing results, currently the text contains a mix of past and present tense.

Shorten figure caption titles.

Keep subheadings to a line or shorter and have them all be specific (i.e. remove 'Discussion', 'Conclusion').

Remove repetitive summary and foreshadowing text from the end of first section (unofficial intro) and the beginning of the section currently labeled 'Discussion'.

Avoid active voice when discussing biological facts and when discussing the downstream effects of using a different study design.

Referee #5 (Remarks to the Author):

Thank you, authors, for accommodating the recommendations in your revision. I commend you for your efforts in reorganizing the manuscript to better emphasize the reproducibility of the BWAS experiment results.

While I cannot provide detailed comments on the decision to avoid using the statistical terms "parameter" and "estimates" in this manuscript and their implications for the intended audience, I understand from your response to the reviewers that the standardized effect size (ES) estimates depend on both the underlying biological effects and the study designs. As a result, two studies examining the same biological effect but with different study designs will yield different standardized ES estimates . Please note the bolded words.

One of my concerns is not related to the manuscript but to the response to the explanation of the standardized effect size based on the regression setting. The authors' response stated the following: "This consists of a part $\frac{\beta_1}{SD(\sqrt{N}\hat{\beta}_1)}$ that is independent of sample size". I want to emphasize that this formula explicitly has \sqrt{N} . Moreover, when the formula depends on the estimate's standard deviation, it must involve the sample (study design) and, thus, the sample size. Because of these reasons, I had a difficult time understanding RESI. I will leave this matter to the experts in your field for further consideration.

Although I do not have any major technical comments, I have noticed some instances where articles (a, an, the) are either missing or incorrectly placed.

Referee #5 (Remarks on code availability):

I skimmed through the codes but haven't tried to reproduce it.

Author Rebuttals to Third Revision:

Referee #1 (Remarks to the Author):

Overall, this latest version of the manuscript is greatly improved in clarity, and its relevance for BWAS more obvious.

The title and abstract and text are all improved over the prior version.

Abstract comments:

The sentence “We demonstrate that commonly used longitudinal models can, counterintuitively, reduce standardized ESs and replicability“ could be sharpened by using a more informative qualifier than ‘commonly used’. What types of longitudinal models have this property? What are the critical factors of these types of models? The next sentence: “The benefit of conducting longitudinal studies depends on the strengths of the between- versus within-subject associations of the brain and non-brain measures,” provides some, but only partial clarification. One possibility could be to delete the first sentence. Alternatively, the logical link between these two sentences could be made stronger. Overall, this two sentence segment feels a little imbalanced given that longitudinal designs are great for looking at age, but there isn’t a sentence dedicated to highlighting their benefits, while there is a full sentence highlighting that they can make things worse. So this section could be more balanced by presenting the results more in the fashion of: longitudinal designs are preferable if X, and cross-sectional ones if Y.

Thank you. We agree this is not a good representation of the findings. We’ve removed the second sentence and clarified the first. It now reads, “We demonstrate that commonly used longitudinal models, which assume equal between- and within-subject changes can, counterintuitively, reduce standardized ESs and replicability. Explicitly modeling the between- and within-subject effects avoids conflating them and enables optimizing the standardized ESs for each separately.”

Minor:

Grammar mismatch “BWAS with larger variability in covariate” ... should that be plural ‘covariates’

We’ve re-reviewed the entire paper for grammar.

Consider replacing subject with participant in the entire document.

We've decided to retain the term "subject" as we mean it in the more general statistical sense of an independent sampling unit and the terms "between-subject" and "within-subject" are more often used in the scientific literature.

Intro:

Need a citation in the sentence that first mentions RESI.

Added.

Replace use of the word 'paper' with 'study' or some other more formal synonym. 'Paper' is too colloquial.

Done.

Consider ending the first section (de facto intro) with the sentence that ends in "that allows us to demonstrate how longitudinal study design directly impacts standardized ESs." The text after that feels too much like foreshadowing without any evidence (yet). The content of the last 5 sentences of this first section should be in the abstract, not at the end of the intro. It's good text, so some of it might fit into the abstract, but it feels out of place at the end of the introduction. It violates formatting conventions, and some readers might get annoyed that there's too much assertion without evidence prior to the results, but outside of the abstract.

Thanks for this suggestion. We cut the abstract substantially per the request from the editing office while merging the components of this part to the abstract and Discussion.

Under which assumptions is $RESI = \frac{1}{2}$ Cohen's d ? is there a short enough way to explain that in the main text. Currently the readers are just sent to the Methods.

Under the situation where the classical Cohen's d was defined (i.e., when there is a test comparing two independent groups with equal sampling proportions and equal variance). We have changed it

to “The RESI is a recently developed index that is equal to $\frac{1}{2}$ Cohen’s d **under the same assumptions for Cohen’s d** (Methods; Supplementary Information)”

Second paragraph under first subheading “Meta-analyses show ...”

It would be helpful to give a one clause motivation for why you’re using the RESI, other than coming up with it. What is its benefit over Cohen’s d ? Do all the results hold with another measure of standardized ES? I’m not suggesting ditching RESI, but without a brief motivational clause, it seems somewhat arbitrary.

We’ve added a sentence after the introduction of the RESI, “ We use the RESI as a standardized ES because it is broadly applicable to many types of models and is robust to model misspecification.”

In the section “Improved sampling schemes ...” it would be appropriate to add a clause that age effects are very large compared to much more subtle cognitive and/or psychopathology effects, given the endless debates over a handful of examples where in retrospect a sample of < 1,000 would’ve led to replicable findings.

We’ve added a clause at the beginning of the following section, “As standardized ESs for age are often larger than those for behavioral associations, we investigate if the proposed sampling schemes are effective on a variety of non-brain covariates and their associations with structural and functional brain measures in all subjects (with and without neuropsychiatric symptoms) with cross-sectional and longitudinal measurements from the ABCD dataset.”

For Figure 2 the caption title (in bold) seems rather too lengthy. Could maybe end it at the colon and then have the other info under the respective sections of the caption text.

We’ve shortened this caption.

First sentence of the second paragraph, should the verb be past tense?

We primarily use present tense throughout the main paper. We’ve checked this section and the main sections of the paper to ensure the present tense. In the Methods, we use only past tense.

“Preferred sampling schemes ...” section

Use past tense to make style consistent with prior sections.

We use present tense throughout the main manuscript.

Second to last sentence, break up into two sentences for improved clarity. Could probably delete ‘to scan’ from the first of those two sentences.

We’ve split this up and simplified the sentence.

Last sentence of the section, consider using passive voice for the sentence’s second clause. In this study, which relies on resampling to simulate various designs, the authors are actively increasing or decreasing factors like the between-participant variability of the covariates, so the active voice seems appropriate. Yet, in the actual conduct of BWAS, researchers cannot and should not change these variables, once the data have been collected. In a real study, we’re not changing the inter or intra-participant variability, we’re only passively receiving them. They’re already facts of the available data sets. It’s purely stylistic but using the active voice too often in such sentences creates the feeling of something actively nefarious, such as targeted post-hoc changes to a sample and other means of p-hacking. In the conduct of an actual study, not the simulation of one, you’re stuck with the inter-participant variability, you can’t increase or decrease it anymore.

We agree and want to avoid the connotation of manipulating analyses, so we’ve changed this to passive voice.

Second paragraph “This finding is consistent when fitting ...” sounds unidiomatic. Consider replacing ‘is consistent’ with something like ‘holds’.

Changed to “holds.”

Last sentence: Text in parentheses and foreshadowing (see below) are stylistically unusual. Omit the parentheses and either delete (see below) or reorganize text so that it isn’t required.

We've deleted the parenthetical clause.

Consider giving the third paragraph in this section its own subheading.

We made these edits and it's called "Accurate longitudinal models are crucial" now.

3rd paragraph: GEEs hasn't been introduced yet as an abbreviation.

It was defined in the meta-analysis section.

For consistency with Nature style: Remove 'Discussion' subheading and first paragraph underneath it which is a repetitive summary. Distribute any unique content to corresponding paragraphs.

We've made these changes.

Final paragraph: replace 'Conclusions' with subheading that carries specific information.

We've changed this to "Design and analysis recommendations".

Questions I was left wanting more discussion of: Why does the benefit of longitudinal designs in the analyses presented taper off after 2 measurements? Are there scenarios where one would predict this not to be the case?

Are there scenarios/reasons why one might not want to estimate within & between-participant effects separately, or can you make a strong recommendation to always estimate them separately?

Can these findings be reduced to more generalized recommendations. ... in detail the results suggest that the answer is often 'it depends', but is there a deeper principle that can be highlighted in the last paragraph?

We gave this a lot of consideration when writing the paper and we believe the recommendations in the last paragraph are as general as we can be without making an inaccurate conclusion statement.

General comments:

Replace 'subject' with 'participant' throughout.

Consistently use past tense when describing results, currently the text contains a mix of past and present tense.

We've changed the term "subject" to "participant", except when talking about common experimental design features (i.e., "between-subject" and "within-subject") as they are more often used in the statistical literature. We now have made sure that we consistently use present tense throughout the main manuscript and past tense in the Methods.

Shorten figure caption titles.

Keep subheadings to a line or shorter and have them all be specific (i.e. remove 'Discussion', 'Conclusion').

We've edited all section headers to meet the journal requirements. Please see response to editor above.

Remove repetitive summary and foreshadowing text from the end of first section (unofficial intro) and the beginning of the section currently labeled 'Discussion'.

Avoid active voice when discussing biological facts and when discussing the downstream effects of using a different study design.

We've made these edits to the discussion, as described above and have reread the text to avoid active voice in describing the results of study design modifications.

Referee #5 (Remarks to the Author):

Thank you, authors, for accommodating the recommendations in your revision. I commend you for your efforts in reorganizing the manuscript to better emphasize the reproducibility of the BWAS experiment results.

While I cannot provide detailed comments on the decision to avoid using the statistical terms "parameter" and "estimates" in this manuscript and their implications for the intended audience, I understand from your response to the reviewers that the standardized effect size (ES) estimates depend on both the underlying biological effects and the study designs. As a result, two studies examining the same biological effect but with different study designs will yield different standardized ES estimates . Please note the bolded words.

One of my concerns is not related to the manuscript but to the response to the explanation of the standardized effect size based on the regression setting. The authors' response stated the following: "This consists of a part $\frac{\beta_1}{SD(\sqrt{N}\hat{\beta}_1)}$ that is independent of sample size". I want to emphasize that this formula explicitly has \sqrt{N} . Moreover, when the formula depends on the estimate's standard deviation, it must involve the sample (study design) and, thus, the sample size. Because of these reasons, I had a difficult time understanding RESI. I will leave this matter to the experts in your field for further consideration.

This was caused by the different ways of parameterization. As we stated in the Methods section for RESI, $SD(\sqrt{N} \times \hat{\beta})$ is the standard deviation of $\sqrt{N}(\hat{\beta})$. When a parameter $\hat{\beta}$ estimator is \sqrt{N} -consistent, its standard deviation is of order $1/\sqrt{N}$. That means that $SD(\hat{\beta}) = O(N^{-1/2})$, so $SD(\sqrt{N} \times \hat{\beta})$ converges to a constant, and therefore, does not depend on the sample size. On the contrary, $SD(\hat{\beta})$ is dependent on sample size. The parameter β itself is not design and model-dependent but $SD(\sqrt{N} \times \hat{\beta})$ is. Therefore, changing the study design will change the standardized effect size parameter.

Although I do not have any major technical comments, I have noticed some instances where articles (a, an, the) are either missing or incorrectly placed.

Thank you, we've reviewed the paper for these and other grammatical errors.

Referee #5 (Remarks on code availability):

I skimmed through the codes but haven't tried to reproduce it.